# Approximation Error Upper and Lower Bounds for Hölder Class with Transformers

**Xin He** [1]  **Yuling Jiao** [2][3]  **Xiliang Lu** [1][3]  **Jerry Zhijian Yang** [1][3]

## Abstract

We explore the expressive power of Transformers by establishing precise approximation error upper and lower bounds for Hölder class. Specifically, a new approximation upper bound is derived for the standard Transformer architecture equipped with Softmax operators, ReLU activation functions, and residual connections. We prove that a Transformer network composed of at most $\mathcal{O}(\varepsilon^{-d_0/\alpha})$ blocks can approximate any bounded Hölder function with $d_0$-dimensional input and smoothness $\alpha \in (0, 1]$ under any accuracy $\varepsilon > 0$. In the case of approximation lower bounds, leveraging the VC-dimension upper bound, we are the first to rigorously prove that Transformers demand for at least $\Omega(\varepsilon^{-d_0/(4\alpha)})$ blocks to achieve the $\varepsilon$ approximation accuracy. As a final step, we extend the derived results for standard Transformers to a general regression task and establish the corresponding excess risk rates demonstrating Transformers' empirical effectiveness in real-world settings.

## 1. Introduction

The self-attention based Transformer (Vaswani et al., 2017) has become ubiquitous in deep learning, driving remarkable progress in natural language processing (NLP) through the models like GPT (Ghojogh & Ghodsi, 2020). It has also emerged as a powerful alternative to complex neural networks like Convolutional Neural Networks (CNN) (Dosovitskiy et al., 2021) and Graph Neural Networks (GNN) (Ying et al., 2021) in image and graph processing tasks. Furthermore, Transformers have become a fundamental component in generative AI systems such as Gemini (Team et al., 2023) and latest GPT-series (Achiam et al., 2023). Given the

widespread practical success, interpretability analysis of Transformers has gained significant attention recently.

The approximation theory (Yarotsky, 2017) is one of the essential aspects of interpretability theory. By examining the error bounds of approximating certain target function classes with particular models, researchers can investigate why and how modern models are well-suited to practical tasks, which helps reveal the expressive power inherent in model structures. Various approximation studies have been conducted for Transformers, including the Universal Approximation Theorem (UAT) (Yun et al., 2020; Kajitsuka & Sato, 2024; Petrov et al., 2024) for Transformers, further UAT under constraints (Kratsios et al., 2022; Kratsios, 2023), memorization capacity of Transformers (Kim et al., 2023; Mahdavi et al., 2024), and rough approximation bounds for Transformer variants (Gurevych et al., 2022; Takakura & Suzuki, 2023; Jiao et al., 2024; 2025; 2026; Xu & Sato, 2025; Hu et al., 2025a;b).

As originally proposed in (Vaswani et al., 2017), a standard Transformer block consists of a self-attention layer with Softmax operator, a feed-forward layer with ReLU activation function, and residual connections in each layer. However, there are almost only UAT results for standard Transformers incorporating these three elements (Yun et al., 2020; Kajitsuka & Sato, 2024). The approximation bounds (or rates) characterizing quantitative relationships between error and complexities of standard Transformers are lacking. Many studies modify Transformers by replacing the Softmax operator with other operators, such as Hardmax operator (Yun et al., 2020; Hu et al., 2024; Xu & Sato, 2025) and Maximum operator (Gurevych et al., 2022; Jiao et al., 2024). Although both Yun et al. (2020) and Hu et al. (2024) transformed the Hardmax into the Softmax by introducing a large multiplicative factor in the Softmax operator, this approach actually leads to extremely large norms of Transformer weight matrices and causes the Lipschitz constants of these transformed layers to explode. Consequently, it will be challenging to maintain a proper trade-off between approximation and statistical errors. Moreover, the Maximum operator differs fundamentally from Softmax in form and function, limiting the result to approximation tasks with Transformer variants (Gurevych et al., 2022; Jiao et al., 2024). Further studies established approximation bounds

[1]School of Mathematics and Statistics, Wuhan University, Wuhan, China [2]School of Artificial Intelligence, Wuhan University, Wuhan, China [3]Hubei Key Laboratory of Computational Science, Wuhan University, Wuhan, China. Correspondence to: Xiliang Lu <xllv.math@whu.edu.cn>.

*Proceedings of the 43rd International Conference on Machine Learning*, Seoul, South Korea. PMLR 306, 2026. Copyright 2026 by the author(s).

*Table 1.* Related results on the approximation theory of Transformers.

| Reference | Function Class | Upper Bound | Lower Bound | Remark |
|---|---|---|---|---|
| (Yun et al., 2020; Kajitsuka & Sato, 2024) | Continuous | UAT (Universal) | | Transformations in operators |
| (Petrov et al., 2024; Kratsios, 2023) | Continuous | UAT (Universal) | | Without residual connections |
| (Xu & Sato, 2025) | Continuous | $\checkmark$ | $\times$ | Variant with Hardmax |
| (Takakura & Suzuki, 2023) | $\gamma$-smooth (Besov) | $\checkmark$ | $\times$ | Coordinate-wise rate |
| (Hu et al., 2025a;b) | Lipschitz continuous | $\checkmark$ | $\times$ | Without residual connections |
| (Jiao et al., 2025; 2026) | Hölder class | $\checkmark$ | $\times$ | Without residual connections |
| **Ours** | **Hölder class** | $\checkmark$ | $\checkmark$ | **Standard Transformers** |

for Softmax Transformers across various function spaces. Notably, theoretical guarantees have been developed for $\gamma$-smooth (Besov-like) functions (Takakura & Suzuki, 2023), Lipschitz continuous functions (Hu et al., 2025a;b), and the broader Hölder class (Jiao et al., 2025; 2026). However, a prevailing limitation across all these analyses is their reliance on Transformer variants without residual connections in the feed-forward layers. Therefore, approximation upper bounds for general function spaces (e.g., Hölder class) with the standard architectures are lacking. Furthermore, the approximation theory of Transformers remains incomplete — there are almost no exact lower bounds further analyzing and assessing existing upper bounds. This paper aims to solve these problems, with detailed comparisons provided in Table 1.

Our work mainly provides precise approximation error upper and lower bounds for Hölder class with standard Transformers, delivering **three fundamental contributions**:

*i*) **A complete framework for approximation error upper bounds for Hölder class with standard Transformers.** The expressive power of standard architectures is characterized through the approximation theory without relying on model transformations, addressing two key challenges:

- Understanding of the capability of self-attention layers. Building on the significant single-layer Softmax-based contextual mapping in (Kajitsuka & Sato, 2024), we avoid transformations between Hardmax and Softmax operators, and recognize that the self-attention mechanism serves to memorize and differentiate tokens within input sequences.

- Preservation of all residual connections. Residual connections are crucial in practical tasks but remain difficult to analyze theoretically due to the deep composition of blocks. We retain all residual connections by employing a novel network construction strategy. This ensures that our theoretical results can be applied directly to the standard architectures used in practice and thereby reflect Transformers' inductive bias.

*ii*) **Complementary approximation lower bounds**. We prove an approximation lower bound for standard Transformers by deriving a VC-dimension upper bound determined by the total number of model operations and parameters, based on (Anthony & Bartlett, 2009). This derived lower bound opens a new direction for future research, advancing the approximation theory of Transformers.

*iii*) **Application on a general regression task.** We investigate the applicability of Transformers in finite-sample regression tasks of Hölder functions. The theoretical analysis demonstrates Transformer's strong approximation capabilities and empirical effectiveness in real-world settings.

## 2. Related Work

**Universality and Approximation Upper Bounds.** Analogous to classical results that sufficiently large Feed-forward Neural Networks (FNNs) can accurately approximate broad function classes (Cybenko, 1989; Hornik et al., 1989), Yun et al. (2020) derived the first Universal Approximation Theorem (UAT) for standard Transformers on continuous sequence-to-sequence functions. Their work introduced a widely adopted framework which treats the self-attention mechanism as a contextual mapping and approximates an intermediate task of piece-wise constant functions using Transformers. Subsequent UAT work (Kajitsuka & Sato, 2024) streamlined (Yun et al., 2020)'s methodology by designing a single layer contextual mapping but modifying the architectures into Transformer variants without residual connections in feed-forward layers. UATs for constrained Transformers were established by (Kratsios et al., 2022; Kratsios, 2023; Petrov et al., 2024). To align the theoretical analysis of Transformers with that of FNNs, precise approximation upper bounds on model complexity are required. Upper bounds of a Transformer variant replacing softmax with Maximum operators were derived by (Gurevych et al., 2022; Jiao et al., 2024). Subsequently, Takakura & Suzuki (2023) developed a coordinate-wise rate for the specific $\gamma$-smooth function class (a Besov-like space) by treating the self-attention layer as a feature extractor for inputs, a formulation limiting the scope of their results. Based on (Kajitsuka & Sato, 2024)'s methodology, Hu et al. (2025a;b) obtained upper bounds for Lipschitz continuous functions,

which which were subsequently extended to the broader Hölder class by Jiao et al. (2025). More recently, Jiao et al. (2026) achieved bounds for Hölder class through a distinctly different approach based on the Kolmogorov-Arnold Superposition Theorem. All the aforementioned works rely on Transformer variants that omit residual connections in the feed-forward layers. Notably, our construction of Transformers is inspired by these mentioned works excluding (Takakura & Suzuki, 2023) and (Jiao et al., 2026), and improves upon (Yun et al., 2020; Kajitsuka & Sato, 2024)'s framework to derive an approximation upper bound for Hölder class with standard Transformers.

**Approximation Lower Bounds.** Approximation lower bounds determine whether corresponding upper bounds are optimal, thus guiding future research directions. Yarotsky (2017); Yarotsky & Zhevnerchuk (2020) proved that their obtained approximation upper bounds of ReLU FNNs on Sobolev functions are nearly optimal (up to logarithmic factors) by deriving approximation lower bounds. Their analysis builds upon the VC-dimension upper bounds provided in (Anthony & Bartlett, 2009, Theorems 8.7-8.8). Furthermore, Shen et al. (2022) established the optimality (up to constants) of derived approximation upper bounds for ReLU FNNs on continuous functions, by leveraging the nearly-tight VC-dimension bounds of (Bartlett et al., 2019). These results indicate the study of approximation bounds for ReLU FNNs has reached a mature stage. In recent years, lower bounds have also been established for other networks like Sigmoid networks (Barron, 1993; Siegel & Xu, 2024) and CNNs (Zhou, 2020; Yang et al., 2025). However, to the best of our knowledge, no lower bounds have been derived for Transformers until now. By analyzing the key VC-dimension upper bound provided by (Anthony & Bartlett, 2009, Theorem 8.14), we may be the first to establish an approximation lower bound for deep Transformers.

**Memorization Capacity of Transformers.** Alternatively, theoretical memorization capacity of networks also contributes to appropriate model size selections, aligning with the principles of approximation bounds. Related studies have explored this for various architectures, including FNNs (Baum, 1988; Huang, 2003; Park et al., 2021; Vardi et al., 2022), CNNs (Nguyen & Hein, 2018), and Transformers (Kim et al., 2023; Mahdavi et al., 2024; Kajitsuka & Sato, 2024; 2025). Recently, Kim et al. (2023) proved that Transformers consisting of ReLU activation feed-forward layers and a single-head attention layer, can memorize $N$ sequences of length $n$ with $d$-dimensional tokens using $\tilde{\mathcal{O}}(d + n + \sqrt{nN})$[1] parameters. Meanwhile, Mahdavi et al. (2024) isolated the role of multi-head attention, showing

---

[1]$\tilde{\mathcal{O}}(\cdot)$ is used to hide logarithmic factors, while $\mathcal{O}(\cdot)$ ignores constant factors.

that a single $H$-head attention layer with $\mathcal{O}(Hd^2)$ parameters can memorize $\mathcal{O}(Hn)$ examples. Refining these results, Kajitsuka & Sato (2025) derived nearly optimal memorization rates, demonstrating that Transformers can memorize $N$ sequences of length $n$ using $\tilde{\mathcal{O}}(\sqrt{nN})$ parameters. Results of (Kim et al., 2023; Kajitsuka & Sato, 2025) build on the concept of contextual mapping introduced in (Yun et al., 2020), which plays a pivotal role in understanding Transformer memorization capacity. While our work does not directly analyze this aspect, future research leveraging our architectural insights might lead to advancements.

## 3. Preliminaries

**Notations.** Let $\mathbb{N}_+ := \{1, 2, 3, \ldots\}$ denote the set of positive integers with $\mathbb{N} := \mathbb{N}_+ \cup \{0\}$. Let $[n]$ denote the set $\{1, 2, \ldots, n\}$ for $n \in \mathbb{N}_+$. We denote the set of real numbers by $\mathbb{R}$ and the non-negative subset by $\mathbb{R}^+$. We write $A \lesssim B$ and $B \gtrsim A$ to denote $A \leq cB$ for some absolute constant $c > 0$. $A \asymp B$ means $A \lesssim B$ and $A \gtrsim B$. Let $\mathbb{1}_d$ and $\mathbb{1}_{d \times L}$ represent the $d$-dimensional vector and the $d \times L$ matrix with all one entries. For a vector $x \in \mathbb{R}^d$, $\|x\|_p$ denotes the $\ell_p$-norm for $p \in [1, +\infty)$. For a matrix $X \in \mathbb{R}^{d \times L}$, $\|X\|_2$ and $\|X\|_F$ denote the 2-norm and Frobenius norm respectively. For a scalar measurable function $f : \mathbb{R}^d \to \mathbb{R}$, its $L_2$-norm is $\|f\|_2 := (\int (f(x))^2 \mathrm{d}x)^{1/2}$. Similarly, for a sequence-to-sequence (seq-to-seq) function $f : \mathbb{R}^{d \times L} \to \mathbb{R}^{d \times L}$, we define $\|f\|_2 := (\int \|f(X)\|_F^2 \mathrm{d}X)^{1/2}$. Appendix A.1 summarizes the notations for quick-reference and cross-checking.

**Target Function Space.** For the multi-index $\boldsymbol{n} = (n_1, \ldots, n_d) \in \mathbb{N}^d$, the $\boldsymbol{n}$-derivative of $f$ is denoted as $\partial^{\boldsymbol{n}} f := (\frac{\partial}{\partial x_1})^{n_1} \cdots (\frac{\partial}{\partial x_d})^{n_d} f$. We use the convention that $\partial^{\boldsymbol{n}} f = f$ if $\|\boldsymbol{n}\|_1 = 0$.

**Definition 3.1** (Hölder Class)**.** Let $d_x, d_y, K \in \mathbb{N}_+, \mathcal{X} \subseteq \mathbb{R}^{d_x}$ and $\alpha = r + \alpha_0 > 0$, where $r = \lfloor \alpha \rfloor$ and $\alpha_0 \in (0, 1]$, we define the Hölder class with smoothness index $\alpha$ and norm constraint parameter $K$ as

$$\mathcal{H}_{d_x, d_y}^{\alpha}(\mathcal{X}, K) := \Big\{ f = (f_1, \ldots, f_{d_y})^\top : \mathcal{X} \to \mathbb{R}^{d_y} \mid$$

$$\max_{\|\boldsymbol{n}\|_1 \leq r} \sup_{x \in \mathcal{X}} |\partial^{\boldsymbol{n}} f_k(x)| \leq K,$$

$$\max_{\|\boldsymbol{n}\|_1 = r} \sup_{x, y \in \mathcal{X}, x \neq y} \frac{|\partial^{\boldsymbol{n}} f_k(x) - \partial^{\boldsymbol{n}} f_k(y)|}{\|x - y\|_2^{\alpha_0}} \leq K, \ k \in [d_y] \Big\}.$$

Without loss of generality, we consider a bounded input domain $\mathcal{X} = [0, 1]^{d_0}$ in this paper. Furthermore, we use $\mathcal{H}_{d_0}^{\alpha}(\mathcal{X}, K) := \mathcal{H}_{d_0, d_0}^{\alpha}(\mathcal{X}, K)$ when $d_x = d_y = d_0$.

**Transformer Architecture.** *i) A Transformer block* consists of a dot-product Softmax self-attention layer, a ReLU feed-forward layer, and a residual connection per layer.

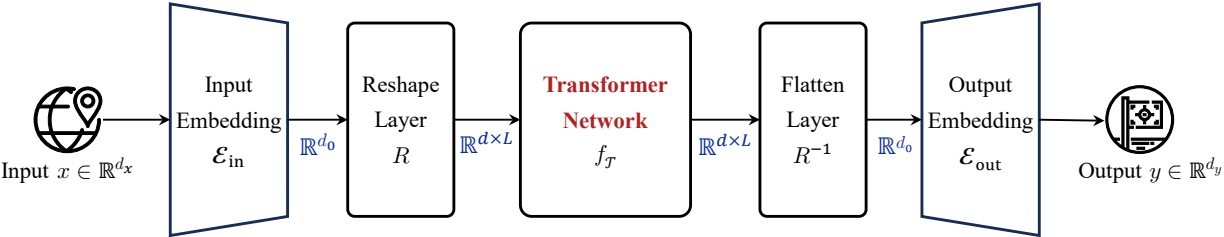

*Figure 1.* Pipeline of the standard Transformer function class: bridging the vector form and sequence form.

For the input sequence $X \in \mathbb{R}^{d \times L}$ with $L$ length and $d$-dimensional tokens, a Transformer block mapping from $\mathbb{R}^{d \times L}$ to $\mathbb{R}^{d \times L}$ is represented as

$$\text{Attn}(X) = X + \sum_{i=1}^{r_1} W_O^i W_V^i X \sigma_S \left[ (W_K^i X)^\top W_Q^i X \right], \quad (1)$$

$$\text{FF} \circ \text{Attn}(X) = \text{Attn}(X) +$$
$$+ \sum_{i=1}^{r_2} W_2^i \sigma_R \left[ W_1^i \cdot \text{Attn}(X) + b_1^i \mathbb{1}_L^\top \right] + b_2 \mathbb{1}_L^\top, \quad (2)$$

where $W_K^i, W_Q^i, W_V^i \in \mathbb{R}^{m \times d}$ are the key, query and value weight matrices, $W_O^i \in \mathbb{R}^{d \times m}$ is the connection matrix, and $W_1^i \in \mathbb{R}^{l \times d}, W_2^i \in \mathbb{R}^{d \times l}, b_1^i \in \mathbb{R}^l, b_2 \in \mathbb{R}^d$ parameterize the feed-forward layer. The Softmax operator $\sigma_S$ applies column-wisely with the ReLU activation $\sigma_R$ applying element-wisely. Indices $r_1$ and $r_2$ are the head number of attention layer and the neuron number of feed-forward layer, respectively.

*ii) A Transformer network (encoder)* is a composition of $D \in \mathbb{N}_+$ Transformer blocks additionally with a positional encoding $E \in \mathbb{R}^{d \times L}$, defined as

$$\mathcal{T}^D := \left\{ f_{\mathcal{T}} : \mathbb{R}^{d \times L} \to \mathbb{R}^{d \times L} \right\}, \quad \text{where}$$
$$f_{\mathcal{T}}(X) = \text{FF}^{(D)} \circ \text{Attn}^{(D)} \circ \cdots \circ \text{FF}^{(1)} \circ \text{Attn}^{(1)}(X + E). \quad (3)$$

Given that the function approximation task is defined on vector inputs, whereas Transformer networks operate on sequences, we introduce a reshape layer to rearrange the input vectors to the sequence format, additionally with the reverse flatten layers.

**Definition 3.2** (Reshape and Flatten Layers). Let $R : \mathbb{R}^{d_0} \to \mathbb{R}^{d \times L}$ be a reshape layer that transforms the $d_0$-dimensional input into a $d \times L$ sequence matrix with $d_0 = dL$. Besides, the corresponding reverse flatten layer is defined by $R^{-1} : \mathbb{R}^{d \times L} \to \mathbb{R}^{d_0}$ as the inverse of $R$.

*iii) The Transformer function class* considered throughout the paper is defined as

$$\mathcal{T}_R^D := \left\{ g = \mathcal{E}_{\text{out}} \circ R^{-1} \circ f_{\mathcal{T}} \circ R \circ \mathcal{E}_{\text{in}} : \mathbb{R}^{d_x} \to \mathbb{R}^{d_y} \right\}, \quad (4)$$

where $\mathcal{E}_{\text{in}} : \mathbb{R}^{d_x} \to \mathbb{R}^{d_0}$ is the input embedding and $\mathcal{E}_{\text{out}} : \mathbb{R}^{d_0} \to \mathbb{R}^{d_y}$ is the output embedding. The overall pipeline is illustrated in Figure 1.

**Problem Setting.** Function approximation tasks of Hölder class $\mathcal{H}_{d_x,d_y}^\alpha(\mathcal{X}, K)$ with Transformer function class $\mathcal{T}_R^D$ aim to study both upper and lower bounds of the approximation error[2]

$$\mathcal{E}_{\text{app}} \left( \mathcal{H}_{d_x,d_y}^\alpha, \mathcal{T}_R^D \right) := \sup_{f \in \mathcal{H}_{d_x,d_y}^\alpha} \inf_{g \in \mathcal{T}_R^D} \|f - g\|_2. \quad (5)$$

In a mainstream way, analysis of the approximation error equals providing upper and lower bounds of Transformer complexities when the error meets the minor accuracy criterion $\varepsilon > 0$, i.e., $\mathcal{E}_{\text{app}} \leq \varepsilon$.

# 4. Approximation Error Upper Bound for Hölder Class with Transformers

Since the input and output embeddings do not impact the core architecture of Transformer networks (See Figure 1), we assume $d_x = d_y = d_0$ and omit the embeddings in this section for the sake of simplicity.

**Theorem 4.1** (Upper Bound of Approximation Error). *Let $\alpha \in (0,1], d_0 \in \mathbb{N}_+, d_0 > 2\alpha$, and $K \in \mathbb{N}_+$. For any target function $f \in \mathcal{H}_{d_0}^\alpha([0,1]^{d_0}, K)$ and any error criterion $\varepsilon > 0$, there exists a Transformer function $g \in \mathcal{T}_R^D$ composed of at most $\mathcal{O}(\varepsilon^{-\frac{d_0}{\alpha}})$ blocks such that $\|f - g\|_2 < \varepsilon$.*

*Proof Sketches.* We follow the basic pipeline of (Yun et al., 2020, Theorem 2) but introduce differences in constructions to ensure results for standard Transformers. The proof is divided roughly into three steps.

### Step 1. Transfer to an intermediate approximation task.

*i)* For the target Hölder class $\mathcal{H}_{d_0}^\alpha$, it can be reshaped into an equivalent sequence function class $\bar{\mathcal{H}}_{d,L}^\alpha$ with $d_0 = dL$.

*ii)* We approximate any $\bar{f} \in \bar{\mathcal{H}}_{d,L}^\alpha$ by a piece-wise constant function $f_\delta \in \mathcal{H}(\delta)$ with cell width $\delta = \mathcal{O}(\varepsilon^{\frac{1}{\alpha}})$ and separation width $\delta^* = \mathcal{O}(\varepsilon^{\frac{2}{d_0} + \frac{1}{\alpha}})$, obtaining the error $\|\bar{f} - f_\delta\|_2 < \frac{\varepsilon}{2}$ based on the regularity of Hölder class.

### Step 2. Approximate $\mathcal{H}(\delta)$ by designed Transformers.

---

[2] The approximation error characterizes the fundamental limit of model's expressivity, For further details, refer to Remark D.2.

For any $f_\delta \in \mathcal{H}(\delta)$, we approximate it under $\frac{\varepsilon}{2}$ error with the standard Transformer network $f_\mathcal{T} = f_{\mathcal{T}3} \circ f_{\mathcal{T}2} \circ f_{\mathcal{T}1} \in \mathcal{T}^D$ having three main parts,

*i*) Quantization module $f_{\mathcal{T}1}$ is composed of $\mathcal{O}(\varepsilon^{-\frac{1}{\alpha}})$ feed-forward layers and a positional encoding $E \in \mathbb{R}^{d \times L}$, maps most of continuous inputs $X \in \bar{\mathcal{X}} = [0,1]^{d \times L}$ into discrete grid points, and constructs a positional grid set $\mathcal{G}_{\delta,\delta^*}^p$.

*ii*) Contextual mapping $f_{\mathcal{T}2}$ is composed of a single self-attention layer, and enhances the separation property of points in $\mathcal{G}_{\delta,\delta^*}^p$ by generating unique contextual ids $f_{\mathcal{T}2}(G)$ for grid points $G \in \mathcal{G}_{\delta,\delta^*}^p$.

*iii*) Value mapping $f_{\mathcal{T}3}$ is composed of $\mathcal{O}(\varepsilon^{-\frac{d_0}{\alpha}})$ feed-forward layers and maps grid points to the target outputs, taking the contextual ids as anchor values.

**Step 3. Verification of the final error.** By triangle inequality, we have $\|\bar{f} - f_\mathcal{T}\| \leq \|\bar{f} - f_\delta\| + \|f_\delta - f_\mathcal{T}\| < \varepsilon$. Furthermore, with the reshape and flatten layers $R, R^{-1}$ and equivalent norm transformations, the final result $\|f - g\|_2 < \varepsilon$ is obtained. $\square$

Theorem 4.1 establishes that a standard Transformer, composed of at most $\mathcal{O}(\varepsilon^{-\frac{d_0}{\alpha}})$ blocks with a single active self-attention layer, serves as an effective approximator of Hölder functions. Increasing the Transformer blocks enhances the model's expressive power. The primary challenge in approximating Hölder functions arises from the ratio between input dimension $d_0$ and smoothness index $\alpha$. Accordingly, we assume $d_0 > 2\alpha$ to ensure a positive minor term $\delta^*$, which is discussed later. Furthermore, the norm constraint $K$ for the Hölder class primarily serves as a stability condition for proof construction, independent of both the block number $D$ and error $\varepsilon$.

**Remark 4.2** (Bounded Width). Notably, our construction employs a fixed-width architecture where inner dimensions $\{r_1 \cdot m, r_2 \cdot l\}$ are bounded by $\mathcal{O}(d_0)$, independent of both the smoothness $\alpha$ and the target accuracy $\varepsilon$. Therefore, the approximation bound is characterized exclusively by the block number $D$.

**Corollary 4.3** (Depth-width Trade-off). *For any $\varepsilon > 0$ and $d_0 = dL$, a Transformer architecture $\mathcal{T}_R^D$ with $D = \mathcal{O}(\varepsilon^{-d_0/\alpha})$ blocks and fixed width $W = 4d$, which is capable of approximating any function $f \in \mathcal{H}_{d_0}^\alpha([0,1]^{d_0}, K)$ with smooth index $\alpha \in (0,1]$ under error $\varepsilon$, can be realized by a Transformer $\mathcal{T}_R^{D', W'}$ comprising variable blocks $D'$, width $W'$, and two additional linear embedding layers, satisfying the condition $D' \cdot W' = \mathcal{O}(\varepsilon^{-d_0/\alpha})$, where $D' \geq 6$ and $W' \geq 4d$.*

This corollary reveals a flexible depth-width trade-off. Rather than strictly adhering to the fixed-width regime in Theorem 4.1, one can achieve the same bound using a fixed-depth architecture with variable width, or more generally, an architecture where both depth and width vary simultaneously, as long as the product satisfies $D' \cdot W' = \mathcal{O}(\varepsilon^{-d_0/\alpha})$.

The following sections highlight the distinctive features of our Transformer architectures for approximation tasks. See Appendix B for full details and proofs.

## 4.1. Approximation for Hölder Class by Piece-wise Constant Function Class

For any target function $f \in \mathcal{H}_{d_0}^\alpha(\mathcal{X}, K)$, we denote the reshaped input set as

$$\bar{\mathcal{X}} := \left\{ X = R(x) \mid X \in \mathbb{R}^{d \times L}, x \in \mathcal{X}, x = R^{-1}(X) \right\},$$

where $R$ and $R^{-1}$ are the reshape and flatten layers. We denote the reshaped target Hölder class by

$$\bar{\mathcal{H}}_{d,L}^\alpha := \left\{ \bar{f} = R \circ f \circ R^{-1} : \mathbb{R}^{d \times L} \to \mathbb{R}^{d \times L} \right\}. \quad (6)$$

Now we define the intermediate piece-wise constant class $\mathcal{H}(\delta)$ approximating the reshaped Hölder class $\bar{\mathcal{H}}_{d,L}^\alpha$.

**Definition 4.4** (Quantization Grid and Cube). Let $d, L \in \mathbb{N}_+$. We set the cell width $\delta > 0$ and the separation width $\delta^* > 0$. Then the number of grid points per dimension can be defined as $M = \lfloor \frac{1}{\delta + \delta^*} \rfloor$. And the grid point set $\mathcal{G}_{\delta,\delta^*}$ over $[0,1]^{d \times L}$ is defined as $\mathcal{G}_{\delta,\delta^*} := \{0, \delta + \delta^*, 2(\delta + \delta^*), \ldots, (M-1)(\delta + \delta^*)\}^{d \times L}$. For any grid point $G \in \mathcal{G}_{\delta,\delta^*}$, its associated hypercube is $\mathcal{S}_G := \prod_{j \in [d], k \in [L]} [G_{j,k}, G_{j,k} + \delta]$.

By construction, the union of these hypercubes satisfies $\bigcup_{G \in \mathcal{G}_{\delta,\delta^*}} \mathcal{S}_G \subset [0,1]^{d \times L}$. And any two cells maintain a separation margin of at least $\delta^*$ along each coordinate. Formally, for any $G_{j,k} \neq G'_{j,k} \in \mathcal{G}_{\delta,\delta^*}$, their corresponding cells satisfy $d_{\min}([G_{j,k}, G_{j,k} + \delta], [G'_{j,k}, G'_{j,k} + \delta]) \geq \delta^*$. This separation margin is crucial for our constructive proof for ReLU-based Transformers, as it guarantees the necessary geometric margin for the subsequent theoretical analysis.

**Definition 4.5** (Piece-wise Constant Function Class). Let $\mathbb{1}\{\cdot\}$ denote the indicator function. For each grid point $G \in \mathcal{G}_{\delta,\delta^*}$, its hypercube $\mathcal{S}_G$, and its output matrix $Y_G \in \mathbb{R}^{d \times L}$, the piece-wise constant function class with $X \in [0,1]^{d \times L}$ is defined as

$$\mathcal{H}(\delta) := \left\{ f_\delta = \sum_{G \in \mathcal{G}_{\delta,\delta^*}} Y_G \cdot \mathbb{1}\{X \in \mathcal{S}_G\} \right\}. \quad (7)$$

**Lemma 4.6** (Approximation Error between $\bar{\mathcal{H}}_{d,L}^\alpha$ and $\mathcal{H}(\delta)$). *Let $\alpha \in (0,1], d_0 \in \mathbb{N}_+$ with $d_0 = dL > 2\alpha$. For any given $\varepsilon > 0$ and a target function $\bar{f} \in \bar{\mathcal{H}}_{d,L}^\alpha$, there exists an $f_\delta \in \mathcal{H}(\delta)$ with $\delta = \mathcal{O}(\varepsilon^{\frac{1}{\alpha}})$ and $\delta^* = \mathcal{O}(\varepsilon^{\frac{2}{d_0} + \frac{1}{\alpha}})$ such that $\|\bar{f} - f_\delta\|_2 \leq \varepsilon$.*

The inequality condition $d_0 > 2\alpha$ can be easily satisfied since $\alpha \in (0,1]$, with its purpose to ensure that $\delta^*$ remains

strictly positive in the proof. In a sense, this highlights that our work can handle a harder approximation task with large input dimension $d_0$ and small function smoothness $\alpha$.

## 4.2. Approximation for Piece-wise Constant Function Class by Standard Transformers

To approximate the piece-wise constant function class $\mathcal{H}(\delta)$, the Transformer network $f_{\mathcal{T}} \in \mathcal{T}^D$ comprises three primary parts: the quantization module, contextual mapping and value mapping.

**Lemma 4.7** (Quantization Module $f_{\mathcal{T}1}$). *Let $\delta, \delta^* \in (0,1)$, $M = \lfloor \frac{1}{\delta+\delta^*} \rfloor$ and $d_0 = dL$. By processing the inputs $X \in \bigcup_{G \in \mathcal{G}_{\delta,\delta^*}} \mathcal{S}_G$, there exists a quantization module $f_{\mathcal{T}1}$ composed of $d_0 M$ feed-forward layers and a positional encoding $E = \mathbb{1}_d \cdot [1, 2, \ldots, L]^\top$, constructing a quantization positional grid $\mathcal{G}^p_{\delta,\delta^*} = \mathcal{G}_{\delta,\delta^*} + E$.*

The quantization module $f_{\mathcal{T}1}$ maps most of received continuous input sequences from $[0,1]^{d \times L}$ to discrete grid points in $\mathcal{G}^p_{\delta,\delta^*}$. For input sequences in the complement set $[0,1]^{d \times L} \setminus \bigcup_{G \in \mathcal{G}_{\delta,\delta^*}} \mathcal{S}_G$, outputs of $f_{\mathcal{T}1}$ remain controllable, since $f_{\mathcal{T}1}$ is bounded and the measure of the complement set is $(1 - \delta \lfloor \frac{1}{\delta+\delta^*} \rfloor)^{d_0}$ which will be negligible with sufficiently small $\delta^*$.

To construct the next contextual mapping $f_{\mathcal{T}2}$ based on the positional grid $\mathcal{G}^p_{\delta,\delta^*}$, we first introduce several necessary concepts.

**Definition 4.8** (Token Vocabulary). *Let $X^{(1)}, \ldots, X^{(N)} \in \mathbb{R}^{d \times L}$ be $N$ input sequences consisting of $L$ tokens and $d$ embedding dimension. The $i$-th vocabulary set is $\mathcal{V}^{(i)} = \bigcup_{k \in [L]} X^{(i)}_{:,k} \subset \mathbb{R}^d$ for $i \in [N]$, and the whole vocabulary set is defined as $\mathcal{V} = \bigcup_{i \in [N]} \mathcal{V}^{(i)} \subset \mathbb{R}^d$.*

**Definition 4.9** (Token-wise Separatedness). *Let $r_{\min}, r_{\max}, \beta \in \mathbb{R}_+$. The $N$ input sequences $X^{(1)}, \ldots, X^{(N)} \in \mathbb{R}^{d \times L}$ are token-wise $(r_{\min}, r_{\max}, \beta)$-separated if the following conditions hold,*

1. *For any $i \in [N]$ and $k \in [L]$, the boundedness $r_{\min} \leq \left\| X^{(i)}_{:,k} \right\|_2 \leq r_{\max}$ holds.*

2. *For any $i, j \in [N]$ and $k, l \in [L]$ with $X^{(i)}_{:,k} \neq X^{(j)}_{:,l}$, $\left\| X^{(i)}_{:,k} - X^{(j)}_{:,l} \right\|_2 \geq \beta$ holds.*

Definition 4.9 only considers separatedness among different tokens. As first clarified in (Yun et al., 2020), sequence models like Transformers should distinguish between different input sequences sharing same tokens. To address this limitation, we adopt the essential single-layer contextual mapping in (Kajitsuka & Sato, 2024), enabling Transformers to capture whole context of each sequence and generate unique context ids.

**Lemma 4.10** (Contextual Mapping $f_{\mathcal{T}2}$). *All the $M^{d_0}$ grid points $G \in \mathcal{G}^p_{\delta,\delta^*}$ are $(\sqrt{d}, \sqrt{d}(L+1), \sqrt{d}(\delta+\delta^*))$-separated. A single self-attention layer equipped with one Softmax head, can be an $(r, \gamma)$-contextual mapping $f_{\mathcal{T}2} : \mathbb{R}^{d \times L} \to \mathbb{R}^{d \times L}$ for the input sequences $G$, satisfying*

1. *For any $i \in [M^{d_0}]$ and $k \in [L]$, the boundedness $\left\| f_{\mathcal{T}2}(G^{(i)})_{:,k} \right\|_2 \leq r$ holds.*

2. *For any $i, j \in [M^{d_0}]$ and $k, l \in [L]$ such that either $\mathcal{V}^{(i)} \neq \mathcal{V}^{(j)}$ or $G^{(i)}_{:,k} \neq G^{(j)}_{:,l}$, the separatedness $\left\| f_{\mathcal{T}2}(G^{(i)})_{:,k} - f_{\mathcal{T}2}(G^{(j)})_{:,l} \right\|_2 \geq \gamma$ holds.*

*with $r$ and $\gamma$ defined by $r = \sqrt{d}(L+1) + \frac{\sqrt{d}(\delta+\delta^*)}{4}$ and $\gamma = \tilde{\mathcal{O}}\left( \exp\left( -\frac{L^6 M^{4d_0}}{\delta+\delta^*} \right) \right)$.*

The case $\mathcal{V}^{(i)} \neq \mathcal{V}^{(j)}$ in the second condition includes the scenario of different sequences sharing some of same tokens. Consequently, $f_{\mathcal{T}2}$ maps all columns of grid points $G \in \mathcal{G}^p_{\delta,\delta^*}$ to the unique vectors named as contextual ids. With the help of the contextual mapping $f_{\mathcal{T}2}$, next step of designing a one-to-one value mapping is feasible.

**Lemma 4.11** (Value Mapping $f_{\mathcal{T}3}$). *Suppose $Y_G$ is the output matrix of $G$. Let $\delta, \delta^* \in (0,1), M = \lfloor \frac{1}{\delta+\delta^*} \rfloor, d, L \in \mathbb{N}_+$ and $d_0 = dL$, there exists a value mapping $f_{\mathcal{T}3} : \mathbb{R}^{d \times L} \to \mathbb{R}^{d \times L}$ composed of $2LM^{d_0} + 1$ feed-forward layers such that $f_{\mathcal{T}3} \circ f_{\mathcal{T}2}(G) = Y_G, \forall G \in \mathcal{G}^p_{\delta,\delta^*}$.*

Anchored on the unique contextual id $f_{\mathcal{T}2}(G)_{:,k}, k \in [L]$, a subunit composed of two feed-forward layers can map this column to the target $Y_{G:,k}$. This indicates that our constructions of $LM^{d_0}$ subunits identify target positions of inputs and assign the corresponding output values. Additionally, the extra layer processes the accumulated calculation results by preceding subunits.

**Proposition 4.12** (Verification). *Set $\varepsilon > 0$ and any $\bar{f} \in \bar{\mathcal{H}}^\alpha_{d,L}$ with $d = d_0 L$. By choosing $\delta = \mathcal{O}(\varepsilon^{\frac{1}{\alpha}})$ and $\delta^* = \mathcal{O}(\varepsilon^{\frac{2}{d_0}+\frac{1}{\alpha}})$, there exists a Transformer network $f_{\mathcal{T}}$ composed of at most $\mathcal{O}(\varepsilon^{-\frac{d_0}{\alpha}})$ blocks such that $\left\| \bar{f} - f_{\mathcal{T}} \right\|_2 < \varepsilon$.*

This result is the reshaped version of Theorem 4.1, see proof in Appendix B.4.

## 4.3. Novelty of Constructions

The novelty of our construction is two-fold. A direct comparison with prior methods is provided in Table 3.

*i*) For the quantization module $f_{\mathcal{T}1}$ and value mapping $f_{\mathcal{T}3}$ composed of feed-forward layers, our constructions are novel and unique since all residual connections and ReLU operators are kept, thereby reflecting the inductive bias of standard Transformers used in practice. For the constructions detail, refer to Appendix B.3.1 & B.3.3.

*ii*) For the contextual mapping $f_{\mathcal{T}2}$ composed of a single standard Softmax-based self-attention layer, the construction has shown that one activated attention layer is sufficient for strong theoretical approximation power. This result underscores the essential role of the attention mechanism's separation enhancement property in distinguishing tokens.

To complement our theoretical framework, further discussions on the connections between theoretical assumptions and practice are provided in Appendix B.5.

# 5. Approximation Lower Bound with VC-dimension

In this section, we derive the approximation error lower bound via a VC-dimension upper bound of the Transformer.

**Definition 5.1** (VC-dimension). Given an indicator function class $\mathcal{M}$ from $\mathcal{X}$ to $\{0, 1\}$. The VC-dimension of $\mathcal{M}$, denoted as $\mathrm{VCdim}(\mathcal{M})$, is the largest integer $n$ such that there exists a sample set $S = \{x_1, \ldots, x_n \mid x_i \in \mathcal{X}, i \in [n]\}$ where the restriction $\mathcal{M}|_S = \{[m(x_1), \ldots, m(x_n)] \mid m \in \mathcal{M}, x_i \in S\}$ has cardinality $|\mathcal{M}|_S| = 2^n$. If $n$ is arbitrarily large, then $\mathrm{VCdim}(\mathcal{M})$ is infinite.

In other words, the VC-dimension quantifies the hypothesis space's maximum capacity to shatter sample points. It is worth analyzing the upper bound of the VC-dimension.

**Theorem 5.2** (Anthony & Bartlett, 2009, Theorem 8.14). *Suppose $m$ is a function from $\mathbb{R}^\omega \times \mathbb{R}^{d_0}$ to $\{0, 1\}$ and let*

$$\mathcal{M} = \left\{ x \mapsto m(a, x) \mid a \in \mathbb{R}^\omega, x \in \mathbb{R}^{d_0} \right\}$$

*be the class determined by $m$ with $\omega$ parameters and input $x$. Suppose that $m$ can be computed by an algorithm that takes as input the pair $(a, x)$ and returns $m(a, x)$ after no more than $t$ operations of the following types:*

1. *the exponential function $\alpha \mapsto e^\alpha$ on real numbers.*

2. *the arithmetic operations $+$, $-$, $\times$, and $/$ on real numbers.*

3. *the jumps conditioned on $>, \geq, <, \leq, =,$ and $\neq$ comparisons of real numbers.*

4. *output $0$ or $1$.*

*Then*

$$\mathrm{VCdim}(\mathcal{M}) \leq t^2 \omega(\omega + 19 \log_2(9\omega)). \quad (8)$$

VC-dimension upper bound (8) is determined by the total number of operations and parameters constituting the model space $\mathcal{M}$. Particularly, it takes into account the exponential function operations, which captures the operational characteristics of Transformers' Softmax operators.

By convention of (Bartlett et al., 2019), for a real-valued function class $\mathcal{M}$, we extend the VC-dimension by defining $\mathrm{VCdim}(\mathcal{M}) := \mathrm{VCdim}(\mathrm{sgn}(\mathcal{M}))$ where

$$\mathrm{sgn}(\mathcal{M}) := \{\mathrm{sgn}(m) \mid m \in \mathcal{M}\}, \text{ and } \mathrm{sgn}(x) = \mathbb{1}[x > 0].$$

The application of $\mathrm{sgn}(\cdot)$ only introduces a single thresholding operation (an extra jump on ">" in item *3* of Theorem 5.2), while leaves the parameter number $\omega$ unchanged. Therefore, we can directly apply the Theorem 5.2 to the standard Transformers.

**Theorem 5.3** (Lower Bound of Approximation Error). *For any $\varepsilon > 0$, a Transformer architecture $\mathcal{T}_R^D$ with $D$ blocks, which is capable of approximating any function $f \in \mathcal{H}_{d_0}^\alpha(\mathcal{X}, K)$ with smooth index $\alpha \in (0, 1]$ under error $\varepsilon$, must possess $t$ operations, $\omega$ parameters and $D$ blocks satisfying*

$$D^4 \gtrsim t^2 \omega(\omega + 19 \log_2(9\omega)) \geq \mathrm{VCdim}(\mathcal{T}_R^D) \gtrsim \varepsilon^{-\frac{d_0}{\alpha}}. \quad (9)$$

*Equivalently, the block number $D \gtrsim \varepsilon^{-\frac{d_0}{4\alpha}}$.*

*Proof Sketches.* The proof is divided into two steps. For details, see Appendix C.

**Step 1. VC-dimension of Transformers.** Due to the detail-oriented structure of constructed Transformers for the approximation upper bound, we can directly derive that $t$ and $\omega$ share the same order of magnitude as the block number $D$, i.e., $t \sim \omega \sim D$. With Theorem 5.2, we have $\mathrm{VCdim}(\mathcal{T}_R^D) \lesssim D^4$.

**Step 2. Error lower bounds.** *i*) The VC-dimension formally quantifies the maximum memorization capacity of the hypothesis space. *ii*) Once the approximation of a special Hölder classification function on $\mathcal{O}(\varepsilon^{-\frac{d_0}{\alpha}})$ points is achieved, we show that Transformers can at least shatter those points, which means $\mathrm{VCdim}(\mathcal{T}_R^D) \gtrsim \varepsilon^{-\frac{d_0}{\alpha}}$. $\square$

Theorem 5.3 establishes the theoretic limit of standard Transformers in approximation theory, which implies such architectures must possess at least $\Omega(\varepsilon^{-\frac{d_0}{4\alpha}})$ blocks.

**Tightness Analysis.** Based on Theorem 4.1 & 5.3, there is a gap between the lower and upper bounds, i.e., $\varepsilon^{-\frac{d_0}{4\alpha}} \lesssim D \lesssim \varepsilon^{-\frac{d_0}{\alpha}}$, which suggests scope for improvement:

*i*) The current token-to-token value mapping of Lemma 4.11 fails to exploit the higher smoothness of the target function class when $\alpha > 1$, and thus increases the Transformer blocks unnecessarily. This stems from the inherent challenge of constructing the piece-wise polynomials via parallel sequence feed-forward layers in standard Transformers, which is a prevalent problem in the field.

***ii***) The VC-dimension upper bound (8) should be tighter. Shen et al. (2022) has proven the optimal lower bound only for deep ReLU FNNs via a nearly-tight VC-dimension result. However, even today, upper and lower bounds of networks incorporating other operators, especially the Sigmoid-like[3] ones, remain inconsistent.

While a gap remains between the upper and lower bounds, to the best of our knowledge, this work establishes the first quantitative characterization of the approximation capability of standard Transformers, serving as a landmark that can inspire a host of subsequent studies. We leave tightening these bounds as a promising direction for future research.

**Remark 5.4** (VC-dimension Lower Bound). Note that the class of fixed-width, depth $D$ feed-forward networks, denoted by $\mathcal{F}^D$, is a strict subset of the Transformer function class $\mathcal{T}_R^D$. This structural inclusion inherently yields $\text{VCdim}(\mathcal{T}_R^D) \geq \text{VCdim}(\mathcal{F}^D)$. Since Bartlett et al. (2019) established that $\text{VCdim}(\mathcal{F}^D) = \Omega(D^2)$, it follows that $\text{VCdim}(\mathcal{T}_R^D) = \Omega(D^2)$. Therefore, fully closing the error gap to match the desired approximation lower bound of $\Omega(\varepsilon^{-d_0/(2\alpha)})$, a tighter VC-dimension upper bound of Transformers approaching this $\Omega(D^2)$ rate is required.

## 6. Applications in a General Regression Task

In this section, we apply obtained results to a general regression task, demonstrating Transformer's approximation capabilities and effectiveness in real-world settings .

**Regression Tasks.** Suppose the random variable pair $z = (x, y) \sim \mathbb{P}$, where $\mathbb{P}$ is the joint distribution on $\mathcal{X} \times \mathcal{Y}$ with $\mathcal{X} = [0,1]^{d_0}$ and $\mathcal{Y} \subset \mathbb{R}$. Based on the finite i.i.d dataset $\mathbb{D} = \{z^{(1)}, \ldots, z^{(N)} \mid z^{(i)} = (x^{(i)}, y^{(i)})\}$, the regression task is to estimate the target Hölder function $f^* : \mathbb{R}^{d_0} \to \mathbb{R}$,

$$f^* = \underset{f \in \mathcal{H}_{d_0,1}^\alpha(\mathcal{X},K)}{\arg\min} \mathcal{L}(f) = \underset{f \in \mathcal{H}_{d_0,1}^\alpha(\mathcal{X},K)}{\arg\min} \mathbb{E}_z\left[(f(x) - y)^2\right],$$

where $\mathcal{L}$ is the population risk under the distribution.

With the $N$ sample pairs, we define the empirical risk minimizer $\hat{f}$ over the Transformer function class $\mathcal{T}_R^D$ (with embedding dimensions $d_x = d_0$ and $d_y = 1$) as

$$\hat{f}(x) = \underset{f \in \mathcal{T}_R^D}{\arg\min} \widehat{\mathcal{L}}(f) = \frac{1}{N} \sum_{i=1}^N \left(f(x^{(i)}) - y^{(i)}\right)^2. \quad (10)$$

By introducing an optimization solver $s(\tau)$, we obtain the actual estimator $f_{s(\tau)} \in \mathcal{T}_R^D$ such that

$$\widehat{\mathcal{L}}(f_{s(\tau)}) \leq \widehat{\mathcal{L}}(\hat{f}) + \tau, \quad (11)$$

where $\tau$ is the optimization error from the solver $s(\tau)$. Therefore, the expectation of excess risk which evaluates

the average performance of the estimator $f_{s(\tau)}$ is defined as

$$\mathbb{E}_{\mathbb{D}}\left[\mathcal{E}(f_{s(\tau)})\right] = \mathbb{E}_{\mathbb{D}}\left[\mathcal{L}(f_{s(\tau)}) - \mathcal{L}(f^*)\right]. \quad (12)$$

Further we can decompose (12) into three parts[4] like

$$\mathbb{E}_{\mathbb{D}}\left[\mathcal{E}(f_{s(\tau)})\right] \leq \mathcal{E}_{\text{sta}} + 2\mathcal{E}_{\text{app}}^2 + 2\tau, \quad (13)$$

where $\mathcal{E}_{\text{app}} = \sup_{p \in \mathcal{H}_{d_0,1}^\alpha} \inf_{f \in \mathcal{T}_R^D} \|f - p\|_2$ is the approximation error and the statistical error is defined as .

$$\mathcal{E}_{\text{sta}} := \mathbb{E}_{\mathbb{D}}[\mathcal{L}(f_{s(\tau)}) - 2\widehat{\mathcal{L}}(f_{s(\tau)}) + \mathcal{L}(f^*)] \quad (14)$$

The upper bound of $\mathcal{E}_{\text{app}}$ is given in Theorem 4.1. Now we derive the upper bound of $\mathcal{E}_{\text{sta}}$.

**Theorem 6.1** (Upper Bound of Statistical Error). *Let $\mu > 0$ and sample size $N \in \mathbb{N}_+$. For the Transformer function class $\mathcal{T}_R^D$ and target $f^* \in \mathcal{H}_{d_0,1}^\alpha([0,1]^{d_0}, K)$, the statistical error $\mathcal{E}_{\text{sta}} = \mathbb{E}_{\mathbb{D}}[\mathcal{L}(f_{s(\tau)}) - 2\widehat{\mathcal{L}}(f_{s(\tau)}) + \mathcal{L}(f^*)]$ is upper bounded by*

$$\mathcal{E}_{\text{sta}} \lesssim \frac{(16K + 8)(D^4 \log(eKND^{-4}/\mu) + 1)}{N} + 3\mu, \quad (15)$$

*where $\mu$ is the adjustable radius parameter.*

Theorem 6.1 characterizes the finite-sample error of estimator $f_{s(\tau)}$ offered by a fixed Transformer. Intuitively, this error approaches 0 as $N \to +\infty$ with a decaying $\mu$. Notably, proving this statistical error upper bound amounts to bounding the uniform covering number of Transformers given a finite sample size, which leverages our derived novel VC-dimension upper bound. (For details, refer to Appendix D.2 & Remark D.7).

**Theorem 6.2** (Convergence Rate of Excess Risk). *Assume the regression function $f^* \in \mathcal{H}_{d_0,1}^\alpha([0,1]^{d_0}, K)$ for some $\alpha \in (0,1]$ and $K > 0$. Let $f_{s(\tau)}$ be the estimator over the i.i.d samples $\{(x_i, y_i)\}_{i=1}^N$, the excess risk bound holds,*

$$\mathbb{E}_{\mathbb{D}}\left[\mathcal{E}(f_{s(\tau)})\right] - 2\tau \lesssim N^{-\frac{\alpha}{2d_0+\alpha}}\log(N) + N^{-\frac{\alpha}{2d_0+\alpha}}, \quad (16)$$

*where $\tau$ is the optimization error.*

*Proof Sketches.* With Theorem 4.1 & 6.1 and equation (13), set $\mu = D^{-\frac{2\alpha}{d_0}}$ and $D \asymp N^{\frac{d_0}{4d_0+2\alpha}}$, the result is proved. $\square$

For this regression task, (16) implies a trade-off between the approximation error (model complexity) and the statistical error (generalization ability) under finite $N$ samples. With a fixed sample size $N$, the excess risk initially decreases and subsequently increases as the model complexity $D$ grows. The optimal estimation case is achieved when the relation $D \asymp N^{\frac{d_0}{4d_0+2\alpha}}$ holds. This insight provides crucial guidance for practical engineering design of Transformers.

---

[3]Operators involve exponential operations (e.g., $e^x$ or $e^{-x}$).

[4]See the error decomposition details in Appendix D.1.

**Remark 6.3** (Sub-optimality). Note that the minimax optimal convergence rate for deep ReLU networks in the regression setting is $N^{-\frac{2\alpha}{d_0+2\alpha}}$ (Schmidt-Hieber, 2020). Our derived rate in (16) is sub-optimal, primarily attributed to the employed loose VC-dimension upper bound (9). Specifically, the nearly-tight VC-dimension upper bound for fixed-width deep ReLU networks scales as $\tilde{\mathcal{O}}(D^2)$ (Bartlett et al., 2019) while ours scales as $\mathcal{O}(D^4)$. Narrowing this gap via aspects discussed in "Tightness Analysis" part after Theorem 5.3 will promisingly yield a tighter convergence rate.

**Remark 6.4** (Optimization Error). Bounding the optimization error $\tau$ in (16) is also necessary for a complete analysis of the excess risk, yet it remains highly challenging in theory at present. Importantly, our analysis holds independently of the optimization process, and we leave the analysis of optimization error $\tau$ for further work.

## 7. Conclusion

This paper establishes error upper and lower bounds for standard Transformers (with Softmax operators, ReLU activations and residual connections) in approximating the bounded Hölder class, thereby quantitatively revealing the fundamental expressive power of Transformers. For the upper bound $\mathcal{O}(\varepsilon^{-d_0/\alpha})$ on the block number, we explicitly construct three key modules of standard Transformers: the quantization module, contextual mapping and value mapping; For the lower bound, a new yet non-optimal conclusion reveals that at least $\Omega(\varepsilon^{-d_0/(4\alpha)})$ blocks are required to achieve $\varepsilon$ accuracy. The error gap of two bounds highlights substantial room for approach improvement and can inspire the subsequent studies. We then derive excess risk rates of Transformers in regression tasks involving the Hölder class.

Our analysis reveals several key theoretical challenges for further work: *i*) Simplifying constructions of the value mapping and utilizing higher smoothness of target functions for tighter upper bounds. *ii*) Tighter VC-dimension upper bounds to obtain (nearly) optimal approximation lower bounds. *iii*) Role of compositions of multiple self-attention layers in the approximation tasks.

## Acknowledgements

We would like to thank the anonymous reviewers and the Area Chair for their constructive comments and suggestions, which have helped improve this paper. This work is supported by the National Key Research and Development Program of China (Grant No. 2025YFA1017800), the National Natural Science Foundation of China (Grant Nos. 12125103, 12371424, 12371441, U24A2002, 12561160122, 12526216), the Natural Science Foundation of Hubei Province (2024AFE006) and the Fundamental Research Funds for the Central Universities.

## Impact Statement

This paper presents theoretical results regarding the approximation capabilities of standard Transformer architectures, contributing to the interpretability theory of Transformers. As a foundational work in statistical learning theory, its immediate societal consequences are primarily indirect. Understanding the theoretical model complexity requirements may help reduce the computational and environmental costs of training. And we acknowledge that our bounds provide theoretical limits and may differ from the empirical scale of current highly complex large language models.

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

## Appendix Index

## A. Additional Technical Notes

### A.1. The Summary of Notations

*Table 2.* The list of notations through the paper.

| Symbols | Descriptions |
|---|---|
| $\mathcal{H}^\alpha_{d_x,d_y}(\mathcal{X}, K)$ | The Hölder class with smoothness $\alpha$ in Definition 3.1 and $\mathcal{H}^\alpha_{d_0}(\mathcal{X}, K) \coloneqq \mathcal{H}^\alpha_{d_0,d_0}(\mathcal{X}, K)$. |
| $\bar{\mathcal{H}}^\alpha_{d,L}$ | Reshaped sequence form of $\mathcal{H}^\alpha_{d_0}(\mathcal{X}, K)$ defined in (6) with $d_0 = dL$. |
| $\mathcal{H}(\delta)$ | Intermediate piece-wise constant function class defined in (7). |
| $\mathcal{T}^D_R$ | The Transformer function class with $D$ blocks and shape layer $R$ in (4). |
| $\delta$ & $\delta^*$ | The cell width $\delta$ and the separation width $\delta^*$ in Definition 4.4. |
| $\mathcal{G}_{\delta,\delta^*}$ & $\mathcal{G}^p_{\delta,\delta^*}$ | Quantization grid in Definition 4.4 and its positional counterpart $\mathcal{G}^p_{\delta,\delta^*} = \mathcal{G}_{\delta,\delta^*} + E$ in Lemma 4.7. |
| $\mathcal{E}_{\text{app}}$ & $f_{\text{app}}$ | The approximation error $\mathcal{E}_{\text{app}}$ defined in (5) and the best approximator $f_{\text{app}}$ defined in (56). |
| $\mathcal{E}_{\text{sta}}$ & $\hat{f}$ | The statistical error $\mathcal{E}_{\text{sta}}$ defined in (14) and the best empirical estimator $\hat{f}$ in (10). |
| $\tau$ & $f_{s(\tau)}$ | The optimization error $\tau$ endowed by the solver $s(\tau)$ and the best numerical solution in (11). |
| $l$ & $\widehat{\mathcal{L}}$ & $\mathcal{L}$ | The square loss function $l(x, y) = (x - y)^2$, the empirical loss $\widehat{\mathcal{L}}$ in (10) and the population risk $\mathcal{L}$. |
| $\mu$ & $N$ | The covering radius $\mu$ in (15) and the sample size $N$. |
| $\lesssim$ & $\gtrsim$ | $A \lesssim B$ and $B \gtrsim A$ denote $A \leq cB$ for some absolute constant $c > 0$. |
| $\asymp$ | $A \asymp B$ means $c_1 A \leq B \leq c_2 A$ for some absolute constants $c_1, c_2 > 0$. |
| $\mathcal{O}(\cdot)$ & $\tilde{\mathcal{O}}(\cdot)$ | $\mathcal{O}(\cdot)$ is used to hide constant factors, while $\tilde{\mathcal{O}}(\cdot)$ ignores logarithmic factors. |
| $\mathbb{1}_d$ & $\mathbb{1}_{d\times L}$ | The $d$-dimensional vector and the $d \times L$ matrix with all one entries. |
| VCdim$(\mathcal{F})$ | VC-dimension of the function class $\mathcal{F}$, see Definition 5.1. |

### A.2. Technical Remarks

**Reshape and Flatten Layers.** In Definition 3.2 and (4), we introduced the reshape layer and its reverse. We now provide an example of such layers transforming representations between vector form and sequence form.

**Example** (Column Reshape). Suppose $d_0, d, L \in \mathbb{N}_+$ with $d_0 = dL$. Let $\{e_k\}^{d_0}_{k=1}$ be the standard basis vectors of $\mathbb{R}^{d_0}$ (where $e_k$ has a 1 at entry $k$ and 0 elsewhere). Let $\{E_{ij}\}^{i=d,j=L}_{i=1,j=1}$ be the canonical basis matrices of $\mathbb{R}^{d\times L}$ (where $E_{ij}$ has a 1 at entry $(i, j)$ and 0 elsewhere). The following constructed $R : \mathbb{R}^{d_0} \to \mathbb{R}^{d\times L}$ can be a reshape layer

$$R(e_k) = E_{i(k),j(k)}, \text{ where } i(k) = ((k - 1) \bmod d) + 1, \text{ and } j(k) = \lfloor \frac{k-1}{d} \rfloor + 1. \tag{17}$$

and its reverse $R^{-1}$ is defined as $R^{-1}(E_{ij}) = e_{(j-1)d+i}$. Therefore, for a general vector $x = \sum^{d_0}_{k=1} x_k e_k$, the output of reshape layer (17) is $R(x) = \sum^{d_0}_{k=1} x_k E_{i(k),j(k)}$. For a matrix $X \in \mathbb{R}^{d\times L}$, $R^{-1}(X) = \sum^d_{i=1}\sum^L_{j=1} X_{ij} \cdot e_{(j-1)d+i}$.

## B. Approximation Error Upper Bound

**Roadmap.** In this section, we derive the error upper bound for the proposed Transformer architecture on the Hölder class, followed by an extended discussion on the underlying theoretical assumptions.

- **Regularity Analysis** (Section B.1 & B.2): Establishes the regularity properties of the Hölder class, which facilitates the reduction to piecewise constant function approximation in the sequence form.

- **Module Construction** (Section B.3): Constructs the three core Transformer modules.

- **Main Result** (Section B.4): Combines the above results to derive the final approximation error upper bound and the corresponding corollary.

- **Extended Discussions** (Section B.5): Justifies the theoretical simplifications while exploring the implications for real-world models.

## B.1. Regularity of the Hölder Class

The Hölder class has the following smoothness property.

**Lemma B.1** (Smoothness Property of Hölder Class). *Let $d_x, d_y, K \in \mathbb{N}_+, \mathcal{X} \subseteq \mathbb{R}^{d_x}$ and $\alpha \in (0,1]$, for any Hölder function $f \in \mathcal{H}^{\alpha}_{d_x,d_y}(\mathcal{X}, K)$ and $x, y \in \mathcal{X}$, we have*

$$\|f(x) - f(y)\|_2 \le \sqrt{d_y} K \|x - y\|_2^{\alpha} := c_{d_y,K} \|x - y\|_2^{\alpha}. \tag{18}$$

*Proof:* Without loss of generality, we take the component $f_1$ of $f = (f_1, \ldots, f_{d_y})$ for instance. In the case of $r = 0$, we have $\alpha = \alpha_0$. Then since

$$|f_1(x) - f_1(y)| \le K \cdot \|x - y\|_2^{\alpha_0}, \text{ for any } x, y \in \mathcal{X},$$

we can get $\|f(x) - f(y)\|_2 \le \left( \sum_{i=1}^{d_y} |f_i(x) - f_i(y)|^2 \right)^{1/2} \le \sqrt{d_y} K \|x - y\|_2^{\alpha_0}$. $\qquad\square$

With equivalent transformations between the vector norm $\|\cdot\|_2$ and matrix norm $\|\cdot\|_F$, we obtain following corollary for the reshaped Hölder class $\bar{f} \in \bar{\mathcal{H}}^{\alpha}_{d,L}$ defined in (6) when $d_x = d_y = d_0$.

**Corollary B.2** (Regularity of Reshaped Hölder Class). *Suppose $d, L, d_0 = dL \in \mathbb{N}_+$ and $\alpha \in (0,1]$. For any two sequences $X, Y \in \bar{\mathcal{X}} = [0,1]^{d \times L}$ and any reshaped Hölder function $\bar{f} \in \bar{\mathcal{H}}^{\alpha}_{d,L}$ w.r.t. $f \in \mathcal{H}^{\alpha}_{d_0}$, we have $\|\bar{f}(X) - \bar{f}(Y)\|_2 \le c_{d_0,K} \|X - Y\|_F^{\alpha}$, where $c_{d_0,K} = \sqrt{d_0} K$.*

## B.2. Approximation Error Between Hölder Class and Piece-wise Constant Function Class

**Lemma 4.6** (Approximation Error between $\bar{\mathcal{H}}^{\alpha}_{d,L}$ and $\mathcal{H}(\delta)$; Restatement). *Let $\alpha \in (0,1], d_0 \in \mathbb{N}_+$ with $d_0 = dL > 2\alpha$. For any given $\varepsilon > 0$ and a target function $\bar{f} \in \bar{\mathcal{H}}^{\alpha}_{d,L}$, there exists an $f_\delta \in \mathcal{H}(\delta)$ with $\delta = \mathcal{O}(\varepsilon^{\frac{1}{\alpha}})$ and $\delta^* = \mathcal{O}(\varepsilon^{\frac{2}{d_0} + \frac{1}{\alpha}})$ such that $\|\bar{f} - f_\delta\|_2 \le \varepsilon$.*

***Proof of Lemma 4.6.*** Assume $d_x = d_y = d_0$. Let $\delta$ and $\delta^*$ be positive values to be determined later.

Suppose we have a grid and cube set:

$$\mathcal{G}_{\delta,\delta^*} = \{0, \delta + \delta^*, 2(\delta + \delta^*), \ldots, (M-1)(\delta + \delta^*)\}^{d \times L} \text{ and } \{\mathcal{S}_G \mid G \in \mathcal{G}_{\delta,\delta^*}\}.$$

We define the following piece-wise constant function for any $\bar{f} \in \bar{\mathcal{H}}^{\alpha}_{d,L}$ by

$$f_\delta(X) = \sum_{G \in \mathcal{G}_{\delta,\delta^*}} \bar{f}(G) \cdot \mathbb{1}\{X \in \mathcal{S}_G\}.$$

For any $X, Y \in [0,1]^{d \times L}$ and $\bar{f} = R \circ f \circ R^{-1} \in \bar{\mathcal{H}}^{\alpha}_{d,L}$ w.r.t. $f \in \mathcal{H}^{\alpha}_{d_0}$, we have

$$\|\bar{f}(X) - \bar{f}(Y)\|_2 = \|R \circ f \circ R^{-1}(X) - R \circ f \circ R^{-1}(Y)\|_2 = \|f \circ R^{-1}(X) - f \circ R^{-1}(Y)\|_2$$
$$\le c_{d_0,K} \|R^{-1}(X) - R^{-1}(Y)\|_2^{\alpha} = c_1 \|X - Y\|_F^{\alpha},$$

with $c_1 := c_{d_0,K} = \sqrt{d_0} K$ defined in (18).

For any $X \in \mathcal{S}_G$ and its corresponding grid point $G$, the relation $\|X - G\|_F \leq \sqrt{dL}\delta = \sqrt{d_0}\delta$ holds. We get

$$\left\|\bar{f}(X) - f_\delta(X)\right\|_F = \left\|\bar{f}(X) - \bar{f}(G)\right\|_F \leq c_1 \|X - G\|_F^\alpha \leq c_1 (\sqrt{d_0}\delta)^\alpha, \tag{19}$$

On the other hand, for any $X \in [0,1]^{d \times L} \setminus \bigcup_{G \in \mathcal{G}_{\delta,\delta^*}} \mathcal{S}_G$, we have

$$\left\|\bar{f}(X) - f_\delta(X)\right\|_F = \left\|\bar{f}(X)\right\|_F \leq \sqrt{dL}K = \sqrt{d_0}K. \tag{20}$$

Therefore, with $M = \lfloor \frac{1}{\delta + \delta^*} \rfloor$ and $\bigcup_G \mathcal{S}_G := \bigcup_{G \in \mathcal{G}_{\delta,\delta^*}} \mathcal{S}_G$,

$$\begin{aligned}
\left\|\bar{f} - f_\delta\right\|_2 &= \Big( \int_{[0,1]^{d \times L}} \left\|\bar{f}(X) - f_\delta(X)\right\|_F^2 \, \mathrm{d}X \Big)^{1/2} \\
&= \Big( \int_{\bigcup_G \mathcal{S}_G} \left\|\bar{f}(X) - f_\delta(C_G)\right\|_F^2 \, \mathrm{d}X \Big)^{1/2} + \Big( \int_{[0,1]^{d \times L} \setminus \bigcup_G \mathcal{S}_G} \left\|\bar{f}(X)\right\|_F^2 \, \mathrm{d}X \Big)^{1/2} \\
&\leq c_1 (\sqrt{d_0}\delta)^\alpha + (1 - M\delta)^{d_0/2} \sqrt{d_0}K.
\end{aligned} \tag{21}$$

For any accuracy $\varepsilon > 0$, we first let $c_1 (\sqrt{d_0}\delta)^\alpha < \varepsilon/2$, and it indicates

$$\delta < \frac{\varepsilon^{1/\alpha}}{(2c_1)^{1/\alpha} d_0^{1/2}} = d_0^{-\frac{\alpha+1}{2\alpha}} (2K)^{-\frac{1}{\alpha}} \varepsilon^{\frac{1}{\alpha}}.$$

Denote $c_2 = \left(\frac{\varepsilon}{2\sqrt{d_0}K}\right)^{2/d_0}$. If we set a smaller $\delta^*$ satisfying $\delta^* < \frac{(c_2 - \delta)\delta}{2} < \frac{(c_2 - \delta)\delta}{1 + \delta - c_2}$, we have

$$(1 - M\delta)^{d_0/2} \sqrt{d_0}K \leq \left(1 - \Big(\frac{1}{\delta + \delta^*} - 1\Big)\delta\right)^{d_0/2} \sqrt{d_0}K < \frac{\varepsilon}{2}. \tag{22}$$

Therefore, we derive the error $\left\|\bar{f} - f_\delta\right\|_2 < \varepsilon$. Equivalently, if we want the right hand of (21) to be smaller than any accuracy $\varepsilon > 0$, we should let

$$\begin{aligned}
\delta &< \varepsilon^{\frac{1}{\alpha}} d_0^{-\frac{\alpha+1}{2\alpha}} (2K)^{-\frac{1}{\alpha}} = \mathcal{O}(\varepsilon^{\frac{1}{\alpha}}), \\
\delta^* &< \frac{(c_2 - \delta)\delta}{2} = \frac{1}{2}\big(\varepsilon^{\frac{2}{d_0}} (2K)^{-\frac{2}{d_0}} d_0^{-\frac{1}{d_0}} - \delta\big)\delta = \mathcal{O}(\varepsilon^{\frac{2}{d_0} + \frac{1}{\alpha}}).
\end{aligned}$$

We clarify that the requirement of $c_2 > \delta$ should be satisfied to keep $\delta^*$ greater than zero. In a sense, this indicates $d_0 > 2\alpha$ and the target function inherently possesses higher input dimension with lower smoothness. □

## B.3. Approximation for Piece-wise Constant Function Class by Standard Transformers

Before starting the proof of components within the standard Transformers, we first demonstrate the novelty and uniqueness of our architecture with related works in this field. See Table 3.

### B.3.1. QUANTIZATION MODULE $f_{\mathcal{T}1}$

**Lemma 4.7** (Quantization Module $f_{\mathcal{T}1}$; Restatement). *Let $\delta, \delta^* \in (0,1)$, $M = \lfloor \frac{1}{\delta + \delta^*} \rfloor$ and $d_0 = dL$. By processing the inputs $X \in \bigcup_{G \in \mathcal{G}_{\delta,\delta^*}} \mathcal{S}_G$, there exists a quantization module $f_{\mathcal{T}1}$ composed of $d_0 M$ feed-forward layers and a positional encoding $E = \mathbb{1}_d \cdot [1, 2, \ldots, L]^\top$, constructing a quantization positional grid $\mathcal{G}_{\delta,\delta^*}^p = \mathcal{G}_{\delta,\delta^*} + E$.*

***Proof of Lemma 4.7.*** Since it will be required in the subsequent constructions of contextual mapping composed of a self-attention layer, we first introduce the following positional encoding $E \in \mathbb{R}^{d \times L}$,

$$\mathrm{PoE} : X \mapsto X + E, \quad \text{where} \quad E = \mathbb{1}_d \cdot [1, 2, \ldots, L]^\top. \tag{23}$$

Suppose $\delta > 0, \delta^* > 0$. For any $j \in [L]$, we create the grid $\mathcal{G}_{\delta,\delta^*}^p = \mathcal{G}_{\delta,\delta^*} + E$ by mapping all the elements in $[j, j+1]$ to $\{j, j + \delta + \delta^*, \ldots, j + (M-1)(\delta + \delta^*)\}$. Specifically, take $i = 1, 2, \ldots, d; j = 1, 2, \ldots, L$, and $k = 0, 1, \ldots, M-1$ respectively with $t = X - (j + k(\delta + \delta^*)) \cdot \mathbb{1}_{d \times L}$, and we design the following feed-forward layer,

$$\mathrm{FF}^{(i,j,k+1)}(X) = X + e_i \Big( -\sigma_R[t] + \frac{\delta + \delta^*}{\delta^*} \sigma_R[t - \delta \cdot \mathbb{1}_{d \times L}] - \frac{\delta}{\delta^*} \sigma_R\big[t - (\delta + \delta^*) \cdot \mathbb{1}_{d \times L}\big] \Big), \tag{24}$$

*Table 3.* Novelty and uniqueness in our constructions of Transformers.

| Part | Paper | Operator | Residual | Remark |
|---|---|---|---|---|
| Quantization Mapping | (Yun et al., 2020) | Designed discontinuous piece-wise linear | $\checkmark$ | Transformations between operators |
| | (Kajitsuka & Sato, 2024; Hu et al., 2025a;b; Jiao et al., 2025) | ReLU | $\times$ | Variant |
| | **Ours** | ReLU | $\checkmark$ | Standard structure |
| Contextual Mapping | (Yun et al., 2020) | Hardmax | $\checkmark$ | Transformations |
| | (Kajitsuka & Sato, 2024; Hu et al., 2025a;b; Jiao et al., 2025) | Softmax | $\checkmark$ | Single layer |
| | **Ours** | Softmax | $\checkmark$ | Single layer |
| Value Mapping | (Yun et al., 2020) | Designed discontinuous piece-wise linear | $\checkmark$ | Transformations between operators |
| | (Kajitsuka & Sato, 2024; Hu et al., 2025a;b; Jiao et al., 2025) | ReLU | $\times$ | Variant |
| | **Ours** | ReLU | $\checkmark$ | Standard structure |

where

$$-\sigma_R[t] + \frac{\delta + \delta^*}{\delta^*}\sigma_R[t - \delta \cdot \mathbb{1}_{d \times L}] - \frac{\delta}{\delta^*}\sigma_R\left[t - (\delta + \delta^*) \cdot \mathbb{1}_{d \times L}\right] = \begin{cases} 0, & t < 0 \text{ or } t \geq \delta + \delta^*, \\ -t, & 0 \leq t \leq \delta, \\ \frac{\delta}{\delta^*}t - \frac{\delta(\delta + \delta^*)}{\delta^*}, & \delta \leq t \leq \delta + \delta^*. \end{cases}$$

- Layer $\text{FF}^{(i,j,k+1)}$ maps the elements of $i$-row of $X$ into $j + k(\delta + \delta^*)$, if those elements are in the target cube $[j + k(\delta + \delta^*), j + k(\delta + \delta^*) + \delta]$.

- For the remaining elements in the separation gap cube $[j + k(\delta + \delta^*) + \delta, j + (k + 1)(\delta + \delta^*)]$, although they are mapped to random values in $[j + k(\delta + \delta^*), j + (k + 1)(\delta + \delta^*)]$, the measure of those points is small and won't cause much disturbance to the final conclusion of approximation upper bound, which will be checked carefully in the subsequent verification part.

Stack all above $dLM = d_0 M$ layers, and we construct

$$f_{\mathcal{T}1}(X) = \text{FF}^{(d,L,M)} \circ \cdots \circ \text{FF}^{(1,1,1)}(X + E). \tag{25}$$

The quantization module $f_{\mathcal{T}1}$ maps elements of $X$ in $\bigcup_G \mathcal{S}_G$ to the left-hand end points of their target cubes with a form of $j + k(\delta + \delta^*)$, as these elements must be in one of the target cubes. Therefore, we create the clean grid $\mathcal{G}^p_{\delta,\delta^*} = \mathcal{G}_{\delta,\delta^*} + E$.  $\square$

**Remark B.3.** Further dive into (25), we can just combine all ReLU units of the $dLM$ layers into a single feed-forward layer by boosting the layer width instead of increasing the depth, which makes room for a simple and flexible trade-off between the depth and width.

### B.3.2. CONTEXTUAL MAPPING $f_{\mathcal{T}2}$

Yun et al. (2020) pointed out that sequence models like Transformers should distinguish between different input sequences sharing same tokens, and then proposed an essential module called the contextual mapping which is composed of numerous self-attention layers with Hardmax operators. Furthermore, Kajitsuka & Sato (2024) simplified (Yun et al., 2020)'s methodology by designing a single self-attention layer contextual mapping based only on the Softmax operator. We utilize this essential contextual mapping in the approximation upper bound tasks.

**Lemma B.4** (Modified Version of (Kajitsuka & Sato, 2024, Lemma 2.2)). *Let $X^{(1)}, \ldots, X^{(N)} \in \mathbb{R}^{d \times L}$ be $N$ input sequences with no duplicate word token in each sequence, that is, $X^{(i)}_{:,k} \neq X^{(i)}_{:,l}$ for any $i \in [N]$ and $k, l \in [L]$ with $k \neq l$. Further assume $X^{(1)}, \ldots, X^{(N)}$ are token-wise $(r_{\min}, r_{\max}, \beta)$-separated. Then, there exists a Softmax self-attention layer which is an $(r, \gamma)$-contextual mapping for the input sequences $X^{(1)}, \ldots, X^{(N)} \in \mathbb{R}^{d \times L}$ with $r$ and $\gamma$ defined by*

$$r = r_{\max} + \frac{\beta}{4}, \text{ and } \gamma = \frac{r_{\min}(\beta \log L)^2}{4dr^2_{\max}(2\log L + 3)(|\mathcal{V}| + 1)^4} \exp\left(-(|\mathcal{V}| + 1)^4 \frac{2dr^2_{\max}(2\log L + 3)}{\beta r_{\min}}\right).$$

**Based on Lemma B.4**, we can prove the following lemma for our constructed contextual mapping $f_{\mathcal{T}2}$.

**Lemma 4.10** (Contextual Mapping $f_{\mathcal{T}2}$; Restatement). *All the $M^{d_0}$ grid points $G \in \mathcal{G}^p_{\delta,\delta^*}$ are $(\sqrt{d}, \sqrt{d}(L+1), \sqrt{d}(\delta+\delta^*))$-separated. A single self-attention layer equipped with one Softmax head, can be an $(r,\gamma)$-contextual mapping $f_{\mathcal{T}2} : \mathbb{R}^{d \times L} \to \mathbb{R}^{d \times L}$ for the input sequences $G$, satisfying*

1. *For any $i \in [M^{d_0}]$ and $k \in [L]$, the boundedness $\left\| f_{\mathcal{T}2}(G^{(i)})_{:,k} \right\|_2 \leq r$ holds.*

2. *For any $i,j \in [M^{d_0}]$ and $k,l \in [L]$ such that either $\mathcal{V}^{(i)} \neq \mathcal{V}^{(j)}$ or $G^{(i)}_{:,k} \neq G^{(j)}_{:,l}$, the separatedness $\left\| f_{\mathcal{T}2}(G^{(i)})_{:,k} - f_{\mathcal{T}2}(G^{(j)})_{:,l} \right\|_2 \geq \gamma$ holds.*

*with $r$ and $\gamma$ defined by $r = \sqrt{d}(L+1) + \frac{\sqrt{d}(\delta+\delta^*)}{4}$ and $\gamma = \tilde{\mathcal{O}}\left( \exp\left( -\frac{L^6 M^{4d_0}}{\delta+\delta^*} \right) \right)$.*

***Proof of Lemma 4.10.*** The positional encoding $E$ helps any grid point $G \in \mathcal{G}^p_{\delta,\delta^*}$ meet the no duplicate token condition. It's easy to check that $|\mathcal{G}^p_{\delta,\delta^*}| = M^{d_0}$ and $|\mathcal{V}_G| = LM^{d_0}$ with $M = \lfloor \frac{1}{\delta+\delta^*} \rfloor$. Furthermore, all grid points of $\mathcal{G}^p_{\delta,\delta^*}$ are $\left( \sqrt{d}, \sqrt{d}(L+1), \sqrt{d}(\delta+\delta^*) \right)$-separated, that is, for any $k,l \in [L], k \neq l$,

$$\|G_{:,k}\|_2 \geq \sqrt{d},$$
$$\|G_{:,k}\|_2 \leq \sqrt{d}(L+1-(\delta+\delta^*)) \leq \sqrt{d}(L+1),$$
$$\|G_{:,k} - G_{:,l}\|_2 \geq \sqrt{d}(\delta+\delta^*).$$

Directly apply Lemma B.4 to the grid set $\mathcal{G}^p_{\delta,\delta^*}$, and we can get the $(r,\gamma)$-contextual mapping $f_{\mathcal{T}2}$ composed of a single self-attention layer where

$$
\begin{aligned}
r &= \sqrt{d}(L+1) + \frac{\sqrt{d}(\delta+\delta^*)}{4}, \\
\gamma &= \frac{(\delta+\delta^*)^2(\log L)^2}{4\sqrt{d}(2\log L+3)(L+1)^2(LM^{d_0}+1)^4} \exp\left( -(LM^{d_0}+1)^4 \frac{2d(2\log L+3)(L+1)^2}{\delta+\delta^*} \right).
\end{aligned}
\tag{26}
$$

We have obtained the main results. $\qquad\square$

Now it is crucial to ensure the correctness of Lemma B.4. We restate several lemmas first.

**Lemma B.5** (Park et al., 2021, Lemma 13). *Let $N, d \in \mathbb{N}_+$. For any $\mathcal{X} = \{x^{(i)} \mid x^{(i)} \in \mathbb{R}^d, i = 1, 2, \ldots, N\}$, there exists a unit vector $u \in \mathbb{R}^d$ such that*

$$\frac{1}{N^2}\sqrt{\frac{8}{\pi d}} \|x - x'\|_2 \leq \left| u^\top(x - x') \right| \leq \|x - x'\|_2, \text{ for all } x, x' \in \mathcal{X}.$$

**Lemma B.6.** *Let $r_{\min}, r_{\max}, \beta > 0$. Given an $(r_{\min}, r_{\max}, \beta)$-separated vocabulary $\mathcal{V} \subset \mathbb{R}^d$ with $|\mathcal{V}| < +\infty$. For any $\delta_0 > 0$, there exists a unit vector $v \in \mathbb{R}^d$ such that for any vectors $u, u' \in \mathbb{R}^m$ with*

$$\left| u^\top u' \right| = (|\mathcal{V}|+1)^4 \frac{d\delta_0}{\beta r_{\min}},$$

*we have*

$$\left| (W_K v_a)^\top W_Q v_c - (W_K v_b)^\top W_Q v_c \right| \geq \delta_0, \text{ and } \frac{1}{(|\mathcal{V}|+1)^2\sqrt{d}} \|v_c\|_2 \leq |v^\top v_c| \leq \|v_c\|_2,$$

*for any $v_a, v_b, v_c \in \mathcal{V}$ with $v_a \neq v_b$, where $W_K = uv^\top \in \mathbb{R}^{m \times d}$ and $W_Q = u'v^\top \in \mathbb{R}^{m \times d}$.*

***Proof of Lemma B.6.*** By applying Lemma B.5 to $\mathcal{V} \cup \{0\}$, we know that there exists a unit vector $v \in \mathbb{R}^d$ such that for any $v_a, v_b \in \mathcal{V} \cup \{0\}$ with $v_a \neq v_b$, we have

$$\frac{1}{(|\mathcal{V}|+1)^2\sqrt{d}} \|v_a - v_b\|_2 \leq \frac{1}{(|\mathcal{V}|+1)^2}\sqrt{\frac{8}{\pi d}} \|v_a - v_b\|_2 \leq |v^\top(v_a - v_b)| \leq \|v_a - v_b\|_2. \tag{27}$$

In particular, (27) also means that for any $v_c \in \mathcal{V}$, we have by setting the other vector equal to zero

$$\frac{1}{(|\mathcal{V}|+1)^2\sqrt{d}}\|v_c\|_2 \le |v^\top v_c| \le \|v_c\|_2. \tag{28}$$

Furthermore, pick up two arbitrary vectors $u, u' \in \mathbb{R}^m$ satisfying

$$|u^\top u'| = (|\mathcal{V}|+1)^4 \frac{d\delta_0}{\beta r_{\min}}.$$

Setting $W_K = uv^\top \in \mathbb{R}^{m\times d}$ and $W_Q = u'v^\top \in \mathbb{R}^{m\times d}$, with (27) and (28), we have

$$\left|(W_K v_a)^\top W_Q v_c - (W_K v_b)^\top W_Q v_c\right|$$
$$= \left|(v_a - v_b)^\top W_K^\top W_Q v_c\right| = \left|(v_a - v_b)^\top v\right| \cdot |u^\top u'| \cdot |v^\top v_c|$$
$$\ge \frac{1}{(|\mathcal{V}|+1)^2\sqrt{d}}\|v_a - v_b\|_2 \cdot (|\mathcal{V}|+1)^4 \frac{d\delta_0}{\beta r_{\min}} \cdot \frac{1}{(|\mathcal{V}|+1)^2\sqrt{d}}\|v_c\|_2 \ge \delta_0,$$

where the last inequality comes from the $(r_{\min}, r_{\max}, \beta)$-separatedness of $\mathcal{V}$. $\qquad\square$

**Lemma B.7** (Kajitsuka & Sato, 2024, Lemma 1). *Let $a^{(1)}, \ldots, a^{(m)} \in \mathbb{R}^L$ be token-wise $(\kappa, \delta_0)$-separated vectors with no duplicate element in each vector and*

$$\delta_0 \ge 2\log L + 3.$$

*Then, the outputs of Boltzmann operator with the form $\mathrm{boltz}(a) = a^\top \sigma_S(a)$ are $(\kappa, \delta_1)$-separated, that is,*

$$\left\|\mathrm{boltz}(a^{(i)})\right\|_2 \le \kappa,$$
$$\left\|\mathrm{boltz}(a^{(i)}) - \mathrm{boltz}(a^{(j)})\right\|_2 > \delta_1 = (\log L)^2 e^{-2\kappa},$$

*hold for each $i, j \in [m]$ with $a^{(i)} \neq a^{(j)}$.*

Based on the core results of Lemma B.6 and B.7, we now can prove Lemma B.4.

***Proof of Lemma B.4.*** There are three main targets,

1. Show that the upper bound of the contextual mapping is $r = r_{\max} + \frac{\beta}{4}$.

2. Prove that the constructed self-attention head maps the different tokens to unique ids.

3. Demonstrate that the self-attention layer distinguishes between the duplicate input tokens within different contexts.

For the first target, since the self-attention layer with one Softmax-head $H$ is represented as

$$\mathrm{Attn}(X) = X + H = X + W_O W_V X \sigma_S\left[(W_K X)^\top W_Q X\right], \tag{29}$$

we can construct a head satisfying $\|H\|_2 \le \beta/4$ such that

$$\left\|\mathrm{Attn}(X^{(i)})_{:,k}\right\|_2 \le \left\|X^{(i)}_{:,k}\right\|_2 + \left\|H^{(i)}_{:,k}\right\|_2 \le r_{\max} + \frac{\beta}{4} := r, \quad \text{for all } i \in [N], k \in [L]. \tag{30}$$

It is obvious that we can also have the lower bound

$$\left\|\mathrm{Attn}(X^{(i)})_{:,k}\right\|_2 \ge \left\|X^{(i)}_{:,k}\right\|_2 - \left\|H^{(i)}_{:,k}\right\|_2 \ge r_{\min} - \frac{\beta}{4} := r_2, \quad \text{for all } i \in [N], k \in [L]. \tag{31}$$

For the second target, based on Lemma B.6, we set $W_K = uv^\top \in \mathbb{R}^{m\times d}$ and $W_Q = u'v^\top \in \mathbb{R}^{m\times d}$ with $u, u' \in \mathbb{R}^m$ and $v \in \mathbb{R}^d$. Let $\delta_0 = 2\log L + 3$ and fix these $u, u'$ by

$$|u^\top u'| = (|\mathcal{V}|+1)^4 \frac{d\delta_0}{\beta r_{\min}}, \tag{32}$$

so we have for any $v_a, v_b, v_c \in \mathcal{V}$ with $v_a \neq v_b$,

$$\left| (W_K v_a)^\top W_Q v_c - (W_K v_b)^\top W_Q v_c \right| \geq \delta_0 \quad \text{and} \quad \frac{1}{(|\mathcal{V}|+1)^2 \sqrt{d}} \|v_c\|_2 \leq |v^\top v_c| \leq \|v_c\|_2. \tag{33}$$

Furthermore, let the other two weight matrices $W_V \in \mathbb{R}^{m \times d}$ and $W_O \in \mathbb{R}^{d \times m}$ satisfy $W_V = u'' v^\top$ for any nonzero vector $u'' \in \mathbb{R}^m$ and $\|W_O u''\|_2 = \frac{\beta}{4 r_{\max}}$. As a result, the self-attention head $H$ can be bounded, since for any $k \in [L]$

$$
\begin{aligned}
\left\| H_{:,k}^{(i)} \right\|_2 &= \left\| W_O W_V X^{(i)} \sigma_S \left[ \left( W_K X^{(i)} \right)^\top \left( W_Q X_{:,k}^{(i)} \right) \right] \right\|_2 = \left\| \sum_{k'=1}^n \sigma_S[\cdot]_{k'} W_O W_V X_{:,k'}^{(i)} \right\|_2 \\
&\leq \sum_{k'=1}^n \sigma_S[\cdot]_{k'} \left\| W_O W_V X_{:,k'}^{(i)} \right\|_2 \leq \max_{k' \in [L]} \left\| W_O W_V X_{:,k'}^{(i)} \right\|_2 \\
&= \max_{k' \in [L]} \left\| W_O u'' v^\top X_{:,k'}^{(i)} \right\|_2 = \max_{k' \in [L]} \left| v^\top X_{:,k'}^{(i)} \right| \cdot \|W_O u''\|_2 \\
&\leq \max_{k' \in [L]} \left\| X_{:,k'}^{(i)} \right\|_2 \cdot \frac{\beta}{4 r_{\max}} \leq \frac{\beta}{4},
\end{aligned}
\tag{34}
$$

where $\sigma_S[\cdot]_{k'}$ represents the $k'$ element of $\sigma_S[(W_K X^{(i)})^\top (W_Q X_{:,k}^{(i)})]$.

With (34) and the $\beta$-separatedness of $X$, it is easy to show that

$$
\begin{aligned}
\left\| \text{Attn}(X^{(i)})_{:,k} - \text{Attn}(X^{(j)})_{:,l} \right\|_2 &= \left\| X_{:,k}^{(i)} - X_{:,l}^{(j)} + H_{:,k}^{(i)} - H_{:,l}^{(j)} \right\|_2 \\
&\geq \left\| X_{:,k}^{(i)} - X_{:,l}^{(j)} \right\|_2 - \left\| H_{:,k}^{(i)} \right\|_2 - \left\| H_{:,l}^{(j)} \right\|_2 \\
&\geq \beta - \frac{\beta}{4} - \frac{\beta}{4} = \frac{\beta}{2},
\end{aligned}
$$

for any $i, j \in [N]$ and $k, l \in [L]$ such that $X_{:,k}^{(i)} \neq X_{:,l}^{(j)}$. Till now we have already shown that this constructed self-attention layer can distinguish different tokens.

For the last and most essential target that the designed self-attention layer $\text{Attn}(X)$ can distinguish $X_{:,k}^{(i)} = X_{:,l}^{(j)}$ with $\mathcal{V}^{(i)} \neq \mathcal{V}^{(j)}$, we first define $a^{(i)}$ and $a^{(j)}$ as

$$a^{(i)} = \left( W_K X^{(i)} \right)^\top W_Q X_{:,k}^{(i)} \in \mathbb{R}^L \quad \text{and} \quad a^{(j)} = \left( W_K X^{(j)} \right)^\top W_Q X_{:,l}^{(j)} \in \mathbb{R}^L,$$

with the weight matrices $W_O, W_V, W_K, W_Q$ defined in the former targets.

From (32)-(33), we have that $a^{(i)}$ and $a^{(j)}$ are token-wise $(\kappa, \delta_0)$-separated where $\kappa$ is computed by for any $k' \in [L]$,

$$
\begin{aligned}
\left| a_{k'}^{(i)} \right| &= \left| \left( W_K X_{:,k'}^{(i)} \right)^\top W_Q X_{:,k}^{(i)} \right| = \left| \left( X_{:,k'}^{(i)} \right)^\top v \right| \cdot |u^\top u'| \cdot \left| v^\top X_{:,k}^{(i)} \right| \\
&\leq (|\mathcal{V}|+1)^4 \frac{d \delta_0 r_{\max}^2}{\beta r_{\min}} := \kappa.
\end{aligned}
$$

Since $\mathcal{V}^{(i)} \neq \mathcal{V}^{(j)}$ and there is no duplicate token in $X^{(i)}$ and $X^{(j)}$ respectively, we obtain from Lemma B.7 that

$$\left| \text{boltz}(a^{(i)}) - \text{boltz}(a^{(j)}) \right| = \left| (a^{(i)})^\top \sigma_S[a^{(i)}] - (a^{(j)})^\top \sigma_S[a^{(j)}] \right| > \delta_1 = (\log L)^2 e^{-2\kappa}. \tag{35}$$

For the assumed $X_{:,k}^{(i)} = X_{:,l}^{(j)}$, we have

$$
\begin{aligned}
&\left| (a^{(i)})^\top \sigma_S[a^{(i)}] - (a^{(j)})^\top \sigma_S[a^{(j)}] \right| \\
&= \left| \left( W_Q X_{:,k}^{(i)} \right)^\top W_K \left( X^{(i)} \sigma_S[a^{(i)}] - X^{(j)} \sigma_S[a^{(j)}] \right) \right| \\
&= \left| \left( X_{:,k}^{(i)} \right)^\top v u'^\top u v^\top \left( X^{(i)} \sigma_S[a^{(i)}] - X^{(j)} \sigma_S[a^{(j)}] \right) \right| \\
&= \left| \left( X_{:,k}^{(i)} \right)^\top v \right| \cdot \left| u'^\top u \right| \cdot \left| v^\top X^{(i)} \sigma_S[a^{(i)}] - v^\top X^{(j)} \sigma_S[a^{(j)}] \right| \\
&\leq (|\mathcal{V}| + 1)^4 \frac{d \delta_0 r_{\max}}{\beta r_{\min}} \cdot \left| v^\top X^{(i)} \sigma_S[a^{(i)}] - v^\top X^{(j)} \sigma_S[a^{(j)}] \right|,
\end{aligned}
\tag{36}
$$

where the last inequality follows from (32)-(33). Therefore, we have when $X_{:,k}^{(i)} = X_{:,l}^{(j)}$,

$$
\begin{aligned}
\left\| \text{Attn}(X^{(i)})_{:,k} - \text{Attn}(X^{(j)})_{:,l} \right\|_2 &= \left\| H_{:,k}^{(i)} - H_{:,l}^{(j)} \right\|_2 \\
&= \left\| W_O W_V X^{(i)} \sigma_S[a^{(i)}] - W_O W_V X^{(j)} \sigma_S[a^{(j)}] \right\|_2 \\
&= \| W_O u'' \|_2 \cdot \left| v^\top X^{(i)} \sigma_S[a^{(i)}] - v^\top X^{(j)} \sigma_S[a^{(j)}] \right| \\
&\geq \frac{\beta^2 r_{\min} \delta_1}{4(|\mathcal{V}| + 1)^4 d \delta_0 r_{\max}^2} := \gamma,
\end{aligned}
$$

where the last inequality gains from (35)-(36), $\delta_0 = 2 \log L + 3$ and $\delta_1 = (\log L)^2 e^{-2\kappa}$ with

$$
\kappa = (|\mathcal{V}| + 1)^4 \frac{d \delta_0 r_{\max}^2}{\beta r_{\min}}.
$$

Until now we have proved that the constructed single self-attention layer (29) is an $(r, \gamma)$-contextual mapping, where $W_V = u'' v^\top, W_K = uv^\top, W_Q = u'v^\top$ and $\| W_O u'' \|_2 = \frac{\beta}{4r_{\max}}$. $\qquad \square$

### B.3.3. VALUE MAPPING $f_{\mathcal{T}3}$

**Lemma 4.11** (Value Mapping $f_{\mathcal{T}3}$; Restatement). *Suppose $Y_G$ is the output matrix of $G$. Let $\delta, \delta^* \in (0, 1), M = \lfloor \frac{1}{\delta + \delta^*} \rfloor, d, L \in \mathbb{N}_+$ and $d_0 = dL$, there exists a value mapping $f_{\mathcal{T}3} : \mathbb{R}^{d \times L} \to \mathbb{R}^{d \times L}$ composed of $2LM^{d_0} + 1$ feed-forward layers such that $f_{\mathcal{T}3} \circ f_{\mathcal{T}2}(G) = Y_G, \forall G \in \mathcal{G}_{\delta, \delta^*}^p$.*

***Proof of Lemma 4.11.*** For the sake of convenience, we denote

$$
\begin{aligned}
r &= \sqrt{d}(L + 1) + \frac{\sqrt{d}(\delta + \delta^*)}{4}, \text{ and } r_2 = \sqrt{d} - \frac{\sqrt{d}(\delta + \delta^*)}{4}, \\
\gamma_1 &= \frac{\delta^2 (\log L)^2}{4\sqrt{d}(2 \log L + 3)(L + 1)^2 (LM^{d_0} + 1)^4} \exp\left( -(LM^{d_0} + 1)^4 \frac{2d(2 \log L + 3)(L + 1)^2}{\delta + \delta^*} \right), \\
\gamma_2 &= \frac{(\delta^*)^2 (\log L)^2}{4\sqrt{d}(2 \log L + 3)(L + 1)^2 (LM^{d_0} + 1)^4} \exp\left( -(LM^{d_0} + 1)^4 \frac{2d(2 \log L + 3)(L + 1)^2}{\delta + \delta^*} \right).
\end{aligned}
$$

Based on (26), for any two different grid points $G^{(i)}, G^{(j)}, i \neq j$, we have

$$
\left\| f_{\mathcal{T}2}(G^{(i)})_{:,k} - f_{\mathcal{T}2}(G^{(j)})_{:,l} \right\|_2 \geq \gamma > \gamma_1 + \gamma_2, \text{ for all } k \neq l, k \in [L], l \in [L].
\tag{37}
$$

Recall that $M = \lfloor \frac{1}{\delta + \delta^*} \rfloor$ is the number of grid points per dimension. We denote the total grid points as $M_{\text{all}} = M^{dL} = M^{d_0}$, arrange all the grid points in a certain order, and assign the serial number $i \in [M_{\text{all}}]$ to each of the point and its output, i.e., $G^{(i)}$ and $Y_G^{(i)}$.

In this proof, we are considering the current input $\bar{X} \in \{f_{\mathcal{T}2} \circ f_{\mathcal{T}1}(X) \mid X \in \bigcup_{G \in \mathcal{G}^p_{\delta,\delta^*}} \mathcal{S}_G\}$. Now take $i = 1, 2, \ldots, M_{\text{all}}$ and $k = 1, 2, \ldots, L$ respectively, and we design a subunit composed of two following feed-forward layers,

$$\text{FF}^{(i,k,1)}(\bar{X}) = \bar{X} + \sigma_{\zeta 1}\left[\bar{X} - f_{\mathcal{T}2}(G^{(i)})_{:,k} \cdot \mathbb{1}_L^\top\right] := Z, \tag{38}$$

where

$$\sigma_{\zeta 1}[t] = \frac{2dr}{\gamma_2}\left(\sigma_R\left[t + \frac{\gamma_1 + \gamma_2}{2\sqrt{d}}\right] - \sigma_R\left[t + \frac{\gamma_1}{2\sqrt{d}}\right] - \sigma_R\left[t - \frac{\gamma_1}{2\sqrt{d}}\right] + \sigma_R\left[t - \frac{\gamma_1 + \gamma_2}{2\sqrt{d}}\right]\right)$$

$$= \begin{cases} 0, & t < -\dfrac{\gamma_1 + \gamma_2}{2\sqrt{d}} \text{ or } t > \dfrac{\gamma_1 + \gamma_2}{2\sqrt{d}}, \\ \dfrac{2dr}{\gamma_2}t + \dfrac{\sqrt{d}r(\gamma_1 + \gamma_2)}{\gamma_2}, & -\dfrac{\gamma_1 + \gamma_2}{2\sqrt{d}} \le t \le -\dfrac{\gamma_1}{2\sqrt{d}}, \\ \sqrt{d}r, & -\dfrac{\gamma_1}{2\sqrt{d}} \le t \le \dfrac{\gamma_1}{2\sqrt{d}}, \\ -\dfrac{2dr}{\gamma_2}t + \dfrac{\sqrt{d}r(\gamma_1 + \gamma_2)}{\gamma_2}, & \dfrac{\gamma_1}{2\sqrt{d}} \le t \le \dfrac{\gamma_1 + \gamma_2}{2\sqrt{d}}, \end{cases}$$

and

$$\text{FF}^{(i,k,2)}(Z) = Z + \left(Y^{(i)}_{G:,k} + (r + K) \cdot \mathbb{1}_d - f_{\mathcal{T}2}(G^{(i)})_{:,k}\right)\sigma_{\zeta 2}\left[\mathbb{1}_d^\top \cdot Z - d\sqrt{d}r \cdot \mathbb{1}_L^\top\right] - \sigma_{\zeta 3}\left[Z - \sqrt{d}r\mathbb{1}_d \cdot \mathbb{1}_L^\top\right], \tag{39}$$

where

$$\sigma_{\zeta 2}[t] = \frac{1}{\gamma_2}\sigma_R[t] - \frac{1}{\gamma_2}\sigma_R[t - \gamma_2] = \begin{cases} 0, & t \le 0, \\ \dfrac{1}{\gamma_2}t, & 0 \le t \le \gamma_2, \\ 1, & t \ge \gamma_2, \end{cases}$$

$$\sigma_{\zeta 3}[t] = \frac{\sqrt{d}r}{\gamma_2}\sigma_R[t + \gamma_2] - \frac{\sqrt{d}r}{\gamma_2}\sigma_R[t] = \begin{cases} 0, & t \le -\gamma_2, \\ \dfrac{\sqrt{d}r}{\gamma_2}t + \sqrt{d}r, & -\gamma_2 \le t \le 0, \\ \sqrt{d}r, & t \ge 0. \end{cases}$$

• Layer $\text{FF}^{(i,k,1)}(\bar{X})$ aims to endow the label $\sqrt{d}r$ to those elements of $\bar{X}$ in $[-\gamma_1/(2\sqrt{d}), \gamma_1/(2\sqrt{d})]$ and to output results with a form of "$\bar{X}$ + label". Specifically, suppose we have the target column $\bar{X}_{:,p}$ close to $f_{\mathcal{T},c2}(G^{(i)})_{:,k}$ under the criterion $\left\|\bar{X}_{:,p} - f_{\mathcal{T},c2}(G^{(i)})_{:,k}\right\|_\infty \le \gamma_1/(2\sqrt{d})$, every element of $\bar{X}_{:,p}$ will be added with the label $\sqrt{d}r$. Based on (37), this criterion takes effects as

$$\left\|\bar{X}_{:,p} - f_{\mathcal{T}2}(G^{(j)})_{:,l}\right\|_\infty \ge \left\|f_{\mathcal{T}2}(G^{(i)})_{:,k} - f_{\mathcal{T}2}(G^{(j)})_{:,l}\right\|_\infty - \left\|\bar{X}_{:,p} - f_{\mathcal{T}2}(G^{(i)})_{:,k}\right\|_\infty$$

$$\ge \frac{1}{\sqrt{d}}\left\|f_{\mathcal{T}2}(G^{(i)})_{:,k} - f_{\mathcal{T}2}(G^{(j)})_{:,l}\right\|_2 - \left\|\bar{X}_{:,p} - f_{\mathcal{T}2}(G^{(i)})_{:,k}\right\|_\infty \tag{40}$$

$$\ge \frac{\gamma}{\sqrt{d}} - \frac{\gamma_1}{2\sqrt{d}} > \frac{\gamma_1 + 2\gamma_2}{2\sqrt{d}} > \frac{\gamma_2}{2\sqrt{d}}.$$

(40) also indicates that $\sigma_{\zeta 1}$ won't endow labels to the non-target columns because of the smooth gap $\gamma_2/(2\sqrt{d})$. After layer $\text{FF}^{(i,k,1)}(\bar{X})$, only the target column is added with the complete label vector $\sqrt{d}r \cdot \mathbb{1}_d$, while the labels of other columns are incomplete.

• As for layer $\text{FF}^{(i,k,2)}(Z)$, the part "$\mathbb{1}_d^\top \cdot Z - d\sqrt{d}r \cdot \mathbb{1}_L^\top$" first calculates the column-wise sum of elements in $Z$ and then subtracts $d\sqrt{d}r$ element-wisely. Results of all the non-target columns $\text{FF}^{(i,k,1)}(\bar{X}_{:,l}) = Z_{:,l}$ with $l \ne p$ will not be greater than 0, since they have at most $d - 1$ labels and

$$\max_{l \in [L], l \ne p} \sum_{j=1}^{d} Z_{j,l} - d\sqrt{d}r = \max_{l \in [L], l \ne p} \sum_{j=1}^{d} \bar{X}_{j,l} + (d-1)\sqrt{d}r - d\sqrt{d}r$$

$$\le \max_{l \in [L]} \left\|\bar{X}_{:,l}\right\|_1 - \sqrt{d}r \le \sqrt{d}\max_{l \in [L]} \left\|\bar{X}_{:,l}\right\|_2 - \sqrt{d}r \le 0.$$

Furthermore, $\sigma_{\zeta 2}$ maps the target column to 1 successfully, since for the target column, based on (31) we have

$$\left\|\bar{X}_{:,p}\right\|_1 \geq \left\|\bar{X}_{:,p}\right\|_2 \geq r_2 > \gamma_2.$$

The part $\sigma_{\zeta 3}$ just eliminates the temporary labels remaining in other non-target columns. Remind that we suppose the target column $\bar{X}_{:,p}$ is close to $f_{\mathcal{T},c2}(G^{(i)})_{:,k}$. Thus for this current input $\bar{X} \in \{f_{\mathcal{T}2} \circ f_{\mathcal{T}1}(X) \mid X \in \bigcup_{G \in \mathcal{G}_{\delta,\delta^*}^p} \mathcal{S}_G\}$, we have $\mathrm{FF}^{(i,k,2)} \circ \mathrm{FF}^{(i,k,1)}(\bar{X}_{:,p}) = Y_{G:,k}^{(i)} + (r + K) \cdot \mathbb{1}_d$ and other columns remain unchanged. Result of this subunit will not involve in the calculations of other subunits due to the existence of $r + K$.

• We have at most $LM_{\mathrm{all}}$ such subunits with anchor vectors $f_{\mathcal{T}2}(G^{(i)})_{:,k}$, for all $i \in [M_{\mathrm{all}}], k \in [L]$. Stacking these $2LM_{\mathrm{all}} = 2LM^{d_0}$ feed-forward layers, any contextual grid column $f_{\mathcal{T}2}(G^{(i)})_{:,k}$ is mapped to the target value vector $Y_{G:,k}^{(i)} + (r + K) \cdot \mathbb{1}_d$. We only require an extra feed-forward layer $\mathrm{FF}^{(\mathrm{extra})}$ to subtract the redundant $(r + K) \cdot \mathbb{1}_d$ vector in each column. Therefore, we construct the value mapping composed of $2LM^{d_0} + 1$ feed-forward layers

$$f_{\mathcal{T}3} = \mathrm{FF}^{(\mathrm{extra})} \circ \mathrm{FF}^{(M^{d_0},L,2)} \circ \mathrm{FF}^{(M^{d_0},L,1)} \circ \cdots \circ \mathrm{FF}^{(1,1,2)} \circ \mathrm{FF}^{(1,1,1)},$$

such that $f_{\mathcal{T}3} \circ f_{\mathcal{T}2}(G) = Y_G, \forall G \in \mathcal{G}_{\delta,\delta^*}^p$. Till now, we have already proven Lemma 4.11. $\qquad\square$

**Remark B.8** (Extensions). Actually, we can discuss a more general conclusion for the value mapping $f_{\mathcal{T}3}$ applied on

$$\bar{X} \in \{f_{\mathcal{T}2} \circ f_{\mathcal{T}1}(X) \mid X \in \bar{\mathcal{X}}^p = [0,1]^{d \times L} + E, E \text{ is defined in (23)}\}.$$

Lemma 4.11 corresponds to the case when $X \in \bigcup_{G \in \mathcal{G}_{\delta,\delta^*}^p} \mathcal{S}_G$. For completeness, now we discuss the case when $X \in \bar{\mathcal{X}}^p \setminus \bigcup_{G \in \mathcal{G}_{\delta,\delta^*}^p} \mathcal{S}_G$. If the column $\bar{X}_{:,p}$ is close to $f_{\mathcal{T},c2}(G^{(i)})_{:,k}$, we will still obtain the calculation results as the following

$$\mathrm{FF}^{(i,k,2)} \circ \mathrm{FF}^{(i,k,1)}(\bar{X}_{:,p}) = Y_{G:,k}^{(i)} + (r + K) \cdot \mathbb{1}_d + \bar{X}_{:,p} - f_{\mathcal{T}2}(G^{(i)})_{:,k} \approx Y_{G:,k}^{(i)} + (r + K) \cdot \mathbb{1}_d,$$

and $f_{\mathcal{T}3}(\bar{X}_{:,p}) \approx Y_{G:,k}^{(i)}$. On the other hand, the set of non-target $\bar{X}$ whose columns are not close to any columns of the anchor values will remain unchanged. This corresponding part of the value mapping $f_{\mathcal{T}3}$ can be controlled since the measure of complementary set is bounded, which will be important and useful in the subsequent verification part.

## B.4. Verification for the Approximation Error Upper Bound

**Theorem 4.1** (Upper Bound of Approximation Error; Restatement). *Let $\alpha \in (0,1], d_0 \in \mathbb{N}_+, d_0 > 2\alpha$, and $K \in \mathbb{N}_+$. For any target function $f \in \mathcal{H}_{d_0}^\alpha([0,1]^{d_0}, K)$ and any error criterion $\varepsilon > 0$, there exists a Transformer function $g \in \mathcal{T}_R^D$ composed of at most $\mathcal{O}(\varepsilon^{-\frac{d_0}{\alpha}})$ blocks such that $\|f - g\|_2 < \varepsilon$.*

***Proof of Theorem 4.1.*** The Transformer network $f_{\mathcal{T}}(X) = f_{\mathcal{T}3} \circ f_{\mathcal{T}2} \circ f_{\mathcal{T}1}(X + E)$ is composed of $d_0 M + 2LM^{d_0} + 1$ feed-forward layers and one self-attention layer, i.e., $d_0 M + 2LM^{d_0} + 2 = \mathcal{O}(\varepsilon^{-\frac{d_0}{\alpha}})$ blocks, since

$$M = \lfloor \frac{1}{\delta + \delta^*} \rfloor \leq \frac{1}{\delta + \delta^*} < \frac{1}{\delta} \lesssim \varepsilon^{-\frac{1}{\alpha}}, \text{ where } \delta = \mathcal{O}(\varepsilon^{\frac{1}{\alpha}}). \tag{41}$$

Now we first verify the upper bound of approximation error between the piece-wise constant function class $\mathcal{H}(\delta)$ and the Transformer network class $\mathcal{T}^D$.

Based on Lemmas from 4.7 to 4.11, for any $f_\delta \in \mathcal{H}(\delta)$, there exists an $f_{\mathcal{T}} \in \mathcal{T}^D$ such that

$$f_{\mathcal{T}}(X) = f_\delta(X), \text{ for } X \in \bigcup_{G \in \mathcal{G}_{\delta,\delta^*}} \mathcal{S}_G.$$

For $X \in [0,1]^{d \times L} \setminus \bigcup_{G \in \mathcal{G}_{\delta,\delta^*}} \mathcal{S}_G$, every element of $f_{\mathcal{T}1}(X)$ is in $[1, L+1]$ based on (23) and (24). Furthermore, based on (34) we have

$$\begin{aligned}
\left\|\bar{X}\right\|_2 &:= \|f_{\mathcal{T}2}(f_{\mathcal{T}1}(X))\|_2 \\
&\leq \|f_{\mathcal{T}1}(X)\|_2 + \left\|W_O W_V f_{\mathcal{T}1}(X)\sigma_S\left[(W_K f_{\mathcal{T}1}(X))^\top W_Q f_{\mathcal{T}1}(X)\right]\right\|_2 \\
&\leq \|f_{\mathcal{T}1}(X)\|_2 + \sqrt{L} \max_{k' \in [L]} \|f_{\mathcal{T}1}(X)_{:,k'}\|_2 \cdot \frac{\sqrt{d}(\delta + \delta^*)}{4\sqrt{d}(L+1)} \\
&\leq \sqrt{d_0}(L+1) + \frac{\sqrt{d_0}(\delta + \delta^*)}{4}.
\end{aligned}$$

Then for the last value mapping $f_{\mathcal{T}3}$,

$$
\begin{aligned}
\|f_{\mathcal{T}}(X)\|_2 &= \|f_{\mathcal{T}3}(\bar{X})\|_2 \\
&\leq \|\bar{X}\|_2 + \max_G \left\|Y_G + (r+K)\cdot \mathbb{1}_{d\times L} - f_{\mathcal{T}2}(G)\right\|_2 \\
&\leq \|\bar{X}\|_2 + \left\|(r+K)\cdot \mathbb{1}_{d\times L}\right\|_2 + \max_G\{\|Y_G\|_2 + \|f_{\mathcal{T}2}(G)\|_2\} \\
&\leq \sqrt{d_0}(L+1) + \frac{\sqrt{d_0}(\delta+\delta^*)}{4} + \sqrt{d_0}(r+K) + \sqrt{d_0}K + \sqrt{L}r \\
&= (\sqrt{d_0}(L+1) + \frac{\sqrt{d_0}(\delta+\delta^*)}{4})\cdot(\sqrt{d}+2) + 2\sqrt{d_0}K \\
&\leq 2\sqrt{d_0}(K + \frac{1}{2}(\sqrt{d}L + 2L + \frac{5}{4}\sqrt{d} + \frac{5}{2})) \leq 2\sqrt{d_0}(4d_0+K),
\end{aligned}
$$

where the first inequality comes from that the label $\sqrt{d}r$ will be eliminated in every subunit once being gotten. And the final result is with $r = \sqrt{d}(L+1) + \frac{\sqrt{d}(\delta+\delta^*)}{4}$ and $\delta+\delta^* < 1$.

In a nut, the approximation error for $f_\delta \in \mathcal{H}(\delta)$ by Transformer $f_{\mathcal{T}} \in \mathcal{T}^D$ is

$$
\begin{aligned}
&\|f_\delta - f_{\mathcal{T}}\|_2 \\
&= \left(\int_{\bigcup_G \mathcal{S}_G} \|f_\delta(X) - f_{\mathcal{T}}(X)\|_F^2\, dX + \int_{[0,1]^{d\times L}\setminus\bigcup_G \mathcal{S}_G} \|f_\delta(X) - f_{\mathcal{T}}(X)\|_F^2\, dX\right)^{1/2} \\
&= \left(\int_{\bigcup_G \mathcal{S}_G} \|Y_G - f_{\mathcal{T}3}\circ f_{\mathcal{T}2}(G)\|_F^2\, dX + \int_{[0,1]^{d\times L}\setminus\bigcup_G \mathcal{S}_G} \|f_\delta(X) - f_{\mathcal{T}}(X)\|_F^2\, dX\right)^{1/2} \\
&= \left(\int_{[0,1]^{d\times L}\setminus\bigcup_G \mathcal{S}_G} \|f_{\mathcal{T}}(X)\|_F^2\, dX\right)^{1/2} \leq (1-M\delta)^{d_0/2}\cdot 2\sqrt{d_0}(4d_0+K).
\end{aligned}
\tag{42}
$$

Furthermore, denote $c_3 = (\frac{\varepsilon}{4\sqrt{d_0}(4d_0+K)})^{2/d_0}$ and let $\delta^* < \frac{(c_3-\delta)\delta}{2} = \mathcal{O}(\varepsilon^{\frac{2}{d_0}+\frac{1}{\alpha}})$ and $\delta = \mathcal{O}(\varepsilon^{\frac{1}{\alpha}})$. Similar to (22), we have

$$
\|f_\delta - f_{\mathcal{T}}\|_2 < \frac{\varepsilon}{2}.
$$

Directly with Lemma 4.6 and above results, we get the approximation upper bound for any $\bar{f} \in \bar{\mathcal{H}}_{d,L}^\alpha$ by Transformer $f_{\mathcal{T}}$ under accuracy $\varepsilon > 0$,

$$
\|\bar{f} - f_{\mathcal{T}}\|_2 \leq \|\bar{f} - f_\delta\|_2 + \|f_\delta - f_{\mathcal{T}}\|_2 < \varepsilon.
\tag{43}
$$

Now we have proved the Proposition 4.12.

Further with the reshape layer $R$ and its inverse $R^{-1}$, we can get the approximation error upper bound for any $f \in \mathcal{H}_{d_0}^\alpha$ by a Transformer function $g \in \mathcal{T}_R^D$,

$$
\begin{aligned}
\|f - g\|_2 &= \|R^{-1}\circ \bar{f}\circ R - R^{-1}\circ f_{\mathcal{T}}\circ R\|_2 \\
&= \|\bar{f} - f_{\mathcal{T}}\|_2 \leq \|\bar{f} - f_\delta\|_2 + \|f_\delta - f_{\mathcal{T}}\|_2 < \varepsilon.
\end{aligned}
\tag{44}
$$

And (44) means $\mathcal{E}_{\text{app}} := \sup_{f\in\mathcal{H}_{d_0}^\alpha} \inf_{g\in\mathcal{T}_R^D} \|f - g\|_2 < \varepsilon$ with $D \lesssim \varepsilon^{-\frac{d_0}{\alpha}}$ for any $\varepsilon \in (0,1)$. Now we prove Theorem 4.1. $\qquad\square$

**Corollary 4.3** (Depth-width Trade-off; Restatement)**.** *For any $\varepsilon > 0$ and $d_0 = dL$, a Transformer architecture $\mathcal{T}_R^D$ with $D = \mathcal{O}(\varepsilon^{-d_0/\alpha})$ blocks and fixed width $W = 4d$, which is capable of approximating any function $f \in \mathcal{H}_{d_0}^\alpha([0,1]^{d_0}, K)$ with smooth index $\alpha \in (0,1]$ under error $\varepsilon$, can be realized by a Transformer $\mathcal{T}_R^{D',W'}$ comprising variable blocks $D'$, width $W'$, and two additional linear embedding layers, satisfying the condition $D'\cdot W' = \mathcal{O}(\varepsilon^{-d_0/\alpha})$, where $D' \geq 6$ and $W' \geq 4d$.*

***Proof of Corollary 4.3.*** Our construction in Theorem 4.1 utilizes three key components: the quantization module, contextual mapping, and value mapping. From the proofs of Lemma 4.7 (quantization module) and Lemma 4.11 (value mapping), the

sequential construction requires $\mathcal{O}(d_0 M)$ and $\mathcal{O}(LM^{d_0})$ total neurons, respectively. Crucially, subunits composed of these neurons operate on disjoint indices and do not interfere with one another (See discussions below (24) & (39)).

Leveraging the additive nature of neurons in the quantization module and value mapping, we can transform sequential depth into parallel width. Specifically, the weight matrices of the original modules are embedded into a block-diagonal structure within wider layers. For instance, to compress $n$ sequential FNN layers, parameterized by weight matrices $W_1^{(i)}, W_2^{(i)}$ and bias terms $b_1^{(i)}, b_2^{(i)}, i \in [n]$, into a single wider layer, we can construct as following,

$$
\tilde{W}_j = \begin{bmatrix} W_j^{(1)} & 0 & \cdots & 0 \\ 0 & W_j^{(2)} & \cdots & 0 \\ \vdots & \vdots & \ddots & \vdots \\ 0 & 0 & \cdots & W_j^{(n)} \end{bmatrix} \text{ and } \tilde{b}_j = \begin{bmatrix} b_j^{(1)} \\ b_j^{(2)} \\ \vdots \\ b_j^{(n)} \end{bmatrix} \text{ for } j \in \{1, 2\}.
$$

To satisfy the Transformer's requirement for equal input and output dimensions, we employ an input replication scheme: expanding the input $X \in \mathbb{R}^{d \times L}$ to $\tilde{X} = [X; X; \ldots; X] \in \mathbb{R}^{nd \times L}$. This allows the wide layer to process $n$ independent functional channels simultaneously, i.e.,

$$
\tilde{FF}(\tilde{X}) = \tilde{X} + \tilde{W}_2 \sigma_R \left[ \tilde{W}_1 \tilde{X} + \tilde{b}_1 \mathbb{1}_L^\top \right] + \tilde{b}_2 \mathbb{1}_L^\top.
$$

By redistributing the total neuron budget, we obtain a parallel quantization module and value mapping satisfying $D_1' \cdot W_1' = \mathcal{O}(d_0 M)$ and $D_2' \cdot W_2' = \mathcal{O}(LM^{d_0})$, respectively. To aggregate the independent channels after the parallel quantization, we introduce an additional linear embedding layer of $(I_d, I_d, \ldots, I_d) \in \mathbb{R}^{d \times nd}$ ($I_d$ denotes the $d \times d$ identity matrix), followed by the subtraction of the residual term $(n-1)X$. The intermediate result is then processed by the contextual mapping. Similarly, for the parallel value mapping, we expand the input and aggregate the outputs. Through these structural alignments, the constructed parallel Transformer exactly recovers the output of the initial sequential model.

Accounting the two additional embedding layers and setting $M \lesssim \varepsilon^{-1/\alpha}$ yields the final trade-off relation $D' \cdot W' = \mathcal{O}(\varepsilon^{-d_0/\alpha})$, where the effective depth is $D' = D_1' + D_2' + 3$ and width $W' = \max(W_1', W_2', d)$. Since $D_1' \geq 1, D_2' \geq 2$ and $W = 4d$, the constraints $D' \geq 6$ and $W' \geq W$ ensure the minimal architectural budget. $\qquad\square$

### B.5. Further Discussions: Assumptions, Implications, and Practical Architectures

To rigorously establish the approximation bounds, we introduce structural simplifications that isolate the network's fundamental mechanisms. Below, we bridge the gap between theory and practice by clarifying the connections between these mathematical assumptions and the empirical design of modern Transformers:

- **Activation Functions.** We adopt ReLU activation to establish a rigorous theoretical baseline, supported by the fact that any continuous piecewise linear activation function can be exactly realized by ReLU combinations (Yarotsky, 2017). Besides, real-world models frequently employ smoother variants, such as GELU or SwiGLU, to facilitate better gradient flow and enhance local expressivity. Our ReLU-based analysis defines the fundamental limits of approximation power, theoretically justifying the use of these advanced and smoother activations.

- **Normalization.** Normalization techniques, such as LayerNorm, are omitted to cleanly isolate and analyze the core approximation power of the self-attention and feed-forward parts. While practically essential for stabilizing deep network optimization, normalization does not expand the theoretical approximation capacity. Mathematically, its magnitude adjustments can be effectively absorbed by appropriately scaling the adjacent weight parameters.

- **Attention Depth.** Our analysis focuses on a single attention layer, demonstrating it as the minimal structural requirement to achieve contextual mapping. Since stacking attention layers strictly preserves or improves the overall approximation power, our single-layer framework provides a baseline guarantee for the expressive capacity of more complex architectures. Deriving bounds for deep Transformers introduces significant theoretical complexity due to their highly compositional nature, making multiple attention layer extensions a natural direction for future work.

## C. Approximation Error Lower Bound

**Roadmap.** This section establishes the error lower bound by analyzing the VC-dimension upper bound of Transformers.

*Table 4.* Compulsory numbers of operations and parameters of designed Transformer.

| Transformer part | Component | Number of operations | Number of para. |
|---|---|---|---|
| Quantization module $f_{\mathcal{T}1}$ | Positional encoding $E$ | $dL$ | $M+4$ |
| | $dLM$ layers of FF$_1$ in (46) | $13(dL)^2M+5LM+3$ | |
| Contextual mapping $f_{\mathcal{T}2}$ | One self-attention layer | $dL(8d+4L-4)+2L^2-L$ | $4d^2$ |
| Value mapping $f_{\mathcal{T}3}$ | $LM^{dL}$ layers of FF$_2$ in (47) | $(13dL+4d)LM^{dL}+7$ | $2dLM^{dL}+6$ |
| | $LM^{dL}$ layers of FF$_3$ in (48) | $(12dL+5L+2d)LM^{dL}+7$ | |
| | One layer of FF$_4$ | $dL$ | |

- **Bounding the VC-dimension** (Section C.1): quantifies the total number of operations and parameters of proposed Transformers to derive the VC-dimension upper bound.

- **Main Result** (Section C.2): Derives the final approximation error lower bound based on above results.

### C.1. Total Numbers of Operations and Parameters

A detail-oriented structure of Transformer is used for obtaining the approximation upper bound in this paper. Therefore, we can calculate the number of operations and parameters part by part. Derived results will be utilized in the subsequent applications of general regression tasks in the proof of Theorem 5.3 (Section C.2) and Lemma D.5 (Section D.2). Set $\delta > 0, \delta^* > 0, M = \lfloor \frac{1}{\delta+\delta^*} \rfloor$, and $d, L, d_0 = dL \in \mathbb{N}_+$. See the final results in Table 4.

**As a result**, we have the following total $t$ operations and $\omega$ parameters of $f_{\mathcal{T}}$

$$t = (25dL + 6d + 5L)LM^{dL} + 13d^2L^2M + 8d^2L + 4dL^2 + 2L^2 + 5LM - 2dL - L + 17,$$
$$\omega = 2dLM^{dL} + 4d^2 + M + 10.$$

Further from Section B.4, the Transformer function class $\mathcal{T}_R^D$ which approximates Hölder class $\mathcal{H}_{d_0}^{\alpha}(\mathcal{X}, K)$ under the accuracy $\varepsilon > 0$ has $dLM + 2LM^{dL} + 1$ blocks. Therefore, it is rational to assume $t, \omega$ and $D$ share the same order, that is, $t \sim \omega \sim D$. Immediately, we obtain

$$\text{VCdim}(\mathcal{T}_R^D) \leq t^2\omega(\omega + 19\log_2(9\omega)) \lesssim D^4. \tag{45}$$

#### C.1.1. DETAILS OF COMPUTATION

The following are details of computations of total number of parameters and operations, which serves as reference if needed.

**Quantization Module.** $f_{\mathcal{T}1}$ is composed of $d_0M$ feed-forward layers and the positional encoding $E$. The positional encoding $E$ defined in (23) involves $dL$ addition operations and the subsequent feed-forward layer is briefly like

$$\text{FF}_1^{(i,j,k+1)}(X) = X + e_i\left(-\sigma_R[t] + \frac{\delta+\delta^*}{\delta^*}\sigma_R[t - \delta \cdot \mathbb{1}_{d\times L}] - \frac{\delta}{\delta^*}\sigma_R\left[t - (\delta+\delta^*) \cdot \mathbb{1}_{d\times L}\right]\right), \tag{46}$$

where $i \in [d], j \in [L], k+1 \in [M]$, and $t = X - (j + k(\delta + \delta^*)) \cdot \mathbb{1}_{d\times L}$. This module has $M + 4$ total parameters "$-1, 0, \ldots, M, \delta, \delta^*$".

In the case of number of operations, there are 3 ground operations for preparing "$\delta + \delta^*, \frac{\delta}{\delta^*}, \frac{\delta+\delta^*}{\delta^*}$" and $5LM$ operations for getting "$j + k(\delta + \delta^*) - \delta$" and "$j + (k-1)(\delta + \delta^*)$". Besides, one such feed-forward layer needs at most $10dL$ element-wise arithmetic operations and $3dL$ jump comparisons from $\sigma_R$. Consequently, there are $dLM \cdot 13dL + 5LM + 3 = 13(dL)^2M + 5LM + 3$ operations in total.

**Contextual Mapping.** $f_{\mathcal{T}2}$ takes the form

$$\text{Attn}(X) = X + W_O W_V X \sigma_S\left[(W_K X)^\top W_Q X\right],$$

where $W_O, W_K, W_Q, W_V \in \mathbb{R}^{m \times d}$. Here, we set $m = d$ for simplicity since the head size $m$ can be arbitrary. There are $4d^2$ parameters.

Now we consider the number of operations in this single-layer contextual mapping. Suppose we have a single vector $x \in \mathbb{R}^L$, the Softmax operator $\sigma_S$ needs $L$ exponential $\alpha \mapsto e^\alpha$ operations and $2L - 1$ arithmetic operations on $x$. Further for $L$ columns input, $\sigma_S$ needs $L(3L - 1)$ operations. As a result, $f_{\mathcal{T}2}$ has total $dL(8d + 4L - 4) + 2L^2 - L$ operations, including $L^2$ exponential, $4d^2L + 2dL^2 + L^2$ of "$\times$" and "/", and $4d^2L + 2dL^2 - 4dL - L$ of "$+$" and "$-$".

**Value Mapping.** $f_{\mathcal{T}3}$ consists of $LM^{dL}$ subunits and an extra feed-forward layer. For any $i \in [M^{dL}]$ and $k \in [L]$, every subunit has $\mathrm{FF}_2^{(i,k,1)}$ and $\mathrm{FF}_3^{(i,k,2)}$ defined by

$$\mathrm{FF}_2^{(i,k,1)}(\bar{X}) = \bar{X} + \sigma_{\zeta 1}[\bar{X} - f_{\mathcal{T}2}(G^{(i)})_{:,k} \cdot \mathbb{1}_L^\top] := Z, \tag{47}$$

where

$$\sigma_{\zeta 1}[t] = \frac{2dr}{\gamma_2}\left(\sigma_R\left[t + \frac{\gamma_1 + \gamma_2}{2\sqrt{d}}\right] - \sigma_R\left[t + \frac{\gamma_1}{2\sqrt{d}}\right] - \sigma_R\left[t - \frac{\gamma_1}{2\sqrt{d}}\right] + \sigma_R\left[t - \frac{\gamma_1 + \gamma_2}{2\sqrt{d}}\right]\right),$$

and

$$\mathrm{FF}_3^{(i,k,2)}(Z) = Z + (Y_{G:,k}^{(i)} + (r + K) \cdot \mathbb{1}_d - f_{\mathcal{T}2}(G^{(i)})_{:,k})\sigma_{\zeta 2}\left[\mathbb{1}_d^\top \cdot Z - d\sqrt{d}r \cdot \mathbb{1}_L^\top\right] - \sigma_{\zeta 3}\left[Z - \sqrt{d}r\mathbb{1}_d \cdot \mathbb{1}_L^\top\right], \tag{48}$$

where

$$\sigma_{\zeta 2}[t] = \frac{1}{\gamma_2}\sigma_R[t] - \frac{1}{\gamma_2}\sigma_R[t - \gamma_2], \text{ and } \sigma_{\zeta 3}[t] = \frac{\sqrt{d}r}{\gamma_2}\sigma_R[t + \gamma_2] - \frac{\sqrt{d}r}{\gamma_2}\sigma_R[t].$$

In a nut, there are 6 additional global parameters "$\gamma_1, \gamma_2, r, K, d, \sqrt{d}$" and $2d$ temporary parameters "$Y_{G:,k}^{(i)}, f_{\mathcal{T}2}(G^{(i)})_{:,k}$" in every two layers, resulting in $2dLM^{dL} + 6$ total parameters.

Moreover, these two layers need 14 operations to prepare "$\frac{2dr}{\gamma_2}, 2\sqrt{d}, \frac{\gamma_1}{2\sqrt{d}}, \frac{\gamma_1 + \gamma_2}{2\sqrt{d}}, \sqrt{d}r, d\sqrt{d}r, r + K, \frac{1}{\gamma_2}, \frac{\sqrt{d}r}{\gamma_2}, d\sqrt{d}r + \gamma_2, \gamma_2 - \sqrt{d}r$". Layer $\mathrm{FF}_2^{(i,k,1)}(\bar{X})$ needs $9dL + 4d$ arithmetic operations (where $4d$ steps are for linear operations between $f_{\mathcal{T}2}(G^{(i)})_{:,k}$ and 4 constants in $\sigma_R[\cdot]$) and $4dL$ jumps of $\sigma_R$. On the other hand, $\mathrm{FF}_3^{(i,k,2)}(Z)$ needs $10dL + 3L + 2d$ arithmetic operations and $2dL + 2L$ jumps. Due to $LM^{dL}$ sets, we get $(13dL + 4d)LM^{dL} + (12dL + 5L + 2d)LM^{dL}$ total operations. The last layer $\mathrm{FF}_4$ of subtracting the constant $r + K$ element-wisely, needs $dL$ operations.

## C.2. Lower Bound of the Approximation Error

**Theorem 5.3** (Lower Bound of Approximation Error; Restatement). *For any $\varepsilon > 0$, a Transformer architecture $\mathcal{T}_R^D$ with $D$ blocks, which is capable of approximating any function $f \in \mathcal{H}_{d_0}^\alpha(\mathcal{X}, K)$ with smooth index $\alpha \in (0, 1]$ under error $\varepsilon$, must possess $t$ operations, $\omega$ parameters and $D$ blocks satisfying*

$$D^4 \gtrsim t^2\omega(\omega + 19\log_2(9\omega)) \geq \mathrm{VCdim}(\mathcal{T}_R^D) \gtrsim \varepsilon^{-\frac{d_0}{\alpha}}.$$

*Equivalently, the block number $D \gtrsim \varepsilon^{-\frac{d_0}{4\alpha}}$.*

***Proof of Theorem 5.3***. Our proof strategy adapts the technique developed by (Shen et al., 2022), extending it to handle the analysis of Transformers. As the VC-dimension roots in the scalar function class, we degenerate the output dimension of the Hölder class into one, that is, $f \in \mathcal{H}_{d_0}^\alpha : \mathbb{R}^{d_0} \to \mathbb{R}$ with $d_x = d_0$ and $d_y = 1$. This degeneration won't influence the main structure of Transformer network $\mathcal{T}^D$ since we apply the proper $\mathcal{E}_{\mathrm{in}} = \mathbb{1}_{d_0} : \mathbb{R}^{d_0} \to \mathbb{R}^{d_0}$ and $\mathcal{E}_{\mathrm{out}} = e_1 : \mathbb{R}^{d_0} \to \mathbb{R}$. Let

$$\mathcal{T}_R^D = \left\{g = \mathcal{E}_{\mathrm{out}} \circ R^{-1} \circ f_{\mathcal{T}} \circ R \circ \mathcal{E}_{\mathrm{in}} : \mathbb{R}^{d_0} \to \mathbb{R} \mid f_{\mathcal{T}} \in \mathcal{T}^D, d_0 = dL\right\}.$$

Then Theorem 5.2 demonstrates that

$$\mathrm{VCdim}(\mathcal{T}_R^D) \leq t^2\omega(\omega + 19\log_2(9\omega)), \tag{49}$$

where $t$ and $\omega$ are the total numbers of operations and parameters of $\mathcal{M}$ respectively.

Given a positive integer $M = \lfloor\frac{1}{\delta + \delta^*}\rfloor$ to be defined later, we divide $[0, 1]^{d_0}$ into $M^{d_0}$ non-overlapping sub-cubes $\{S_\theta\}_\theta$, and set

$$S_\theta = \left\{x = [x_1, x_2, \ldots, x_{d_0}]^\top \mid x_j \in [\theta_j(\delta + \delta^*), \theta_j(\delta + \delta^*) + \delta], j \in [d_0]\right\},$$

for any index vector $\theta = [\theta_1, \theta_2, \ldots, \theta_{d_0}]^\top = \{0, 1, \ldots, M-1\}^{d_0}$.

Let $S := S(\hat{x}, \mu)$ denote the closed cube with center $\hat{x} \in \mathbb{R}^{d_0}$ and side-length $l > 0$. Define $\zeta_S$ on $[0,1]^d$ corresponding to $S$, satisfying i) $\zeta_S(\hat{x}) = (l/2)^\alpha/2$; ii) $\zeta_S(x) = 0$ for any $x \notin S\backslash\partial S$, where $\partial S$ is the boundary of $S$; iii) $\zeta_S$ is linear on the line that connects $\hat{x}$ and $x$ for any $x \in \partial S$.

Define
$$\Phi := \left\{ \phi : \phi \text{ is a map from } \{0, (\delta+\delta^*), 2(\delta+\delta^*), \ldots, (M-1)(\delta+\delta^*)\}^{d_0} \text{ to } \{-1, 1\} \right\}.$$
For each $\phi \in \Phi$, we define
$$f_\phi(x) := \sum_{\theta \in \{0,1,\ldots,M-1\}^{d_0}} \phi(\theta)\zeta_{S_\theta}(x).$$

We can directly obtain that $\{f_\phi : \phi \in \Phi\}$ shatters $M^{d_0} = \mathcal{O}(\varepsilon^{-d_0/\alpha})$ points based on the definition.

Furthermore, we demonstrate that $\{f_\phi : \phi \in \Phi\} \subseteq \mathcal{H}_{d_0}^\alpha(\mathcal{X}, 1)$ with $\alpha \in (0, 1]$. For any defined $f_\phi$ and $x, y \in [0,1]^{d_0}$, if $x, y$ are in a same cube of center $\hat{x}$ with $\|x-y\| \le l$,

$$|f_\phi(x) - f_\phi(y)| = |\zeta_{S_\theta}(x) - \zeta_{S_\theta}(y)| = \left| \frac{l^{\alpha-1}}{2^\alpha}\|x - \hat{x}\| - \frac{l^{\alpha-1}}{2^\alpha}\|y - \hat{x}\| \right|$$

$$\le \frac{l^{\alpha-1}}{2^\alpha}\|x-y\|^{1-\alpha+\alpha} \le \frac{1}{2^\alpha}\|x-y\|^\alpha \le \|x-y\|^\alpha.$$

If $x, y$ are in different cubes with $\|x-y\| \ge \frac{l}{2}$,

$$|f_\phi(x) - f_\phi(y)| \le |\zeta_{S_\theta}(x)| + |\zeta_{S_{\theta'}}(y)| \le \frac{1}{2}\cdot\left(\frac{l}{2}\right)^\alpha + \frac{1}{2}\cdot\left(\frac{l}{2}\right)^\alpha \le \|x-y\|^\alpha.$$

For any accuracy $\varepsilon > 0$ and each $\phi \in \Phi$, based on Theorem 4.1, there is a Transformer function $g_{\phi,t_\mathcal{T},\omega_\mathcal{T}} \in \mathcal{T}_R^D$ with $t_\mathcal{T}$ operations and $\omega_\mathcal{T}$ properly adjusted parameters, which can approximate the $f_\phi \in \mathcal{H}_{d_0}^\alpha$ such that

$$\|g_{\phi,t_\mathcal{T},\omega_\mathcal{T}} - f_\phi\|_2 \le \varepsilon.$$

As $g_{\phi,t_\mathcal{T},\omega_\mathcal{T}}$ and $f_\phi$ are both bounded and defined on the compact domain $[0,1]^{d_0}$, the metrics $L^2$ and $L^\infty$ are equal. Consequently, we can also have

$$\|g_{\phi,t_\mathcal{T},\omega_\mathcal{T}} - f_\phi\|_\infty \le \varepsilon + \varepsilon/81.$$

Therefore, we obtain for each $\phi \in \Phi$,

$$|g_{\phi,t_\mathcal{T},\omega_\mathcal{T}}(x) - f_\phi(x)| \le \frac{82}{81}\varepsilon, \quad \text{for any } x \in [0,1]^{d_0}\backslash\mathcal{X}_{\phi,\text{zero}},$$

where $\mathcal{X}_{\phi,\text{zero}}$ is a null set under the Lebesgue measure. And $\mathcal{X}_{\text{zero}} := \bigcup_\phi \mathcal{X}_{\phi,\text{zero}}$ is also a null set. So

$$|g_{\phi,t_\mathcal{T},\omega_\mathcal{T}}(x) - f_\phi(x)| \le \frac{82}{81}\varepsilon, \quad \text{for any } \phi \in \Phi \text{ and } x \in [0,1]^{d_0}\backslash\mathcal{X}_{\text{zero}}. \tag{50}$$

Now set $\frac{1}{\delta+\delta^*} = \left(\frac{9\varepsilon}{2}\right)^{-\frac{1}{\alpha}}$, and the side-length of $S_\theta$ is $\frac{1}{M} = \frac{1}{\lfloor(9\varepsilon/2)^{-1/\alpha}\rfloor}$. We have for each $\theta \in \{0, 1, \ldots, M-1\}^{d_0}$ and any $x \in \frac{1}{10}S_\theta$ (the closed cube shares $\frac{1}{10}$ of side-length and same center $x_{S_\theta}$ of $S_\theta$),

$$|f_\phi(x)| = |\zeta_{S_\theta}(x)| \ge \frac{9}{10}|\zeta_{S_\theta}(x_{S_\theta})| = \frac{9}{10}\cdot\frac{1}{2}\frac{1}{(2M)^\alpha} \ge \frac{81}{80}\varepsilon. \tag{51}$$

Since $(\frac{1}{10}S_\theta)\backslash\mathcal{X}_{\text{zero}}$ is not empty, for each $\theta \in \{0, 1, \ldots, M-1\}^{d_0}$ and each $\phi \in \Phi$, with (50) and (51) there exists a point $x_\theta \in (\frac{1}{10}S_\theta)\backslash\mathcal{X}_{\text{zero}}$ such that

$$|f_\phi(x_\theta)| \ge \frac{81}{80}\varepsilon > \frac{82}{81}\varepsilon \ge |g_{\phi,t_\mathcal{T},\omega_\mathcal{T}}(x_\theta) - f_\phi(x_\theta)|. \tag{52}$$

(52) means $g_{\phi,t_\mathcal{T},\omega_\mathcal{T}}(x_\theta)$ and $f_\phi(x_\theta)$ have the same sign for each $\theta \in \{0, 1, \ldots, M-1\}^{d_0}$ and $\phi \in \Phi$. This demonstrates $\{g_{\phi,t_\mathcal{T},\omega_\mathcal{T}}(x_\theta) \mid \phi \in \Phi\}$ shatters $\{x_\theta \mid \theta \in \{0, 1, \ldots, M-1\}^{d_0}\}$. Consequently,

$$\text{VCdim}(\mathcal{T}_R^D) \ge \text{VCdim}(\{g_{\phi,t_\mathcal{T},\omega_\mathcal{T}}(x_\theta) \mid \phi \in \Phi\}) \ge M^{d_0} = \lfloor(9\varepsilon/2)^{-1/\alpha}\rfloor^{d_0} \gtrsim \varepsilon^{-d_0/\alpha}.$$

With (49), there exists

$$t^2\omega(\omega + 19\log_2(9\omega)) \ge \text{VCdim}(\mathcal{T}_R^D) \gtrsim \varepsilon^{-d_0/\alpha}. \tag{53}$$

Further with (45), the main result of Theorem 5.3 is derived. $\qquad\square$

# D. Details of Applications in the Regression Task

**Roadmap.** This section details the application of our theory to the general regression task.

- **Background** (Section D.1): Provides the basic knowledge of the error decomposition of excess risk.

- **Statistical Analysis** (Section D.2): Establishes the upper bound of the statistical error.

- **Main Result** (Section D.3): Combines above results and derives the final convergence rate of the excess risk.

## D.1. Error Decomposition of Excess Risk

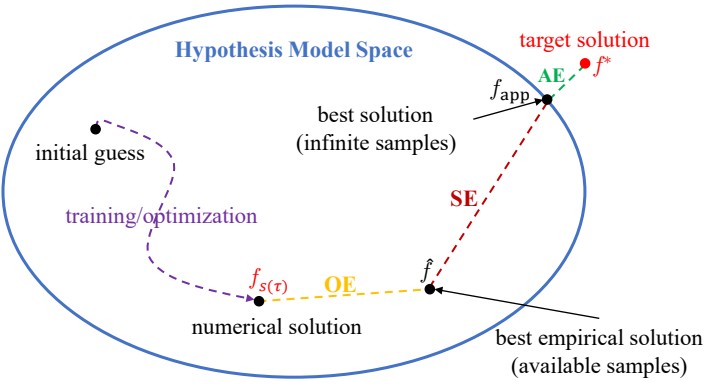

AE = approximation error;  SE = statistical error;  OE = optimization error

*Figure 2.* Diagram: Error decomposition of the excess risk.

Via the statistical learning, the total error between any obtainable estimator $f_{s(\tau)}$ and the target function $f^*$ can be measured by the expectation of the excess risk,. The error can be decompose into three parts.

**Lemma D.1** (Error Decomposition of Excess Risk). *Given an i.i.d sample set $\mathbb{D} = \{z^{(i)} = (x^{(i)}, y^{(i)})\}_{i=1}^N$, the target function space $\mathcal{H}_{d_0,1}^\alpha$ and the model space $\mathcal{T}_R^D$, for the actual estimator $f_{s(\tau)} \in \mathcal{F}$ obtained via minimizing the empirical loss $\widehat{\mathcal{L}}(f) = \frac{1}{N} \sum_{i=1}^N \left( f(x^{(i)} - y^{(i)}) \right)^2$ by a solver $s(\tau)$, the excess risk has the following decomposition*

$$\mathbb{E}_{\mathbb{D}} \left[ \mathcal{E}(f_{s(\tau)}) \right] = \mathbb{E}_{\mathbb{D}} \left[ \mathcal{L}(f_{s(\tau)}) - \mathcal{L}(f^*) \right] \leq \mathcal{E}_{\mathrm{sta}} + 2\mathcal{E}_{\mathrm{app}}^2 + 2\tau, \tag{54}$$

*where $\mathcal{L}$ is the population risk, $\mathcal{E}_{\mathrm{app}} = \sup_{p \in \mathcal{H}_{d_0,1}^\alpha} \inf_{f \in \mathcal{T}_R^D} \|f - p\|_2$ is the approximation error, $\mathcal{E}_{\mathrm{sta}} := \mathbb{E}_{\mathbb{D}}[\mathcal{L}(f_{s(\tau)}) - 2\widehat{\mathcal{L}}(f_{s(\tau)}) + \mathcal{L}(f^*)]$ is the statistical error, and the optimization error $\tau$.*

**Remark D.2** (Roles of Each Error). Before the proof of Lemma D.1, we discuss the intuitions behind this decomposition. For an intuitive grasp of the error decomposition, consider the illustration in Figure 2.

1. $\mathcal{E}_{\mathrm{app}}$: The approximation error serves as a measure of the bias induced by the choice of the model architecture. Even with infinite training samples (i.e., the oracle setting), the model cannot further reduce the total error below the approximation error, thus characterizing the fundamental limit of the model's expressivity.

2. $\mathcal{E}_{\mathrm{sta}}$: The statistical error serves as a measure of the variance induced by the finiteness of training data in the real-word settings. This error can be mitigated by increasing sample size, thus characterizing the generalization capability of the model.

3. $\tau$:  The optimization error serves as a measure of the discrepancy between the numerical solution returned by the training algorithm (or solver) and the best empirical solution. It characterizes the computational difficulty of finding the best model parameter sets within the hypothesis space.

***Proof of Lemma D.1.*** Consider the expectation of the excess risk

$$
\begin{aligned}
\mathbb{E}_{\mathbb{D}}\left[\mathcal{E}(f_{s(\tau)})\right] &= \mathbb{E}_{\mathbb{D}}\left[\mathcal{L}(f_{s(\tau)}) - \mathcal{L}(f^*)\right] \\
&= \mathbb{E}_{\mathbb{D}}\left[\mathcal{L}(f_{s(\tau)}) - 2\widehat{\mathcal{L}}(f_{s(\tau)}) + \mathcal{L}(f^*)\right] + 2\mathbb{E}_{\mathbb{D}}\left[\widehat{\mathcal{L}}(f_{s(\tau)}) - \mathcal{L}(f^*)\right] \\
&= \mathcal{E}_{\mathrm{sta}} + 2\mathbb{E}_{\mathbb{D}}\left[\widehat{\mathcal{L}}(f_{s(\tau)}) - \mathcal{L}(f^*)\right].
\end{aligned}
\tag{55}
$$

For the second part, define the best approximator

$$
f_{\mathrm{app}} \in \arg\min_{f \in \mathcal{T}_R^D} \mathcal{L}(f),
\tag{56}
$$

and we have

$$
\begin{aligned}
\mathcal{E}_{\mathrm{app}}^2 &= \sup_{p \in \mathcal{H}_{d_0,1}^\alpha} \inf_{f \in \mathcal{T}_R^D} \|f - p\|_2^2 \geq \inf_{f \in \mathcal{T}_R^D} \|f - f^*\|_2^2 = \mathcal{L}(f_{\mathrm{app}}) - \mathcal{L}(f^*) \\
&= \mathbb{E}_{\mathbb{D}}\left[\widehat{\mathcal{L}}(f_{\mathrm{app}}) - \mathcal{L}(f^*)\right] \geq \mathbb{E}_{\mathbb{D}}\left[\widehat{\mathcal{L}}(\hat{f}) - \mathcal{L}(f^*)\right] \geq \mathbb{E}_{\mathbb{D}}\left[\widehat{\mathcal{L}}(f_{s(\tau)}) - \mathcal{L}(f^*)\right] - \tau,
\end{aligned}
$$

where the third equation comes from the independence of $f_{\mathrm{app}}$ on the samples. Equivalently,

$$
\mathbb{E}_{\mathbb{D}}\left[\widehat{\mathcal{L}}(f_{s(\tau)}) - \mathcal{L}(f^*)\right] \leq \mathcal{E}_{\mathrm{app}}^2 + \tau.
\tag{57}
$$

Combine (55) and (57), we finish the proof. $\qquad\square$

## D.2. Upper Bound of the Statistical Error

To bound a possible infinite function class $\mathcal{F}$, we first introduce the covering number.

**Definition D.3** (Covering Number). Given a metric space $(\mathcal{X}, \rho)$ and a subset $S \subseteq \mathcal{X}$. Let the radius $\mu > 0$, a set $T \subseteq \mathcal{X}$ is called a $\mu$-covering of $S$ if $S \subseteq \bigcup_{y \in T} B_\rho(y, \mu)$, where $B_\rho(y, \mu)$ is the ball centered at $y$. The $\mu$-covering number of $S$ is denoted by

$$
\mathcal{N}(\mu, S, \rho) = \min\{|T| \mid T \text{ is a } \mu\text{-covering of } S\}.
$$

For a function class $\mathcal{F}$ from $\mathcal{X}$ to $\mathbb{R}$. Given a sample sequence $x_{1:n} = (x^{(1)}, x^{(2)}, \ldots, x^{(n)})$ for $n \in \mathbb{N}$, let $\mathcal{F}(x_{1:n}) = \{(f(x^{(1)}), f(x^{(2)}), \ldots, f(x^{(n)}) \mid f \in \mathcal{F}\}$. The uniform covering number of $\mathcal{F}$ is denoted by

$$
\mathcal{N}_\rho(\mu, \mathcal{F}, n) = \max\{\mathcal{N}(\mu, \mathcal{F}(x_{1:n}), \rho) \mid x_{1:n} \in \mathcal{X}^n\}.
$$

We prove Theorem 6.1 based on the following two lemmas.

**Lemma D.4** (Bounding Statistical Error via Covering Numbers). *Let $N$ be the number of samples. Suppose that $\|f\| \leq B$ for any $f \in \mathcal{F}$, the loss function $l(f, z)$ is a $\lambda$-Lipschitz function on $f$, and $\mathcal{L}(f)$ is locally strong convex with parameter $c$ around the target function $f^*$, i.e., $\mathcal{L}(f) - \mathcal{L}(f^*) \geq c\|f - f^*\|_2^2$. We have*

$$
\mathcal{E}_{\mathrm{sta}} \leq \frac{(8\lambda^2/c + 16\lambda B)(\log \mathcal{N}_\infty(\mu, \mathcal{T}_R^D, N) + 1)}{N} + 3\lambda\mu,
\tag{58}
$$

*where $\mathcal{N}_\infty(\mu, \mathcal{T}_R^D, N) := \mathcal{N}$ is the covering number of $\mathcal{T}_R^D$ with $N$ samples and the covering $\mathcal{C} = \{f_1, \ldots, f_\mathcal{N} \mid f \in \mathcal{T}_R^D\}$ under radius $\mu$ and metric $\|\cdot\|_\infty$.*

**Lemma D.5.** *Let $\mu > 0$, and $N \in \mathbb{N}_+$. For the designed Transformer function class $\mathcal{T}_R^D$, the covering number can be bounded, i.e., $\log \mathcal{N}_\infty(\mu, \mathcal{T}_R^D, N) \lesssim D^4 \log(eKND^{-4}/\mu)$,*

**Theorem 6.1** (Upper Bound of Statistical Error; Restatement). *Let $\mu > 0$ and dataset size $N \in \mathbb{N}_+$. For the Transformer function class $\mathcal{T}_R^D$ and target $f^* \in \mathcal{H}_{d_0,1}^\alpha([0,1]^{d_0}, K)$, the statistical error $\mathcal{E}_{\mathrm{sta}} = \mathbb{E}_{\mathbb{D}}[\mathcal{L}(f_{s(\tau)}) - 2\widehat{\mathcal{L}}(f_{s(\tau)}) + \mathcal{L}(f^*)]$ is upper bounded by*

$$
\mathcal{E}_{\mathrm{sta}} \lesssim \frac{(16K + 8)(D^4 \log(eKND^{-4}/\mu) + 1)}{N} + 3\mu,
$$

*where $\mu$ is the adjustable radius parameter.*

**Proof of Theorem 6.1.** We can derive that all $f \in \mathcal{H}^{\alpha}_{d_0,1}(\mathcal{X}, K)$ holds $\|f\| \leq K$ and $l(f, z) = (f(x) - y)^2$ is a 1-Lipschitz function, along with local strong convexity $c = 1$ of $\mathcal{L}$ at $f^*$ as

$$
\begin{aligned}
\mathcal{L}(f) - \mathcal{L}(f^*) &= \mathbb{E}_z \left[ (f(x) - f^*(x))^2 \right] + 2\mathbb{E}_z \left[ (f(y) - f^*(x))(f^*(x) - y) \right] \\
&= \mathbb{E}_z \left[ (f(x) - f^*(x))^2 \right] + 2\mathbb{E}_y \mathbb{E}_x \left[ (f(x) - f^*(x))(f^*(x) - y) \mid x \right] \\
&= \mathbb{E}_z \left[ (f(x) - f^*(x))^2 \right],
\end{aligned}
$$

where the last equation comes from $f^* = \mathbb{E}[y \mid x]$ under the loss function $l(f, z) = (f(x) - y)^2$. Consequently, we can derive the result based on Lemma D.4,

$$
\mathcal{E}_{\text{sta}} = \mathbb{E}_{\mathbb{D}} \left[ \mathcal{L}(f_{s(\tau)}) - 2\widehat{\mathcal{L}}(f_{s(\tau)}) + \mathcal{L}(f^*) \right] \leq \frac{(16K + 8)(\log \mathcal{N}_{\infty}(\mu, \mathcal{T}^D_R, N) + 1)}{N} + 3\mu. \tag{59}
$$

Further with Lemma D.5, we can obtain $\mathcal{E}_{\text{sta}} \lesssim (16K + 8)(D^4 \log(eKND^{-4}/\mu) + 1)/N + 3\mu.$ $\qquad\square$

**Proof of Lemma D.4.** Let $\mathbb{D}' = \{z_g^{(1)}, \ldots, z_g^{(N)}\}$ be an i.i.d. ghost sample set independent of $\mathbb{D} = \{z^{(1)}, \ldots, z^{(N)}\}$. For the sake of simplicity, denote $g(f, z) = l(f, z) - l(f^*, z)$ and $G(f, z) = \mathbb{E}_{\mathbb{D}'}[g(f, z_g) - 2g(f, z)]$ in this proof, and we have for the statistical error

$$
\begin{aligned}
\mathcal{E}_{\text{sta}} &= \mathbb{E}_{\mathbb{D}} \left[ \mathcal{L}(f_{s(\tau)}) - 2\widehat{\mathcal{L}}(f_{s(\tau)}) + \mathcal{L}(f^*) \right] = \mathbb{E}_{\mathbb{D}} \left[ \mathcal{L}(f_{s(\tau)}) - \mathcal{L}(f^*) - 2(\widehat{\mathcal{L}}(f_{s(\tau)}) + \mathcal{L}(f^*)) \right] \\
&= \mathbb{E}_{\mathbb{D}} \left[ \mathbb{E}_{\mathbb{D}'} \left[ \frac{1}{N} \sum_{i=1}^N g(f_{s(\tau)}, z_g^{(i)}) \right] - \frac{2}{N} \sum_{i=1}^N g(f_{s(\tau)}, z^{(i)}) \right] \\
&= \mathbb{E}_{\mathbb{D}} \left[ \frac{1}{N} \sum_{i=1}^N \mathbb{E}_{\mathbb{D}'} \left[ g(f_{s(\tau)}, z_g^{(i)}) - 2g(f_{s(\tau)}, z^{(i)}) \right] \right] = \mathbb{E}_{\mathbb{D}} \left[ \frac{1}{N} \sum_{i=1}^N G(f_{s(\tau)}, z^{(i)}) \right],
\end{aligned}
$$

where we use the fact that $\mathcal{L}(f) = \mathbb{E}_{\mathbb{D}}[\widehat{\mathcal{L}}(f)] = \mathbb{E}_{\mathbb{D}}[\frac{1}{N} \sum_{i=1}^N l(f, z^{(i)})] = \mathbb{E}_{\mathbb{D}'}[\frac{1}{N} \sum_{i=1}^N l(f, z_g^{(i)})].$

Suppose the covering $\mathcal{C}$ of $\mathcal{F}$ is $\{f_1, \ldots, f_{\mathcal{N}}\}$ with the covering number $\mathcal{N}_{\infty}(\mu, \mathcal{F}, N) := \mathcal{N}$. For any $f \in \mathcal{F}$, there exists a $\tilde{f} \in \mathcal{C}$ such that

$$
\left| g(f, z) - g(\tilde{f}, z) \right| = \left| l(f, z) - l(\tilde{f}, z) \right| \leq \lambda \left| f(z) - \tilde{f}(z) \right| \leq \lambda \mu, \text{ and } G(f, z) \leq G(\tilde{f}, z) + 3\lambda\mu,
$$

resulting in that for any $t > 0$

$$
\mathbb{P} \left[ \frac{1}{N} \sum_i G(f_{s(\tau)}, z^{(i)}) > t \right] \leq \mathbb{P} \left[ \max_{f \in \mathcal{F}} \frac{1}{N} \sum_i G(f, z^{(i)}) > t \right] \leq \mathbb{P} \left[ \max_{\tilde{f} \in \mathcal{C}} \frac{1}{N} \sum_i G(\tilde{f}, z^{(i)}) > t - 3\lambda\mu \right].
$$

Further, for any $\tilde{f} \in \mathcal{C}$, we clarify that $g(\tilde{f}, z^{(i)})$ is a bounded random variable since

$$
\left| g(\tilde{f}, z^{(i)}) \right| = \left| l(\tilde{f}, z^{(i)}) - l(f^*, z^{(i)}) \right| \leq 2\lambda B, \text{ and } \left| g(\tilde{f}, z^{(i)}) - \mathbb{E}_{\mathbb{D}} \left[ g(\tilde{f}, z^{(i)}) \right] \right| \leq 4\lambda B,
$$

with the expectation satisfying

$$
\begin{aligned}
\mathbb{E}_{\mathbb{D}} \left[ g(\tilde{f}, z^{(i)}) \right] &= \mathbb{E}_{\mathbb{D}} \left[ l(\tilde{f}, z^{(i)}) - l(f^*, z^{(i)}) \right] = \left[ \mathcal{L}(\tilde{f}) - \mathcal{L}(f^*) \right] \\
&\geq c \cdot \mathbb{E}_{\mathbb{D}} \left[ |\tilde{f} - f^*|^2 \right] \geq \frac{c}{\lambda^2} \mathbb{E}_{\mathbb{D}} \left[ g(\tilde{f}, z^{(i)})^2 \right] \geq \frac{c}{\lambda^2} \text{Var} \left[ g(\tilde{f}, z^{(i)})^2 \right] := \frac{c\sigma^2}{\lambda^2},
\end{aligned}
$$

where we use the $\lambda$-Lipschitz continuity of $l(f, z)$ on $f$ and the locally strong convexity of $\mathcal{L}(f)$ with parameter $c$. Therefore,

for any $\tilde{f} \in \mathcal{C}$,

$$
\begin{aligned}
\mathbb{P}\Big[\frac{1}{N}\sum_i G(\tilde{f}, z^{(i)}) > t\Big] &= \mathbb{P}\Big[\frac{1}{N}\sum_i \mathbb{E}_{\mathbb{D}'}\big[g(\tilde{f}, z_g^{(i)})\big] - \frac{2}{N}\sum_i g(\tilde{f}, z^{(i)}) > t\Big] \\
&= \mathbb{P}\Big[\mathbb{E}_{\mathbb{D}}\Big[\frac{1}{N}\sum_i g(\tilde{f}, z^{(i)})\Big] - \frac{1}{N}\sum_i g(\tilde{f}, z^{(i)}) > \frac{t}{2} + \frac{1}{2}\cdot \mathbb{E}_{\mathbb{D}}\Big[\frac{1}{N}\sum_i g(\tilde{f}, z^{(i)})\Big]\Big] \\
&\leq \mathbb{P}\Big[\mathbb{E}_{\mathbb{D}}\Big[\frac{1}{N}\sum_i g(\tilde{f}, z^{(i)})\Big] - \frac{1}{N}\sum_i g(\tilde{f}, z^{(i)}) > \frac{t}{2} + \frac{c\sigma^2}{2\lambda^2}\Big] \\
&\leq \exp\Big(-\frac{N}{2(\sigma^2 + 4\lambda B(\frac{t}{2} + \frac{c\sigma^2}{2\lambda^2}))}\cdot\Big(\frac{t}{2} + \frac{c\sigma^2}{2\lambda^2}\Big)^2\Big) \\
&\leq \exp\Big(-\frac{N}{4(\lambda^2/c + 2\lambda B)}\cdot\Big(\frac{t}{2} + \frac{c\sigma^2}{2\lambda^2}\Big)\Big) \\
&\leq \exp\Big(-\frac{Nt}{8\lambda^2/c + 16\lambda B}\Big),
\end{aligned}
$$

where the second inequality comes from the Bernstein's inequality for bounded random variables, the third inequality comes from $\sigma^2 \leq \frac{2\lambda^2}{c}(\frac{t}{2} + \frac{c\sigma^2}{2\lambda^2})$ and the last result is obtained by $\frac{t}{2} + \frac{c\sigma^2}{2\lambda^2} \leq \frac{t}{2}$.

It is noticeable that for any $t > 3\lambda\mu$, with assistance of the $\mu-$covering, we have

$$
\mathbb{P}\Big[\max_{\tilde{f}\in\mathcal{C}} \frac{1}{N}\sum_i G(\tilde{f}, z^{(i)}) > t - 3\lambda\mu\Big] \leq \mathcal{N}\cdot\exp\Big(-\frac{N(t-3\lambda\mu)}{8\lambda^2/c + 16\lambda B}\Big). \tag{60}
$$

Further with (60), we choose $c_0 = (8\lambda^2/c + 16\lambda B)\log\mathcal{N}/N$ and get

$$
\begin{aligned}
\mathcal{E}_{\mathrm{sta}} = \mathbb{E}_{\mathbb{D}}\Big[\frac{1}{N}\sum_{i=1}^{N} G(f_{s(\tau)}, z^{(i)})\Big] &= \int_0^{+\infty} \mathbb{P}\Big[\frac{1}{N}\sum_i G(f_{s(\tau)}, z^{(i)}) > t\Big]\mathrm{d}t \\
&\leq 3\lambda\mu + c_0 + \int_{3\lambda\mu+c_0}^{+\infty} \mathbb{P}\Big[\max_{\tilde{f}\in\mathcal{C}} \frac{1}{N}\sum_i G(\tilde{f}, z^{(i)}) > t\Big]\mathrm{d}t \\
&\leq 3\lambda\mu + c_0 + \int_{3\lambda\mu+c_0}^{+\infty} \mathcal{N}\cdot\exp\Big(-\frac{N(t-3\lambda\mu)}{8\lambda^2/c + 16\lambda B}\Big)\mathrm{d}t \\
&\leq 3\lambda\mu + c_0 + \mathcal{N}\cdot\frac{8\lambda^2/c + 16\lambda B}{N}\exp\Big(-\frac{Nc_0}{8\lambda^2/c + 16\lambda B}\Big) \\
&\leq \frac{(8\lambda^2/c + 16\lambda B)(\log\mathcal{N} + 1)}{N} + 3\lambda\mu.
\end{aligned}
$$

The bound is proved. □

**Bounding the Covering Number.** The current proof of Lemma D.5 turns to bound the uniform covering number $\mathcal{N} = \mathcal{N}_\infty(\mu, \mathcal{F}, n)$ with $\mathcal{F} = \mathcal{T}_R^D$ and $n = N$. We rely on the following theorem.

**Theorem D.6** (Anthony & Bartlett, 2009, Theorem 12.2). *Let $\mathcal{F}$ be a set of real functions from a domain $\mathcal{X}$ to the bounded interval $[0, K]$. Let $\mu > 0$ and suppose that the pseudo-dimension of $\mathcal{F}$ is $d_{\mathrm{p}}$. For $m \geq d_{\mathrm{p}}$,*

$$
\mathcal{N}_\infty(\mu, \mathcal{F}, n) \leq \sum_{i=1}^{d_{\mathrm{p}}} \binom{n}{i}\Big(\frac{K}{\mu}\Big)^i \leq \Big(\frac{enK}{\mu d_{\mathrm{p}}}\Big)^{d_{\mathrm{p}}}.
$$

***Proof of Lemma D.5.*** As for any real function class $\mathcal{F}$, its pseudo-dimension is equal to the VC-dimension of the relative subgraph class defined as

$$
B_{\mathcal{F}} = \{B_f(x, y) = \mathrm{sgn}(f(x) - y) : f \in \mathcal{F}\}.
$$

Therefore, for the Transformer function class $\mathcal{T}_R^D$, we have $\mathrm{Pdim}(\mathcal{T}_R^D) = \mathrm{VCdim}(B_\mathcal{T})$. Bounding $\mathcal{N}_\infty(\mu, \mathcal{T}_R^D, N)$ now turns to bound the VC-dimension of $B_\mathcal{T}$.

Considering $y$ as an extra input variable, for any $f \in \mathcal{T}_R^D$, $B_\mathcal{T}$ only requires two more operations than that of $\mathcal{T}_R^D$. Based on (45), this also leads to $\mathrm{VCdim}(B_\mathcal{T}) \lesssim D^4$. With Theorem D.6, we get the bound of covering number

$$\log \mathcal{N}_\infty(\mu, \mathcal{T}_R^D, N) \leq \mathrm{VCdim}(B_\mathcal{T}) \cdot \log\left(\frac{2eKN}{\mu \mathrm{VCdim}(B_\mathcal{T})}\right) \lesssim D^4 \log\left(\frac{eKND^{-4}}{\mu}\right), \tag{61}$$

where the second inequality is valid when $N \gtrsim D^4$. And we also omit the constant "2" in $\lesssim$. The proof is complete. $\square$

**Remark D.7** (Novelty of Proof). The proof of statistical error upper bound amounts to bounding the uniform covering number of Transformers given a finite sample size. The derivation of (61) utilizes our novel VC-dimension upper bound (45). Besides, this approach avoids the need to estimate the uniform covering number by computing the Lipschitz constants of Transformer layers. Instead, it relies on the VC-dimension of the model parameter space, which circumvents the exponentially exploding matrix norm term in the covering number upper bound found in (Hu et al., 2024, Lemma F.6) and (Edelman et al., 2022, Theorem 4.7).

### D.3. Convergence Rate of Excess Risk

**Theorem 6.2** (Convergence Rate of Excess Risk; Restatement). *Assume the regression function $f^* \in \mathcal{H}_{d_0,1}^\alpha([0,1]^{d_0}, K)$ for some $\alpha \in (0,1]$ and $K > 0$. Let $f_{s(\tau)}$ be the estimator over the i.i.d samples $\{(x_i, y_i)\}_{i=1}^N$, the excess risk bound holds,*

$$\mathbb{E}_\mathbb{D}\left[\mathcal{E}(f_{s(\tau)})\right] - 2\tau \lesssim N^{-\frac{\alpha}{2d_0+\alpha}} \log(N) + N^{-\frac{\alpha}{2d_0+\alpha}},$$

*where $\tau$ is the optimization error.*

***Proof of Theorem 6.2.*** The approximation upper bound of Theorem 4.1 equivalently reveals that

$$\mathcal{E}_{\mathrm{app}}^2 = \sup_{f \in \mathcal{H}_{d_x,d_y}^\alpha} \inf_{g \in \mathcal{T}_R^D} \|f - g\|_2^2 \lesssim D^{-\frac{2\alpha}{d_0}}. \tag{62}$$

Further with (13), (15) and (62), let $\mu = D^{-\frac{2\alpha}{d_0}}$ and $D = N^{\frac{d_0}{4d_0+2\alpha}}$, we can derive the excess risk rate

$$\begin{aligned}
\mathbb{E}_\mathbb{D}\left[\mathcal{E}(f_{s(\tau)})\right] - 2\tau &\leq \mathcal{E}_{\mathrm{sta}} + 2\mathcal{E}_{\mathrm{app}}^2 \\
&\lesssim (16K+8)N^{-1}(D^4 \log(eKND^{-4}/\mu) + 1) + 3\mu + 2D^{-\frac{2\alpha}{d_0}} \\
&\lesssim N^{-\frac{\alpha}{2d_0+\alpha}} \log(N) + N^{-\frac{\alpha}{2d_0+\alpha}}.
\end{aligned} \tag{63}$$

The result is proved. $\square$

