# OpenReview forum: "Approximation Error Upper and Lower Bounds  for Hölder Class with Transformers"
_ICML.cc/2026/Conference — ICML 2026 regular_

### Official Review · Reviewer_oDY8 · 2026-03-12

**Soundness:** 3
**Presentation:** 3
**Significance:** 3
**Originality:** 3
**Overall Recommendation:** 5
**Confidence:** 4

**Summary:**

This paper studies the approximation power of standard Transformers with Softmax attention, ReLU activations, and residual connections. The authors analyze the approximation error for Hölder function classes and provide both upper and lower bounds on the number of Transformer blocks required to achieve a given approximation accuracy.

Specifically, they show that a standard Transformer can approximate Hölder functions with approximation error ε using $O(\varepsilon^{-d_0/\alpha})$ blocks, while at least $\Omega(\varepsilon^{-d_0/(4\alpha)})$ blocks are necessary based on VC-dimension arguments. The paper also connects these approximation results to statistical learning by deriving excess risk bounds for regression problems.

A key technical contribution is a constructive approximation framework consisting of three modules: quantization, contextual mapping via self-attention, and value mapping. Importantly, the analysis applies to the standard Transformer architecture without modifying Softmax attention or removing residual connections.

**Compliance With Llm Reviewing Policy:**

Affirmed.

**Final Justification:**

I am increasing my score from 4 (weak accept) to 5 (accept) after reading the authors’ rebuttal. The rebuttal clearly addressed my main concerns in a thoughtful and technically grounded way. In particular, the explanation that the gap in the approximation rates primarily stems from known limitations of VC-dimension bounds is convincing and appropriately contextualizes the current results.

The authors also clarified the role of a single attention layer as a minimal and clean theoretical abstraction, which helped me better appreciate the conceptual contribution of the work. Additionally, their positioning of the results as expressivity guarantees---rather than claims about optimization or practical training---makes the scope of the paper clearer and more appropriate.

Overall, I find the paper to be sound and well-presented, with solid originality in analyzing standard Transformers without architectural simplifications. While the bounds are not yet tight, the work provides a meaningful and timely step toward understanding Transformer expressivity. The rebuttal strengthened my confidence in the significance and correctness of the contributions, leading me to a stronger acceptance recommendation.

**Key Questions For Authors:**

1. The upper and lower bounds differ by a factor in the exponent. Do the authors believe this gap mainly comes from the VC-dimension bound or from the constructive approximation scheme?

2. The construction relies primarily on a single self-attention layer for contextual mapping. Could multiple attention layers improve the approximation rate or simplify the construction?

3. How should the theoretical construction be interpreted relative to practical Transformers trained by gradient methods? Is the result mainly an expressivity guarantee, or does it suggest something about learned representations?

**Limitations:**

yes

**Strengths And Weaknesses:**

## Strengths

The main strength of the paper is that it provides theoretical guarantees for standard Transformers, rather than simplified variants often considered in prior work. The analysis keeps Softmax attention and residual connections, making the results more relevant to practical architectures.

Another strength is that the paper provides both upper and lower bounds for approximation error. Lower bounds for Transformer expressivity are relatively scarce, and the VC-dimension–based analysis provides useful insight into the minimal number of blocks required.

The paper also goes beyond pure approximation theory by connecting the results to statistical learning, deriving excess risk bounds in regression settings. This helps place the theoretical results in a broader learning-theoretic context.

Overall, the work contributes to a growing body of literature aiming to understand the theoretical foundations of Transformer models.

## Weaknesses

The main limitation is that the gap between the upper and lower bounds remains fairly large. The upper bound scales as $O(\varepsilon^{-d_0/\alpha})$, while the lower bound is $\Omega(\varepsilon^{-d_0/(4\alpha)})$. The authors acknowledge that this gap partly comes from the construction used for the value mapping and from loose VC-dimension bounds.

Another limitation is that the constructive approximation framework is somewhat complex and may not fully reflect how Transformers operate in practice. In particular, the role of self-attention in the construction is relatively limited compared to the feed-forward components.

Finally, the statistical rates derived for regression are not optimal compared to known minimax rates for neural networks, largely due to the loose complexity bounds.

Despite these limitations, the work provides useful theoretical insight and a meaningful step toward understanding Transformer approximation capabilities.

---

> ### Author Rebuttal · Authors · 2026-03-29
>
> We thank the reviewer for appreciating our constructive approximation framework and our comprehensive theoretical guarantees. We address your comments (W1-W3, Q1-Q3) below:
>
> **1. Source of the gap in the exponent (W1, W3, Q1)**
>
> **Answer:** We thank the reviewer for this insightful question. We believe this gap in approximation bounds results **from the looseness of current VC-dimension bound**. Consequently, this gap also affects the optimality of our excess risk rates in regression.
> - **Looseness of the lower bound:** The current VC-dimension tools for Softmax activations (e.g., Theorem 8.14 in Anthony & Bartlett, 2009) are known to be loose, yet the tightest available. Deriving tighter bounds for such activations remains an open mathematical challenge.
> - **Potential for a tighter upper bound:** Our current approximation upper bounds match Yarotsky's [1] optimal rates for pure FFNs. However, because Transformers incorporate self-attention mechanisms, they inherently possess greater expressive capacity than pure FFNs. This inherent advantage indicates that deriving even tighter upper bounds is a highly promising direction for future research. In the meantime, **our current approximation serves as the  theoretical baseline** necessary to support those advancements.
> - [1] Yarotsky, Dmitry. "Error bounds for approximations with deep ReLU networks." _Neural networks_ 94 (2017): 103-114.
>
> **2. Clarification on analyzing a single attention layer (W2, Q2)**
>
> **Answer:** We appreciate the opportunity to clarify our theoretical design choice of analyzing a single attention layer.
> - **Sufficiency (W2):**  As discussed in Line 287, a single attention layer is the minimal structural requirement to achieve the contextual mapping. This part enables sequence models to grasp contextual meaning in the mathematical setting.
> - **Theoretical construction (Q2):**  Analyzing **multiple layers does not simplify** the theoretical construction. Instead, by focusing on a single layer, our proof isolates the core mechanism of self-attention and provides **a clean, rigorous baseline** for its minimal requirements in approximation theory.
> - **Approximation rates and complexity (Q2):** While stacking attention layers strictly preserves or improves the approximation power of the model, deriving bounds for multiple layers introduces significant theoretical complexity due to the highly compositional nature of deep Transformers. We leave multiple attention layers to future work.
>
> **3. Theoretical insights for practical Transformers (Q3)**
>
> **Answer:** We appreciate the reviewer's question, which allows us to clarify the focus and scope of our work.
> - **Expressivity guarantee:** Our approximation rates establish a rigorous expressivity guarantee. We mathematically prove that the standard Transformer architecture is powerful enough to represent complex (e.g. Hölder) functions.
> - **Relation to practical Transformers:** These rates demonstrate that the Transformer provides a sufficiently rich hypothesis space for complex tasks, where **the "best approximant" $f_{app}$ in Eq.(56) serves as the ultimate target for optimizers** like gradient descent. However, as is standard in statistical learning theory, we decouple this inherent expressive power from the highly non-trivial optimization dynamics.  Analyzing such dynamics falls outside our current scope. Thus, we leave the theoretical study of how models converge to this approximant to future work, for which our bounds provide a rigorous baseline.
> - **Insights into learned representations:** While we do not study how the representations are learned or the learning process, our work provides insights into the **underlying functional mechanism** of standard Transformers.  Specifically, the attention layer handles cross-token contextual mapping, while FFNs perform local function fitting.
>
> We sincerely thank the reviewer for the rigorous evaluation and for recognizing the value of our constructive framework and complete bounds. We hope our clarifications have fully addressed your concerns.

---

> > ### Author Rebuttal · Reviewer_oDY8 · 2026-04-01
> >
> > The authors provide clear and thoughtful responses to my main concerns. In particular, they convincingly explain that the gap in the exponent is primarily due to the looseness of current VC-dimension bounds, which is a known limitation rather than a flaw specific to this work. The clarification on the role of a single attention layer also helps better understand the design choice as a minimal and clean theoretical baseline.
> >
> > Overall, the responses appropriately contextualize the limitations and strengthen my confidence in the paper’s contributions as a solid foundational step in understanding Transformer expressivity.
> > So I will raise my recommendation from 4 to 5.

---

> > > ### Author Response · Authors · 2026-04-06
> > >
> > > We thank the reviewer for acknowledging our rebuttal and raising the score. We are encouraged that our clarifications regarding the VC-dimension bounds and the single-layer theoretical baseline addressed the reviewer's concerns. These insightful comments have helped us properly contextualize these limitations and highlight the foundational value of our work. We sincerely appreciate the time and effort the reviewer has dedicated to reviewing our paper.

---

### Official Review · Reviewer_77ow · 2026-03-13

**Soundness:** 1
**Presentation:** 2
**Significance:** 2
**Originality:** 2
**Overall Recommendation:** 4
**Confidence:** 3

**Summary:**

This paper studies the approximation and learning capacity of standard Transformer over Hölder smooth function classes, with an explicit constructive approximation scheme, and derives a complementary depth lower bound using VC dimension. It also links capacity control to a regression generalization guarantee.

**Compliance With Llm Reviewing Policy:**

Affirmed.

**Final Justification:**

The rebuttal addressed my main concerns and improved my evaluation. It made the bounded width scaling regime more explicit, and refined the regression discussion as a purely theoretical statement rather than an empirical claim about practical performance. On that basis, I am raising my score.

**Key Questions For Authors:**

1. The proof upgrades an $L^2$ approximation bound to an $L^\infty$ uniform bound. In general this implication can fail via spiky functions that have small $L^2$ error but large pointwise error on a small-measure set. Which additional property in the paper’s setting makes this step valid?

2. The complexity bounds are expressed mainly in terms of $D$. What is held fixed versus allowed to scale with $D$ (number of heads, embedding size), and how would the stated bounds change if these quantities scale?

3. Which component is viewed as the limiting factor for tightening the lower bound? A short discussion would help interpret whether the current gap is methodological or intrinsic.

**Limitations:**

Yes

**Strengths And Weaknesses:**

Strengths

- The paper provides a concrete, modular constructive scheme that realizes the approximation of Hölder smooth functions with a standard Transformer architecture, making the proof ideas relatively explicit.

- It provides a capacity-based lower bound and connects the approximation story to a regression generalization bound.

Weaknesses

- The technical novelty is limited, as the upper bound follows a discretize then lookup blueprint and the lower bound follows a standard VC dimension route, while leaving a substantial gap between upper and lower bounds.

- The paper claims empirical effectiveness in real world regression settings but does not provide corresponding experimental results or an evaluation section to substantiate the claim.

---

> ### Author Rebuttal · Authors · 2026-03-29
>
> We thank the reviewer for recognizing our approximation results. We address your comments (W1-W2, Q1-Q3) below:
>
> **1. Clarifications for the proof strategy (Q1)**
>
> **Answer:** We completely agree that a global $L^2$ bound does not imply a global $L^\infty$ bound due to potential spiky functions on small-measure sets. To clarify, **our proof rigorously avoids this:**
> - **Strict reliance on $L^2$:**  Our main theorems and proofs rely **strictly on the $L^2$-norm**. We make no claims regarding  $L^\infty$-norm approximation results.
> - **Theoretical extension to $L^{\infty}$:** While our current results do not require it, one could theoretically overcome spiky points using a medium-value selection FFN block [1]. This technique replaces spiky points with adjacent regular points to establish a uniform bound.
> In summary, we do not improperly upgrade an $L^2$ bound to a global $L^\infty$ bound.
> - **Ref:** [1] Lu, Jianfeng, et al. "Deep network approximation for smooth functions." _SIAM Journal on Mathematical Analysis_ 53.5 (2021): 5465-5506.
>
> **2. States of parameters with scaling $D$ (Q2)**
>
> **Answer:** In our asymptotic regime, the block number $D$ is the primary scaling variable. Inner parameters are fixed or bounded:
> - **Fixed:**
> 	- Heads number $r_1=1$ (Lemma 4.9), and $r_{2}\in\{1,3,4\}$ (see Eq.(24), Eq.(38), and $\mathrm{FF^{(extra)}}$ in Line 1056).
> 	- Embedding size $d_0$ is fixed once the input embedding is finished. $d_0=d\times L$, with fixed token dimension $d$ and fixed sequence length $L$.
> 	- Feed-forward width $(l)$: Tied to the token dimension $d$ due to matrix multiplication mechanism of the network, yielding $l=d$.
> - **Bounded:**
> 	- Attention size $(m)$: As demonstrated in Line 1203, $m$ is theoretically arbitrary but practically bounded $m\le d$.
> - **Impact of scaling inner parameters:**
> 	- **Scaling embedding size ($d_0$):** If $d_0$ scales, the approximation task becomes more challenging due to increased dimension. But our bounds remain fully valid.
> 	- **Scaling the width $W$:** Our current technical route considers a bounded-width ($W= \max (r_1\cdot m, r_2\cdot l )$) regime and establishes bounds of depth $D$ for theoretical convenience. With Remark B.3 in Line 752, our architecture allows for converting the depth burden into a larger width. That is, scaling the width $W$ can also decrease the approximation error.  We will further discuss on this depth-width relationship in the final version.
> 	- **Future regimes:** Scaling $W$ asks for different theoretical regimes, which is a promising future work. Our bounded-width theorem serves as a rigorous baseline.
>
> **3. Loose lower bound & gap (Q3)**
>
> **Answer:** With brief discussions in "Tightness Analysis" (point ii, page 7), we clarify that this gap is **intrinsic**, not methodological:
> - **Limiting factor:** The bottleneck for tightening the lower bound is **the inherent looseness of current VC-dimension upper bounds** for networks employing transcendental (Softmax) activation functions. We use Theorem 8.14 in Anthony & Bartlett (2009), the tightest available tool. Tighter bounds for such activations remain an open mathematical challenge.
>
> **4. Clarification on the technical novelty (W1)**
>
> **Response:** We thank the reviewer for this feedback. We would like to clarify our technical contributions:
> - **Architecture novelty:** While "discretize-then-lookup" is a known high-level blueprint, implementing it with the standard Transformer architecture (Softmax self-attention, feed-forward layer, residual connections) is highly **non-trivial** and mathematically challenging.
> - **Complete approximation framework:** Our core contribution is beyond a single proof. We establish complementary approximation lower bounds along with precise upper bounds, advancing the fundamental theory of Transformers and rigorously defining this **intrinsic** gap.
> - **Functional roles in Transformers:** Our analysis clarifies functional roles of Transformers: the attention handles context routing, while FFNs perform function fitting. This can guide possible architectural adjustments for specific tasks.
>
> **5. Clarification on the theoretical scope (W2)**
>
> **Response:**  We completely understand how our introduction might have naturally led to expectations for experiments.
> - **Terminology:** We respectfully clarify that our work is strictly theoretical. The term "empirical" refers to the empirical risk minimization (ERM) within statistical learning theory, not real-world coding experiments.
> - **Scope refinement**: Accordingly, we will revise the introduction to better highlight our **purely theoretical regression setting**, separating it from practical optimization tasks in the final version.
>
> We appreciate your help in making our scope clearer and hope these resolve your concerns.

---

> > ### Author Rebuttal · Reviewer_77ow · 2026-04-04
> >
> > The rebuttal addresses my questions well and clarifies several points that were previously ambiguous. I still think the paper would benefit from a more careful presentation of its soundness, especially in clearly delimiting scope and stating the mathematical arguments as rigorously as possible, but my main rebuttal concerns have been adequately addressed so that I will raise my score.

---

> > > ### Author Response · Authors · 2026-04-06
> > >
> > > We thank the reviewer for acknowledging our rebuttal and raising the score. We greatly appreciate the reviewer's suggestions regarding the presentation of our theoretical soundness. In the final version, we will carefully incorporate this advice to further polish the manuscript. Specifically, we will explicitly delimit the scope and ensure all mathematical arguments are stated rigorously (e.g. by clarifying the specific norm utilized and detailing how parameters scale when necessary). We thank the reviewer again for their time and valuable input.

---

### Official Review · Reviewer_28hM · 2026-03-13

**Soundness:** 3
**Presentation:** 3
**Significance:** 3
**Originality:** 3
**Overall Recommendation:** 4
**Confidence:** 3

**Summary:**

This paper establishes both upper and lower bounds on the approximation error of standard Transformers, which comprises Softmax attention, ReLU activations, and residual connections, for H\"{o}lder smooth functions. For the upper bound, the authors construct a Transformer with at most $\mathcal{O}(\epsilon^{-d_0/\alpha})$ blocks that can approximate any function in a H\"{o}lder class with smoothness $\alpha$ and input dimension $d_0$ within an error $\epsilon$ in $L^2$ norm. For the lower bound, leveraging VC-dimension arguments, the paper demonstrates that any such Transformer achieving an $\epsilon$-approximation must have at least $\Omega(\epsilon^{-d_0/(4\alpha)})$ blocks.

**Compliance With Llm Reviewing Policy:**

Affirmed.

**Key Questions For Authors:**

Please see the weaknesses.

**Limitations:**

Please see the weaknesses.

**Strengths And Weaknesses:**

Strengths:
1. The paper provides the first quantitative characterization of the approximation capabilities of standard Transformers for Hölder classes. While universal approximation theorems for Transformers exist, concrete approximation rates are scarce, and this work fills an important gap in the theoretical understanding of these architectures.
2. The work covers both upper and lower bounds, providing a complete picture of the approximation power.

Weaknesses:
1. Some symbols are used incorrectly. In abstract, the authors said "Transformers demand for at least $\mathcal{O}(\varepsilon^{−d_0/(4α)})$ blocks to achieve the $\varepsilon$ approximation accuracy". Here the lower bound of the number of blocks should use $\Omega$ but not $\mathcal{O}$.
2. The lower bound and the statistical analysis hinge on the VC-dimension bound from Anthony & Bartlett (2009, Theorem 8.14). As noted by authors, this bound is likely not tight for the specific Transformer architecture, which may cause a loose lower bound.
3. The paper argues the width is bounded, but the depth is astronomical. However, it seems that practical transformer-based models are not too deep. And deep transformers are more difficult to train and deploy than wide ones with the same number of neurons, since that depth blocks parallel computing. Hence, this point weakens the value of the research work.

---

> ### Author Rebuttal · Authors · 2026-03-29
>
> We thank the reviewer for recognizing our work's novelty and completeness. We address your weaknesses & questions (Q1-Q3) below:
>
> **1. Incorrect use of some symbols (Q1)**
>
> **Answer:** We thank the reviewer for pointing out this notation error. We agree that $\Omega$ is the correct symbol to denote the lower bound of the number of blocks required. We have corrected $\mathcal{O}$ to $\Omega$ in the abstract and throughout the manuscript to ensure mathematical rigor in the final revision.
>
> **2. The gap between upper and lower bounds (Q2)**
>
> **Answer:** We appreciate the reviewer for identifying this as a potential weakness and question. We fully acknowledge that using the VC-dimension bound from Anthony & Bartlett (2009) results in a conservative lower bound. With discussions in our **"Tightness Analysis" (point ii, page 7)**, we further offer the following justifications:
> - **Technical necessity:** Tighter VC-dimension results for Sigmoid-like operators (including Softmax) remain a significant open challenge in the field. Currently, Theorem 8.14 in Anthony & Bartlett (2009) is the most rigorous tool available for such activations.
> - **Statistical validity:** Although the VC-dimension bound is loose, it provides a principled upper bound on model complexity. This ensures that our analysis remains **theoretically sound**.
> - **Significance:** Even with this gap, our result establishes the first quantitative (upper and lower) characterization of the approximation capabilities of standard Transformers for Hölder classes, providing a baseline for future theoretical improvements.
>
> **3. Deep or wide Transformers (Q3)**
>
> **Answer:** We acknowledge the challenges of practical training and deployment with extremely deep Transformers. However, our work focuses on approximation theory rather than hardware or optimization constraints.
> - **Theoretical paradigm:** Bounding width to study depth is a standard and effective approach in approximation theory. It allows us to isolate and quantify a network's compositional power and analyze the **fundamental limit of model’s expressivity**.
> - **Trade-off between depth and width.** As discussed in Remark B.3 (Line 752), our theoretical architecture allows for a structural trade-off, converting the depth burden into a larger width budget using the same number of neurons. We will further discuss on this depth-width relationship in the final version.
>
> We thank the reviewer again for the constructive evaluation. We hope our clarifications regarding the theoretical scope have fully addressed your concerns.

---

> > ### Author Rebuttal · Reviewer_28hM · 2026-04-04
> >
> > We thank the authors for their reply. To be fair, this paper would not have major issues even if it were accepted. However, there remains a certain gap between the upper and lower bounds on the number of Transformer blocks required to achieve an arbitrarily given approximation error. I understand that closing this gap is theoretically nontrivial, but I think at the very least one could compute a lower bound on the VC dimension of Transformers and then indicate the limits of the VC-dimension-based approach in proving approximation lower bounds. Regarding the depth–width trade-off issue, I do not see a clear statement in the form of a theorem in the main text. I hope that the authors can address this in the next version of the paper.

---

> > > ### Author Response · Authors · 2026-04-06
> > >
> > > We thank the reviewer for the constructive follow-up. To address the reviewer's points, **we will add** i) a remark on  the VC-dimension lower bound of Transformers and ii) a formal corollary stating the depth–width trade-off.  Specifically, the content will be:
> > >
> > > **1. Remark on the VC-dimension lower bound**
> > >
> > > **Response:** The following remark will be added after the "Tightness Analysis" (Line 346, page 7).
> > > - **Remark (On the VC-dimension lower bound):** Note that the class of fixed-width, depth $D$ feed-forward networks, denoted by $\mathcal{F}^D$, is a strict subset of the Transformer function class $\mathcal{T}_R^{D}$. This structural inclusion inherently yields $\mathrm{VCdim}(\mathcal{T}^D_R) \ge \mathrm{VCdim}(\mathcal{F}^D)$. Since Bartlett et al. [1] established that $\mathrm{VCdim}(\mathcal{F}^D) \ge \Omega(D^2)$, it follows that $\mathrm{VCdim}(\mathcal{T}^D_R)\ge \Omega(D^2)$. Therefore, to fully close the error gap and approach the desired approximation lower bound of $\Omega(\varepsilon^{-d_0/(2\alpha)})$, a tighter VC-dimension upper bound approaching this $\Omega(D^2)$ rate is required.
> > > - **Note:** This remark still aligns with our "Tightness Analysis" and Remark 6.4 (Optimality) in the main text regarding how to further bridge the theoretical gap.
> > > - **Ref:** [1] Bartlett, Peter L., et al. "Nearly-tight VC-dimension and pseudodimension bounds for piecewise linear neural networks." _Journal of Machine Learning Research_ 20.63 (2019): 1-17.
> > >
> > > **2. Theorem of the depth–width trade-off**
> > >
> > > **Response:** We formalize the depth–width trade-off as **Corollary xxx**, inserted immediately after Remark 4.2 (Bounded Width) in Line 237.
> > > - **Corollary xxx. (Depth-width Trade-off)**  For any $\varepsilon>0$ and $d_0=dL$, a Transformer architecture $\mathcal{T}_R^D$ with $D=\mathcal{O}(\varepsilon^{-d_0/\alpha})$ blocks and fixed width $W=4d$, capable of approximating any function $f\in{\mathcal{H} _ {d _ 0}^{\alpha}}(\mathcal{X},K)$ with error $\varepsilon$, can be realized by a Transformer $\mathcal{T}_R^{D',W'}$ with variable blocks $D'$ and width $W'$ when satisfying the condition $D'\cdot W'=\mathcal{O}(\varepsilon^{-d_0/\alpha}),$ where $D'\ge5$ and $W'\ge 4d$.
> > > - **Remark:** This corollary reveals a flexible depth-width trade-off. Rather than strictly adhering to our fixed-width regime, one can achieve the same approximation bound using a fixed-depth architecture with variable width, or more generally, an architecture where both depth and width vary simultaneously, as long as their product satisfies $D' \cdot W' = \mathcal{O}(\varepsilon^{-d_0/\alpha})$.
> > > - **We provide a proof sketch below (the full derivation will be provided in the final version):**
> > > 	- **Pipeline:** Our construction utilizes three key components: the quantization module, contextual mapping, and value mapping. The constraints $D'\ge5$ and $W'\ge 4d$ ensure the minimal architectural budget.
> > > 	- **Neuron capacity:** From the proofs of Lemma 4.6 (quantization module) and Lemma 4.10 (value mapping), the sequential construction requires $\mathcal{O}(d_0M)$ and $\mathcal{O}(LM^{d_0})$ total neurons, respectively. Crucially, these subunits operate on disjoint indices and do not interfere with one another (Lines 737 and 1050, those links will be well-specified in the final version).
> > > 	- **Block-diagonalization:** Leveraging the additive nature of quantization and value mapping, we can transform the sequential depth into parallel width. Specifically, the weight matrices of the original modules are embedded into a block-diagonal structure within the wider layers.
> > > 	- **Input replication:** To compress $n$ sequential layers into a single wider layer while satisfying the Transformer’s requirement for equal input and output dimensions, we employ an input replication scheme: expanding the input $X \in \mathbb{R}^{d \times L}$ to $\tilde{X} = [X; X; \dots; X] \in \mathbb{R}^{nd \times L}$. This allows the wide layer to process $n$ independent functional channels simultaneously.
> > > 	- **Final results:** By redistributing the total neuron budget, we obtain a parallel quantization module and value mapping satisfying $D'_1\cdot W_1'=\mathcal{O}(d_0M)$ and $D_2'\cdot W_2'=\mathcal{O}(LM^{d_0})$, respectively. Setting $M\lesssim \varepsilon^{-1/\alpha}$ yields the final trade-off relation $D' \cdot W' =\mathcal{O} (\varepsilon^{-d_0/\alpha})$, where the effective depth is $D' = D_1' + D_2' + 1$ and width $W' = \max(W_1', W_2', d)$.
> > >
> > > We thank the reviewer again for the constructive feedback, which greatly improves our theoretical completeness.

---

### Official Review · Reviewer_iFBF · 2026-03-13

**Soundness:** 3
**Presentation:** 3
**Significance:** 2
**Originality:** 2
**Overall Recommendation:** 4
**Confidence:** 3

**Summary:**

The paper provides approximation lower and upper bounds of transformer architectures for the Holder class. The transformer model includes softmax operators, ReLU activation and residual connections.
The authors prove that the bounded functions in Holder class can be approximated with the number of blocks scaling with inverse error with exponent proportional with input dimension. They give an application of such bounds for the case of regression, which bounds the excess risk as a function of number of samples. The main results are as follows:
* Theorem 4.1 provides the main upper bound. The key idea in the proof is to first approximate the function of interest with  piecewise linear function $f_\delta$, and then approximate each linear function with a transformer network (which actually has only a single attention layer). The transformer layer serves as the generator of the contextual id/mapping.
* Theorem 5.3 presents the lower bound using VC dimension results of Anthony and Bartlett.  The proof relies on approximation on the quantized function and use VC-dimension to find the dimension required for shattering the quantized points. Although there is a factor difference between the exponents of lower and upper bound, the other dependencies match, which is encouraging.
* Section 6 focuses on the application to the regression task in a realistic setting, where an ERM is considered for N data samples. They decompose the error into approximation error and statistical error, first one bounded from the result of previous sections. The statistical error is bounded using covering number trick for the empirical sum (in my opinion similar to steps of Dudley’s inequality in Rademacher analysis).

**Compliance With Llm Reviewing Policy:**

Affirmed.

**Final Justification:**

The rebuttal has clarified many points for me, particularly regarding the extension of the results to modern architectures.
As I indicated in my response, I support the acceptance of the paper in the conference.

**Key Questions For Authors:**

* One question is about the assumption we have about the practical tasks. How do we know that the practical tasks fall in the current categories that is Holder class?
* In functional analysis, there are various classes of functions used in different conexts (Besov, Sobolev, etc.). Why is the choice of Holder class important? In context of transformer models, which class of functions should be the focus of the theory? For instance, in PDE theory, the focus is not much on Holder class. A theoretical result in modern ML should clarify the importance of such choices.
* In Definition 3.1, clarify if $r$ and $\alpha_0$ are fixed in advance or free parameters summing up to $\alpha$.
* Misc. Something wrong with the hyperlink. (line 122, 145, 149, ...)
* In Theorem  4.1: add a remark and clarify the role of norm constraint $K$ in the number of blocks required for the target approximation error.
* Also the result holds for $d_x=d_y=d_0$. In many cases, this does not hold, for instance in many LLMs, the input dimension is different from the vocabulary size. Please comment on these cases.
* The Step 2 of the proof shows that a single self-attention layer suffices  and the rest of the layers are feed forward layers. This basically means that in the D-layer network is mostly FF with a single transformer. This is in contrast with many practical networks. Can the authors explain the discrepancy, and the contribution of multiple attention layers to the approximation power?
* Regarding the fundamental limit of model’s expressivity: In practice, the model size is chosen using empirical scaling laws (for instance  Kaplan's or Chinchilla scaling law). How does the current result relates to it?
* I want to request a similar clarification regarding the results of Section 6 on regression task. It seems to me that this framework provides a way of theoretically grounding the empirical scaling laws, which show dependence on both model parameters and the training data size. Can authors comments on this connection?
* Since only a single attention layer is used (for contextual mapping), what is the gain of using attention for such mapping compared with using a feedforward layer?
* In Theorem 6.2, maybe a tighter bound can be obtained if $\mu$ is chosen to minimize the term in Theorem 6.1. How does such optimization impact the optimality in relation with Remark 6.4?

**Limitations:**

See above for my questions and comments.

**Strengths And Weaknesses:**

**Strengths**
* The prior works were largely focused on universal approximation theorem results, did not consider softmax directly and did not have residual connection in their analysis. The paper provides an analysis of such general networks as one of the contributions.
* The authors also provide a lower bound on the approximation error using VC dimension analysis, which is widely ignored.

**Weakness**
* One general weakness of such theoretical results, at least the way they are presented, is that it does not provide additional insight to the practitioners. For instance, the strong approximation power of transformers and their effectiveness is known and admitted by the community given its wide adoption.  The authors should really focus on design guidelines, and unexpected insights that the theory provides beyond the current state of knowledge.
* It is not clear whether the finding of the paper holds for many variations of the analyzed architecture. What if we have SwiGLU and GELU and other new activations? Also what about the normalization layers? The modern architectures are different that the work currently presented.
* In section on Regression, the loss function is first chosen as the squared error (eq. 10) but then in eq. 13, the loss function is changed to $\ell_2$ norm. This is inconsistent.
* Following that, in page 26, line 1387/8, the equality   $\inf_{f \in T_R^D} \|| f- f^* \|| = \mathcal{L}(f_{app})-\mathcal{L}(f^*)$  seems incorrect to me. First. it would not work with the squared loss $\mathcal{L}$. Second, even with $\ell_2$-loss, it should be an inequality (from triangle inequality if I am not mistaken). Even with the choice of $\ell_2$-norm, one needs to make sure the rest of the steps are correct.
* The presence of $d_0$ is a bit unintuitive. In practice, $d_0$ can be quite large, which makes the bound vacuous.

---

> ### Author Rebuttal · Authors · 2026-03-29
>
> We thank the reviewer for recognizing our work. **Due to character limit**, we address your points (W1-W5, Q1-Q11) below:
>
> **1. Theoretical clarifications**
> - **Problem formulation**
> 	- **Hölder class (Q1, Q2, Q3):** While Sobolev spaces dominate PDEs, ML prioritizes **pointwise stability** (similar inputs yield similar outputs). Hölder class's extended Lipschitz continuity captures this local geometry, which is standard in robust ML **(Q1, Q2)**. We leave other function spaces to future work, though our analysis naturally extends to broader classes **(Q2)**. In Def 3.1, we will clarify $r=\lfloor\alpha\rfloor$ and $\alpha_0=\alpha-r$ are fixed by $\alpha$, not free parameters **(Q3)**.
> 	- **Dimensions (Q6):**  Setting $d_x=d_y=d_0$ isolates the core architecture for analysis (Line 181 & Fig 1), since Transformers require equal input and output dimensions. For $d_x\neq d_y$, input/output embeddings handle the dimension transformations.
> - **Approximation bounds**
> 	- **$d_0$ dependence (W5):** This dependence is standard for worst-case approximation. $d_0$'s presence in both the upper and lower bounds **reflects the inherent nature** of minimax approximation problem.
> 	- **Low-dimensional structure (W5):** Real-world data often lies on low-dimensional manifolds (intrinsic $d^*\ll d_0$), mitigating the "too large curse". Our bounds readily adapt to this assumption, and still explain empirical success.
> 	- **Role of $K$ (Q5):** $K$ is merely a stability constraint for target function boundedness, and does not affect block number $D$ or error $\varepsilon$. We will add a remark.
> - **Model constructions**
> 	- **Necessity of attention (Q10):** Sequence FFNs process tokens independently, yielding identical values for same tokens. They fail when a token $x$ must map to different $y$ values based on $x$'s context. As noted in Line 287, attention is mathematically essential for cross-token contextual mapping. FFNs cannot fulfill this.
> 	- **Single attention layer sufficiency (Q7):** One attention layer is the theoretical minimum for contextual mapping.
> 	- **Explanation of discrepancy (Q7):** Though LLMs stack attention layers for hierarchical features, our single-layer proof establishes a rigorous worst-case baseline. Extra attention layers preserve or improve approximation power.  We leave multiple attention layers to future work.
>
> **2. Proof justifications & clarifications**
> - **Corrections (W3, W4):** We **apologize for omitting the square** $(\cdot)^2$ on $\mathcal{E}_{app}$ in Eq.(13) and L1387. Restoring it resolves the inconsistency **(W3)** and validates the equality **(W4)**:
> 	- Eq.(13) becomes $\cdots\le\mathcal{E_{sta}}+2\mathcal{E}_{app}^2 + 2\tau$.
> 	- In Line 1387, $\Vert f - f^* \Vert_2^2=\mathcal{L}(f)-\mathcal{L}(f^*)$ is a standard identity for square loss in statistical learning as the cross-term vanishes via the law of total expectation. (Eq.(1.1) in [1])
> 	- [1] Györfi, László, et al. _A distribution-free theory of nonparametric regression_. 2002.
> - **Excess risk & $\mu$ (Q11):**  With squares restored, **Thm 6.2** yields $N^{-\frac{\alpha}{2d_0+\alpha}}\log(N)+N^{-\frac{\alpha}{2d_0+\alpha}}$ with $\mu=D^{-(2\alpha)/d_0}$, which still aligns with Remark 6.4. And Eq. (63) inherently optimizes $\mu$ to balance approximation and statistical errors.
> - **We will correct these in the final version!**
>
> **3. Practical significance**
> - **Insights for practitioners (W1):** Our analysis clarifies Transformers' functional roles: attention for context routing and FFNs for function fitting. This can guide task-specific architectural adjustments (to be discussed in revision).  While Transformers have well-known approximation power, our lower bound characterizes their mathematical limits. We complement existing theory by defining the architecture's theoretical boundaries.
> - **Model variations (W2):** Our construction is a  baseline holding for variations.
> 	- **Activations:** Smoother activations like GELU/SwiGLU can improve ReLU rates. For instance, SwiGLU can directly achieve the multiplication "$\times$", while ReLU FFNs must approximate "$\times$".
> 	- **Normalization:**  LayerNorms can be mathematically folded into weight parameters scales. We omit them for analytical simplicity, a convention in approximation theory.
> - **Scaling law.**
> 	- **Relationship with scaling laws (Q8):** Empirical scaling laws reflect average performances on real-world data, whereas we establish worst-case theoretical bounds. By covering the hardest functions, our bounds are more conservative and may be looser than empirical guidance.
> 	- **Parameter/Data trade-off (Q9):** Section 6 bounds excess risk by sum of approximation error (bias) and statistical error (variance). We derive a dual trade-off on parameter number $N$ and data size $D$.  Though the derived order may not be strictly optimal, it theoretically grounds relationships similar to empirical scaling laws.
>
> **4. Minor issue (Q4):** Broken links will be fixed in the revision.

---

> > ### Author Rebuttal · Reviewer_iFBF · 2026-04-02
> >
> > I would like to thank the authors for their answers. I understand the character limitations and am fine with the selected answers. To summarize my position, I believe the paper has the merit of being accepted and presented in the venue. I also understand that having a theoretical result for the most general practical cases would be the desideratum for a distinguished paper and is currently challenging in general. Nonetheless. I appreciate if the authors can add extensive discussions about the restrictive assumptions of the theory, and its potential implications and connections with practically used architectures.

---

> > > ### Author Response · Authors · 2026-04-06
> > >
> > > We thank the reviewer for their strong support and constructive feedback! To address the reviewer's points, **we will add a discussion in the final version** to clarify the connections and implications between theoretical assumptions and practical architectures. Specifically, we will discuss:
> > >
> > > **1. Activation functions**
> > > - **Theoretical assumption:** **We adopt ReLU in theory**, due to its rich mathematical toolkit for establishing approximation bounds. Furthermore, Yarotsky [1] proves that any continuous piecewise linear activation function with finite breakpoints can be exactly realized by a linear combination of ReLUs. **This supports ReLU's theoretical universality.**
> > > - **Practical connection:**
> > > 	1. **Baseline guarantee:** Real-world models use globally smooth functions (like GELU or SwiGLU) for better gradient flow and stable optimization. Since ReLU networks are capable of approximating any continuous function, our ReLU-based results serve as a theoretical baseline for practical architectures that replace ReLUs with smooth variants.
> > > 	2. **Theoretical justification:** Structurally, SwiGLU incorporates the Hadamard product ($xW_1\otimes(xW_2)$) to directly compute element-wise multiplication. In contrast, standard ReLU networks must approximate multiplication, a result shown in Yarotsky’s Proposition 2 [1]. This distinction highlights that our ReLU analysis defines the core approximation limits, thereby theoretically justifying the use of those smoother activations which possess stronger local expressive power.
> > > - **Ref:** [1] Yarotsky, Dmitry. "Error bounds for approximations with deep ReLU networks." _Neural networks_ 94 (2017): 103-114.
> > >
> > > **2. Normalization**
> > > - **Theoretical assumption:** We omit normalization techniques (LayerNorm) to strictly isolate and analyze the approximation power of the core self-attention and FFN mechanisms.
> > > - **Practical connection:**
> > > 	1. **Unchanged approximation limits:** Crucially, normalization does not expand the approximation power limits established by our results. Mathematically, the magnitude adjustments performed by LayerNorm can be achieved by carefully scaling the adjacent weight parameters.
> > > 	2. **Optimization necessity:** In practice, normalization is essential for resolving optimization difficulties, such as smoothing the loss landscape and stabilizing gradients in deep networks.
> > > 	3. **Distinct research scope:** While optimization is a highly important question, analyzing its dynamics is beyond the scope of our approximation-focused study.
> > >
> > > **3. Attention depth**
> > > - **Theoretical assumption:** In Line 287, we demonstrate that **a single attention layer is the minimal structural requirement** to achieve the contextual mapping. Using a single layer isolates the core mechanism of self-attention and provides a clean and rigorous baseline, where the attention mechanism enables sequence models to grasp contextual meaning in a mathematical setting.
> > > - **Practical connection:**
> > > 	1. **Theoretical grounding:** Real-world models stack multiple layers to efficiently extract hierarchical features. Stacking attention layers strictly preserves or improves the approximation power of the model. Therefore, our single-layer framework represents a rigorous "worst-case" guarantee, theoretically grounding the expressive capacity of other complex, multi-layer architectures.
> > > 	2. **Compositional complexity:** Indeed, deriving bounds for multiple attention layers introduces significant theoretical complexity due to the highly compositional nature of deep Transformers. We leave multi-layer extensions to future work, which can naturally build upon our current results.
> > >
> > > We thank the reviewer again for their constructive feedback, which greatly improves our theoretical framework.

---

### Decision · Program_Chairs · 2026-04-30

**Decision:**

Accept (regular)

**Comment:**

This paper proves that the standard Transformer can approximate any bound Holder function with polynomially many blocks. Both upper and lower bounds for the number of blocks are given, and the two quantities almost match. This is a very interesting theoretical result.